# Genetic gains underpinning a little-known strawberry Green Revolution

Mitchell J. Feldmann ⓘ[1], Dominique D. A. Pincot[1], Glenn S. Cole ⓘ[1] & Steven J. Knapp ⓘ[1] ✉

The annual production of strawberry has increased by one million tonnes in the US and 8.4 million tonnes worldwide since 1960. Here we show that the US expansion was driven by genetic gains from Green Revolution breeding and production advances that increased yields by 2,755%. Using a California population with a century-long breeding history and phenotypes of hybrids observed in coastal California environments, we estimate that breeding has increased fruit yields by 2,974-6,636%, counts by 1,454-3,940%, weights by 228-504%, and firmness by 239-769%. Using genomic prediction approaches, we pinpoint the origin of the Green Revolution to the early 1950s and uncover significant increases in additive genetic variation caused by transgressive segregation and phenotypic diversification. Lastly, we show that the most consequential Green Revolution breeding breakthrough was the introduction of photoperiod-insensitive, *PERPETUAL FLOWERING* hybrids in the 1970s that doubled yields and drove the dramatic expansion of strawberry production in California.

Cultivated strawberry (*Fragaria × ananassa*) has a unique domestication history spanning approximately 300 years[1]. The recent origin of this artificial hybrid species has meant that the near complete genealogy of extinct and living individuals could be reconstructed[2], albeit with sparse pedigree records in the early years (1766–1812) following the discovery by Antoine Nicolas Duchesne[3] that strawberry plants with novel phenotypes observed in the Gardens of Versailles were spontaneous hybrids between non-sympatric octoploid ($2n = 8x = 56$) species (*F. chiloensis* and *F. virginiana*) imported from the Americas. Once the interspecific hybrid origin of these phenotypically unique plants was discovered, the conscious domestication of *F. × ananassa* began with artificial hybridization, the introgression of alleles from wild octoploid ancestors (migration), admixing of the parental genomes (recombination), and artificial selection to improve agriculturally important traits[1–5]. While the horticultural characteristics of the modern hybrids that have since emerged are widely known to transcend those of their forebears, genetic gains for yield and other traits vital for year-round production on a large scale are not well documented[1,6–10].

United Nations Food and Agricultural Organization (UN-FAO) statistics show that 9.2 million tonnes of strawberries were produced worldwide in 2022 and that strawberry production has increased 1075% worldwide since 1960, whereas yields have only increased 187% (https://www.fao.org/faostat/). Hence, to meet increased global demand over the last half-century, the area harvested has increased by 308% worldwide, a trend observed for many other fruits and vegetables because of economic inequities and minimal yield increases, or even yield declines[11,12]. UN-FAO statistics show that strawberry yields have not increased, slightly increased, or greatly increased across different regions and countries since 1960 (https://www.fao.org/faostat/), implying that improved production practices, genetic gains from breeding for increased yield, or both have been geographically uneven. The contributions of breeding to the strawberry yield and production increases and other genetic changes vital for meeting year-round demand have not been explored to a depth or breadth in strawberry comparable to that in maize and other staple crops[8–10,13–19].

Strong directional selection, population bottlenecks, and selective sweeps have profoundly reshaped DNA variation in the genomes

---

[1]Department of Plant Sciences, University of California Davis, One Shields Avenue, Davis, CA 95616, USA. ✉e-mail: sjknapp@ucdavis.edu

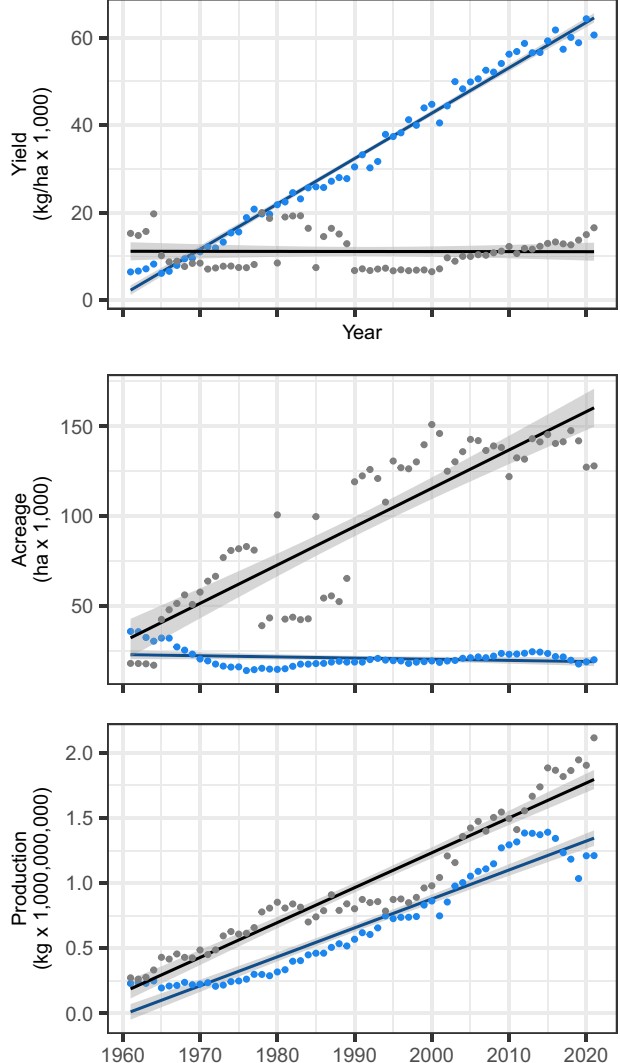

**Fig. 1 | Strawberry production in the US and Europe (1961–2021).** Strawberry yield, area harvested, and production statistics compiled by the Food and Agriculture Organization of the United Nations from 1961 to 2021 for the US and Europe (https://www.fao.org/faostat/en/). Statistics for the US are displayed as solid blue circles, whereas statistics for Europe are displayed as solid gray circles. Linear regression slopes are shown as solid blue lines for the US and solid black lines for Europe with 95% confidence intervals for the predicted values shown as gray bands. Source data are provided as a Source Data file.

of the early and modern strawberry hybrids used in the present study to estimate historical genetic gains from breeding[5]. One of the most unexpected findings from those analyses was that the highest-yielding hybrids reported to date are moderately to highly inbred, e.g., 69% of the DNA was identical-by-descent in the highly contiguous haplotype-phased assemblies of the genome of the modern hybrid Royal Royce, one of the highest-yielding hybrids in our study (https://phytozome-next.jgi.doe.gov/info/FxananassaRoyalRoyce_v1_0). This inbreeding was predicted from genome-wide analyses of single nucleotide polymorphisms (SNPs) among heirloom and modern hybrids, which showed that nucleotide diversity and heterozygosity have steadily declined over the last half-century in highly structured populations that have been the source of elite genetics for large-scale strawberry production in North America[5]. The greatest declines were observed in the historically important California population, which has been shaped by nearly a century of breeding at the University of California, Berkeley (1924–1945) and the University of California, Davis (1952-present)[2,5,9,10,20,21]. That population was the source of the elite Green

Revolution parents and hybrids that we phenotyped to estimate historical genetic gains in strawberry from undomesticated founders to modern cultivars. UC Davis cultivars originating from the California population have had a significant impact, with publicly documented ancestry in at least 764 cultivars worldwide, cultivation in 82 countries, and clonal propagation in 34 countries (https://itc.ucdavis.edu/strawberry-licensing-program/)[2].

Here we show that Royal Royce and other modern California population hybrids produce high yields despite the inbreeding that has accrued in their genomes[5]. Using the phenotypes of modern UC Davis cultivars (asexually propagated hybrid individuals) as a benchmark, we further show that genetic gains from breeding in the California population increased the yields of less perishable fruit, enabled daylength-independent year-round production, and drove a little-known strawberry Green Revolution in the second half of the twentieth century in California that paralleled the widely known Green Revolutions in wheat, rice, and other staple crops[13–16,18,22,23]. We use genetic relationships among early and modern cultivars, important founders, and other ancestors to statistically model changes in population means, breeding values, and genetic variances over the history of cultivated strawberry domestication (1775-present). This analysis sheds light on genetic changes associated with Green Revolution breeding that were both expected and unexpected. We estimate the breeding values of several thousand individuals from the phenotypes of training population hybrids using single-step best linear unbiased prediction (ss-BLUP)[24,25]. We combine the pedigree records of extinct and living individuals with SNP genotypes of living individuals to estimate a unified additive genetic relationship matrix that connects the past with the present[24,25]. Our time-series analyses of genetic changes over the last 250 years are modern by archaeogenetic standards[26–28], with pedigree records substituting for the ancient DNA of extinct individuals spanning the recorded history of strawberry domestication. Using genomic prediction methods for simulating segregating populations[29], we show that modern breeding has greatly increased additive genetic variation in the California population despite breeding-associated declines in heterozygosity and nucleotide diversity[5], a finding important for understanding and interpreting the effects of modern breeding on the maintenance and erosion of genetic variation[30]. Finally, we discuss the sustainability of the strawberry yield and production increases that have accrued over the course of the Green Revolution.

## Results

### The emergence of improved photoperiod-insensitive hybrids doubled strawberry yields in California

The most consequential Green Revolution breeding breakthrough was the introduction of photoperiod-insensitive (day-neutral) cultivars in the 1970s that laid the foundation for doubling yields and became the backbone for the expansion of strawberry production in California and the US (Figs. 1–2; Supplementary Fig. 1). Their development, initiated by Royce S. Bringhurst and Victor Voth[31] in 1953 using hybrids between photoperiod sensitive cultivars and a photoperiod-insensitive *F. virginiana* subsp. *glauca* ecotype (Wasatch), ultimately paved the way for year-round production[7,9,10]. Even they did not foresee the impact of this breakthrough, stating that their first day-neutral cultivars were not "without serious faults" and "should be considered as candidates for use by home gardeners of California"[20,31].

Despite their early skepticism, increasingly more productive day-neutral (photoperiod-insensitive) cultivars soon emerged that reshaped the strawberry production landscape in California (Figs. 1–2)[7,9,10,20]. Those cultivars were later discovered to carry *PERPETUAL FLOWERING* (PF), a dominant gene that mediates daylength-independent flowering under a wide range of latitudes and temperatures and pleiotropically affects asexual reproduction[32–34]. Analyses of pedigree records and SNPs in linkage disequilibrium with the *PF* locus

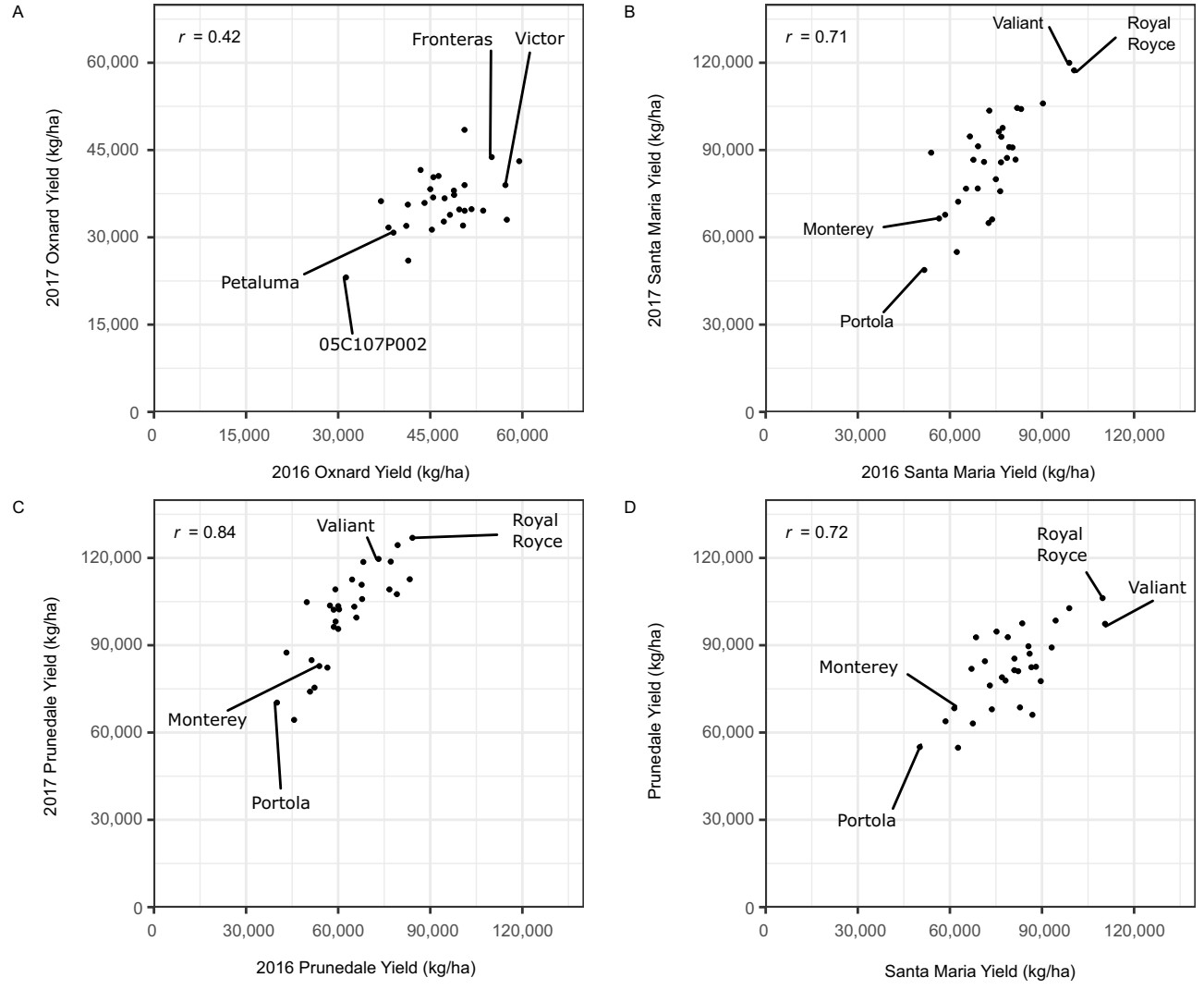

**Fig. 2 | Cumulative marketable fruit yields of modern short-day and day-neutral cultivars (hybrids) grown in coastal California locations over the 2015–16 and 2016–17 growing seasons.** Estimated marginal means (EMMs) for cumulative fruit yields were estimated from 46 to 52 individual harvests of 30 short-day cultivars grown in Oxnard, CA (**A**) and 30 day-neutral cultivars grown in Santa Maria, CA (**B**) and Prunedale, CA (**C**). **A** Yield EMMs for short-day cultivars from the 2016 and 2017 harvest seasons in Oxnard, CA. **B** Yield EMMs for day-neutral cultivars from the 2016 and 2017 harvests seasons in Santa Maria CA. **C** Yield EMMs for day-neutral cultivars from the 2016 and 2017 harvest seasons in Prunedale, CA. **D** Between-year EMMs for day-neutral cultivars from the 2016 and 2017 harvest seasons in Santa Maria and Prunedale, CA.

suggest that photoperiod-insensitive Wasatch descendants in the California population, including the UC Davis cultivars and other California population parents and hybrids phenotyped in our studies, inherited the *PF* allele (Figs. 2–4).

To benchmark the full-season on-farm yields of modern California population cultivars, and document the impact of *PF* on strawberry production, short-day (*pfpf*) and day-neutral (*PF_*) cultivars and other hybrids developed at UC Davis over the last 30 years were phenotyped over two growing seasons (2015–16 and 2016–17) in the three coastal regions (target environments) where strawberries are commercially grown in California: Oxnard (34.20° N) for short-day cultivars and Santa Maria (34.95° N) and Prunedale (36.78° N) for day-neutral cultivars (Fig. 2; Supplementary Fig. 1). Yields were estimated from fruit harvested on regular commercial schedules over the entire growing season on each farm (the number of harvests ranged from 46 to 52/farm/season). The hybrids tested in these studies included every commercially important UC Davis cultivar and several other elite CA population hybrids.

The fruit yields of hybrids were stable across environments (Fig. 2). Hybrid × environment interactions only accounted for 10.6–15.6% of the phenotypic variance for yield. The between-year yield rank changes of hybrids were small and inconsequential, between-year rank correlations were positive (0.42 to 0.84), and broad-sense heritabilities ranged from 0.70 to 0.79 among short-day hybrids and 0.84 to 0.92 among day-neutral hybrids (Fig. 2; Supplementary Fig. 1). While the scope of our study was limited by the number of environments sampled (three locations × two years), and hybrid × environment interaction variances could have been underestimated in our study, our findings are consistent with those reported in several other studies where yield rank changes have been insignificant among hybrids across locations, years, and production systems[9,10,35]. The infrequency of significant cross-over hybrid × environment interactions for yield appears to be the rule not the exception for California population hybrids grown in coastal California environments, as exemplified by our study (Fig. 2; Supplementary Fig. 1) and previous studies over at least two decades and eight coastal

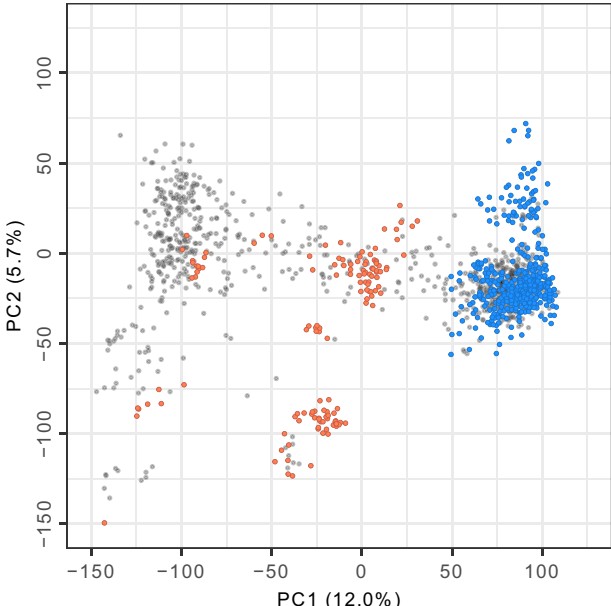

**Fig. 3 | Genetic diversity among early and modern strawberry hybrids.** Genetic relationships were estimated among 1406 hybrids using 28,513 single nucleotide polymorphisms. The birth years of these hybrids ranged from 1775 to 2016. The first two scores (PC1 and PC2) from a principal component analysis of the genomic relationship matrix (G) are plotted for three different groups of genotyped individuals: 405 elite × elite hybrids phenotyped for fruit yield and quality traits (blue points); 131 elite × exotic and eight exotic × exotic hybrids phenotyped for fruit yield and quality traits (coral points); and a genetically diverse collection of 434 California and non-California cultivars and other hybrids without phenotypes developed between 1775 and 2015 (gray points). Source data are provided as a Source Data file.

California locations, from Irvine (33.7° N, 117.8° W) to Watsonville (36.9° N, 121.8° W)[9,10,35].

The introduction of improved photoperiod-insensitive cultivars doubled strawberry yields in California (Fig. 2; Supplementary Fig. 1). The cumulative marketable fruit yields of day-neutral hybrids (79,921 kg/ha) were 93.9% greater than short-day hybrids (41,208 kg/ha) across years and farms ($p \leq 0.0001$). The yield of Royal Royce (107,759 kg/ha), the highest-yielding day-neutral hybrid in our study, was more than double that of 10C144P002 (50,761 kg/ha), the highest-yielding short-day hybrid ($p \leq 0.0001$). The single highest yield recorded in our study was 126,918 kg/ha for the cultivar Royal Royce.

The yield increases associated with the *PF*-mediated daylength-independent flowering phenotype are a consequence of longer harvest seasons and fruit production in months when daily temperatures, solar radiation, and growing degree days are greater compared to those for the *pfpf*-mediated seasonal flowering phenotype of short-day hybrids (Fig. 2; Supplementary Fig. 1). These seasonal differences were reflected in the per harvest yields of day-neutral hybrids (3979 kg/ha/harvest), which were nearly double (88.2% greater than) those of short-day hybrids (2114 kg/ha/harvest) across farms and years ($p \leq 0.001$).

### Strawberry yields have increased 2755% in the US since 1960

The California population has had a nearly uninterrupted century-long breeding history and been the source of several historically and commercially important cultivars with improved characteristics critical for the expansion of large-scale production in California, the source of 88–90% of US-produced strawberries (https://www.nass.usda.gov/)[2,20]. That expansion began slowly in 1945 with the release of Lassen, an heirloom cultivar developed at UC Berkeley in the 1920s[20]

and grandparent of several of the elite × exotic hybrids developed for the present study (Fig. 3). Our analyses show that genetic gains from breeding in the California population drove a Green Revolution in the US where strawberry yields have increased 2755% and annual production has increased by one million tonnes since 1960 (Figs. 1 and 4; Table 1).

The production increase in the US was achieved by introducing short-day and day-neutral cultivars with increased yields of less perishable fruit (Figs. 2 and 4; Table 1; Supplementary Fig. 1), optimization of bare-root plant propagation systems, and widespread adoption of chemically intensive production practices, most importantly soil fumigation with methyl bromide between 1960 and 2005, which effectively prevented losses to diseases caused by soil-borne pathogens and facilitated decades of breeding for increased yield with minimal below-ground disease pressure[10,20,36,37]. To put the Green Revolution documented here in a broader global perspective, 18 of the 20 highest annual country-wide strawberry yields reported by the UN-FAO since 1961 were in the US between 2014 and 2021 (https://www.fao.org/faostat/).

### Genetic gains in a strawberry population with a century-long breeding history

With the yields of modern short-day and day-neutral cultivars benchmarked (Fig. 2; Supplementary Fig. 1), we undertook studies to assess the genetic gains from breeding for yield and other agriculturally important traits in strawberry (Figs. 3–4)[2,20,21]. Using an incomplete factorial mating design with 27 elite CA population parents and three exotic parents (Puget Reliance, Oso Flaco, and Del Norte), we developed 405 elite × elite, 132 elite × exotic, and eight exotic × exotic hybrids with parents spanning the domestication range, from North American ecotypes of octoploid beach strawberry (*F. chiloensis*) that seldom produce fruit (e.g., Del Norte) to the highest-yielding modern cultivars, e.g., the day-neutral cultivar Royal Royce and the short-day cultivar Victor (Fig. 2). The elite parents were selected to survey standing genetic variation in the year 2015 CA population, whereas the exotic parents were selected to sample progressively greater doses of wild species alleles, from 0% for the cultivar Puget Reliance to 50% for Oso Flaco to 100% for Del Norte (Fig. 3; see "Methods" section).

We developed fresh, seed-propagated hybrids to reset and synchronize clone-age clocks and circumvent any clone-age biases that might arise from physiological or genetic changes in the vigor of hybrids caused by differences in the number of generations of asexual reproduction in situ. Hybrids were phenotyped in a single coastal California location (Salinas, CA) through the summer solstice in the 2017–18 and 2018–19 growing seasons (Fig. 4; Table 1). The last harvest (within a few days of the summer solstice) was timed each year to ensure that yields were not affected by photoperiod sensitivity differences among hybrids (segregation of the *PF* locus). Using a SNP in linkage disequilibrium with the *PF* locus, 205 hybrids were predicted to be short-day (*pfpf*), whereas 337 hybrids were predicted to be day-neutral (*PF_*). The statistics reported here were estimated for eight traits from 148,519 phenotypic observations among parents and training population hybrids (Table 1).

We used two low-hybrid phenotypic means (EMM₂) in contrasts ($\Delta G = \text{EMM}_1 - \text{EMM}_2$) to estimate historical genetic gains in strawberry, where the high-hybrid phenotypic mean (EMM₁) was the same for each trait (Table 1). First, we estimated the difference between the pair of elite × elite (California population) hybrids with the lowest and highest phenotypic means for each trait. Second, we estimated the difference between the elite × elite hybrid with the highest phenotypic mean (EMM₁) and elite × wild (elite × Del Norte) hybrid with the lowest phenotypic mean (EMM₂) for each trait. Using these contrasts, we estimated that Green Revolution breeding in the CA population increased fruit yields by 2974–6636%, counts by 1454–3940%, weights by

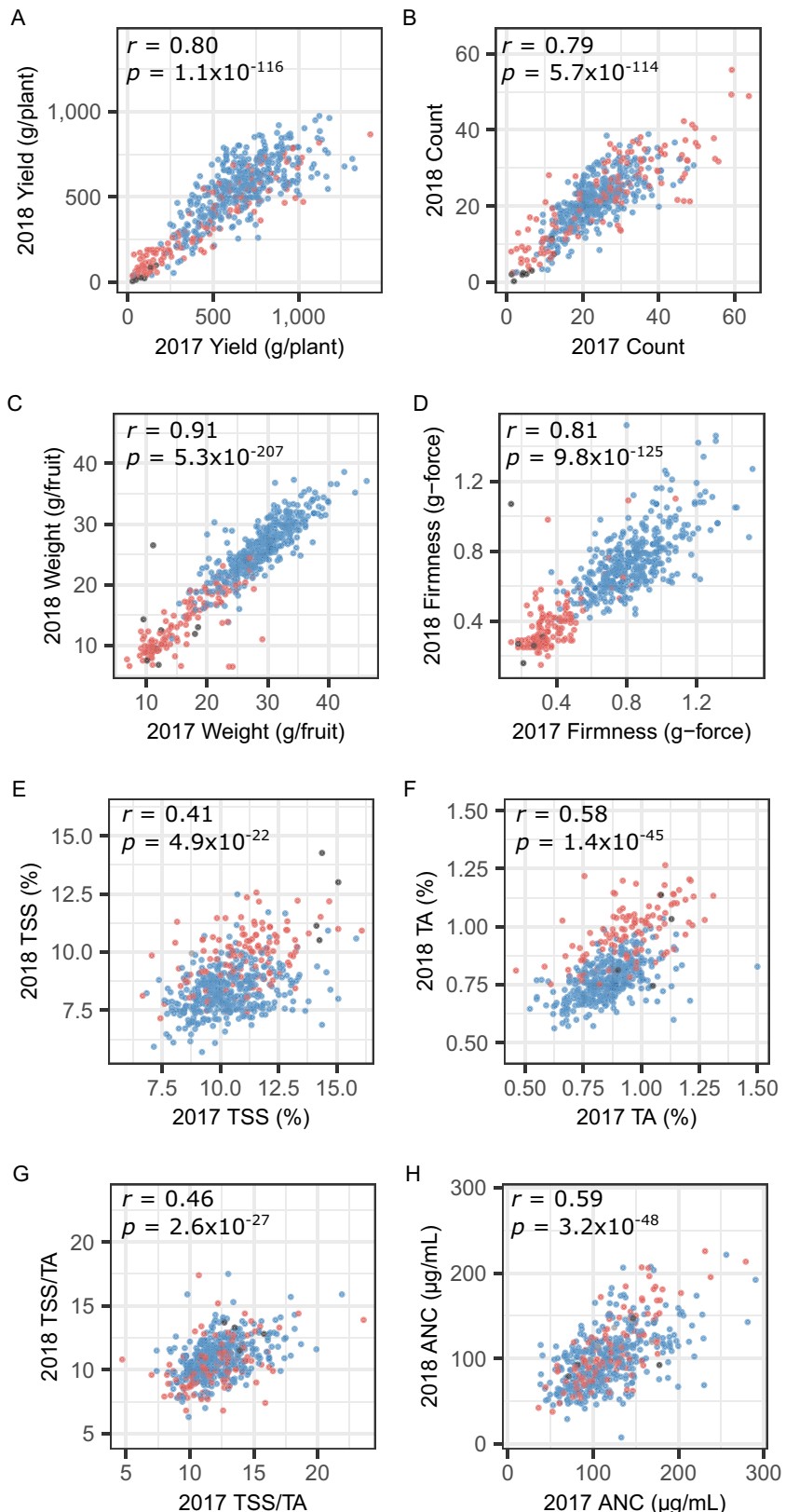

**Fig. 4 | Phenotypic means of training population hybrids.** Within-year estimated marginal means (EMMs) for fruit yield (**A**), count (**B**), weight (**C**), firmness (**D**), total soluble solids (**E**; TSS), titratable acidity (**F**; TA), TSS/TA (**G**), and anthocyanin concentration (**H**; ANC) among 405 elite × elite hybrids (blue points), 132 elite × exotic hybrids (red points), and 8 exotic × exotic hybrids (black points) phenotyped in Salinas over the 2016–17 and 2017–18 growing seasons. EMMs were estimated from fruit harvested from three replicates once per week for 11 to 13 weeks through the summer solstice each year. The between-year rank correlations (*r*) were positive and statistically significant for every trait. Source data are provided as a Source Data file.

## Table 1 | Historical genetic gains

| Trait[a] | $\hat{H}^{2b}$ | $\hat{\sigma}^2_{G \times Y}/\hat{\sigma}^2_P{}^c$ | Benchmark[d] | EMM$_1$ | EMM$_2$ | ΔG | ΔG (%)[e] | $Pr > t$[f] |
|---|---|---|---|---|---|---|---|---|
| Yield (g/plant) | 0.88 | 0.06 | Elite × Elite | 1091.3 | 36.7 | 1054.6 | 2873.6 | 0.003 |
| | | | Elite × Wild | | 16.2 | 1075.1 | 6636.4 | 0.002 |
| Count | 0.92 | 0.04 | Elite × Elite | 40.4 | 2.6 | 37.8 | 1453.9 | 0.002 |
| | | | Elite × Wild | | 1.0 | 39.4 | 3940.0 | 0.002 |
| Weight (g/fruit) | 0.93 | 0.02 | Elite × Elite | 41.7 | 12.7 | 29.0 | 228.3 | 0.004 |
| | | | Elite × Wild | | 6.9 | 34.8 | 504.3 | 0.006 |
| Firmness (kg-force) | 0.75 | 0.04 | Elite × Elite | 1.39 | 0.41 | 0.98 | 239.0 | 0.006 |
| | | | Elite × Wild | | 0.16 | 1.23 | 768.8 | 0.003 |
| TSS (%) | 0.59 | 0.13 | Elite × Elite | 15.2 | 5.5 | 9.7 | 176.4 | 0.110 |
| | | | Elite × Wild | | 7.4 | 7.8 | 105.4 | 0.146 |
| TA (%) | 0.82 | 0.07 | Elite × Elite | 1.16 | 0.57 | 0.59 | 103.5 | 0.096 |
| | | | Elite × Wild | | 0.64 | 0.52 | 81.3 | 0.094 |
| TSS/TA | 0.70 | 0.06 | Elite × Elite | 18.9 | 6.2 | 12.7 | 204.8 | 0.020 |
| | | | Elite × Wild | | 7.7 | 11.2 | 145.5 | 0.011 |
| Anthocyanin (ng/mL) | 0.75 | 0.00 | Elite × Elite | 241.8 | 47.8 | 194.0 | 405.9 | 0.020 |
| | | | Elite × Wild | | 45.4 | 196.4 | 432.6 | 0.019 |

Genetic gains (ΔG = EMM$_1$–EMM$_2$) for cumulative marketable fruit yield and quality traits estimated from linear contrasts among the estimated marginal means (EMMs) for 545 strawberry hybrids phenotyped in Salinas, California over the 2017-18 and 2018-19 growing seasons. EMM$_1$ and EMM$_2$ are the highest and lowest EMMs from linear mixed model analyses of phenotypic observations collected from 10-13 harvests/year and three clonal replicates/hybrid/year.
[a]TSS = total soluble solids and TA = titratable acidity.
[b]REML estimate of the broad-sense heritability on a clone-mean basis.
[c]Ratio of REML estimates of the genotype-by-year interaction variance to the phenotypic variance on a clone-mean basis.
[d]Two ΔG estimates are shown for each trait, one using the low elite × elite hybrid as a benchmark and another using the low elite × wild hybrid as a benchmark. The high hybrid was the same for both contrasts.
[e]ΔG (%) = (EMM$_1$–EMM$_2$)/EMM$_2$ × 100.
[f]The probability of a greater $t$-statistic by chance for tests of the null hypothesis of no genetic gain (H$_0$: EMM$_1$–EMM$_2$ = 0).

228–504%, and firmness by 239–769% (Table 1). The genetic gain estimates for both low-hybrid benchmarks were strikingly similar for every trait and statistically significant from zero ($p \leq 0.05$) for every trait except total soluble solids (TSS) and titratable acidity (TA). The differences in percent genetic gain estimates between the two low-hybrid benchmarks were substantial because of small differences in the divisors (EMM$_2$), even though their numerators (EMM$_1$–EMM$_2$) were virtually identical (Table 1).

The yield stability in our genetic gain study was even greater than that observed in our modern hybrid benchmarking study (Figs. 2 and 4; Supplementary Fig. 1). Hybrid × year interactions accounted for a meager 2–6% of the phenotypic variance on a clone-mean basis, between-year rank correlations ranged from 0.79 to 0.91, and broad-sense heritability estimates ranged from 0.88 to 0.93 for fruit yield, count, weight, and firmness, the four traits that drove the Green Revolution in California (Fig. 4; Table 1). We caution that these estimates could be upwardly biased because of the underestimation of hybrid × environment interactions. Nevertheless, the percent genetic gains estimated by contrasts between hybrids from opposite tails of the yield distribution (EMM$_1$–EMM$_2$) were similar to the yield increase reported by the UN-FAO between 1961 and 2021 in the US (Figs. 1 and 4; Table 1; https://www.fao.org/faostat/). The UN-FAO data alone indicate that the yields of pre-Green Revolution genetic founders of the California population, as exemplified by cultivars developed before 1960 and the elite × exotic hybrids we tested, were 2,755% lower than hybrids widely grown in the US today (Fig. 1; https://www.fao.org/faostat/). Hence, while our single location estimates must be interpreted with caution, and additional studies are needed to corroborate our findings, the trends and estimates reported here are sound first approximations of the fruit yield, count, weight, and firmness increases that have accrued from breeding in the California population since the 1920s when the UC cultivar Lassen was developed.

## Genomic prediction of breeding values pinpointed the origin of the Green Revolution to the 1950s

The hybrids we developed and phenotyped for estimating historical genetic gains (Fig. 4; Table 1) were used as a training population to predict the breeding values of individuals spanning the history of strawberry breeding (Fig. 5; Table 2; Supplementary Figs. 2–3). Genomic-estimated breeding values (GEBVs) were estimated for phenotyped and unphenotyped training population hybrids by combining pedigree and genotypic data in an ss-BLUP analysis (Fig. 5)[24,25,38]. We constructed a unified additive genetic relationship matrix (H) for that analysis by integrating the numerator relationship matrix (A) for 15,649 individuals (extinct or living) with the genomic relationship matrix (G) for 4054 individuals genotyped with a 50K Axiom® SNP array[38,39]. The genotyped individuals included our training population parents and hybrids and a diverse collection of heirloom and modern cultivars and other genetic resources from around the world, the selection of which was informed by breeding history, geography, coancestry, genome-wide analyses of DNA variants, and phenotypes observed in the coastal environments where strawberries are produced in California (Fig. 3)[1,2,5,10,20]. The pedigree database was built by expanding a previously constructed pedigree database[2] that includes every cultivar and several hundred other hybrids with public pedigree records and their ancestors tracing back to 1775, the earliest date for which pedigree records have been recovered.

The birth years for 6442 of the 15,649 individuals in our ss-BLUP analyses were known from pedigree records (796 were genotyped with the SNP array and 5646 were not). GEBVs for those individuals were regressed on birth years to model changes in phenotypic means and ranges from the late 1700s to present (Fig. 5; Table 2; Supplementary Figs. 2–3). The regression lines plotted in Fig. 5 are predicted GEBVs (population means) from piecewise linear regression and change-point (CP) analyses using birth year as the independent variable[40]. Those analyses identified a single change-point year for each trait where

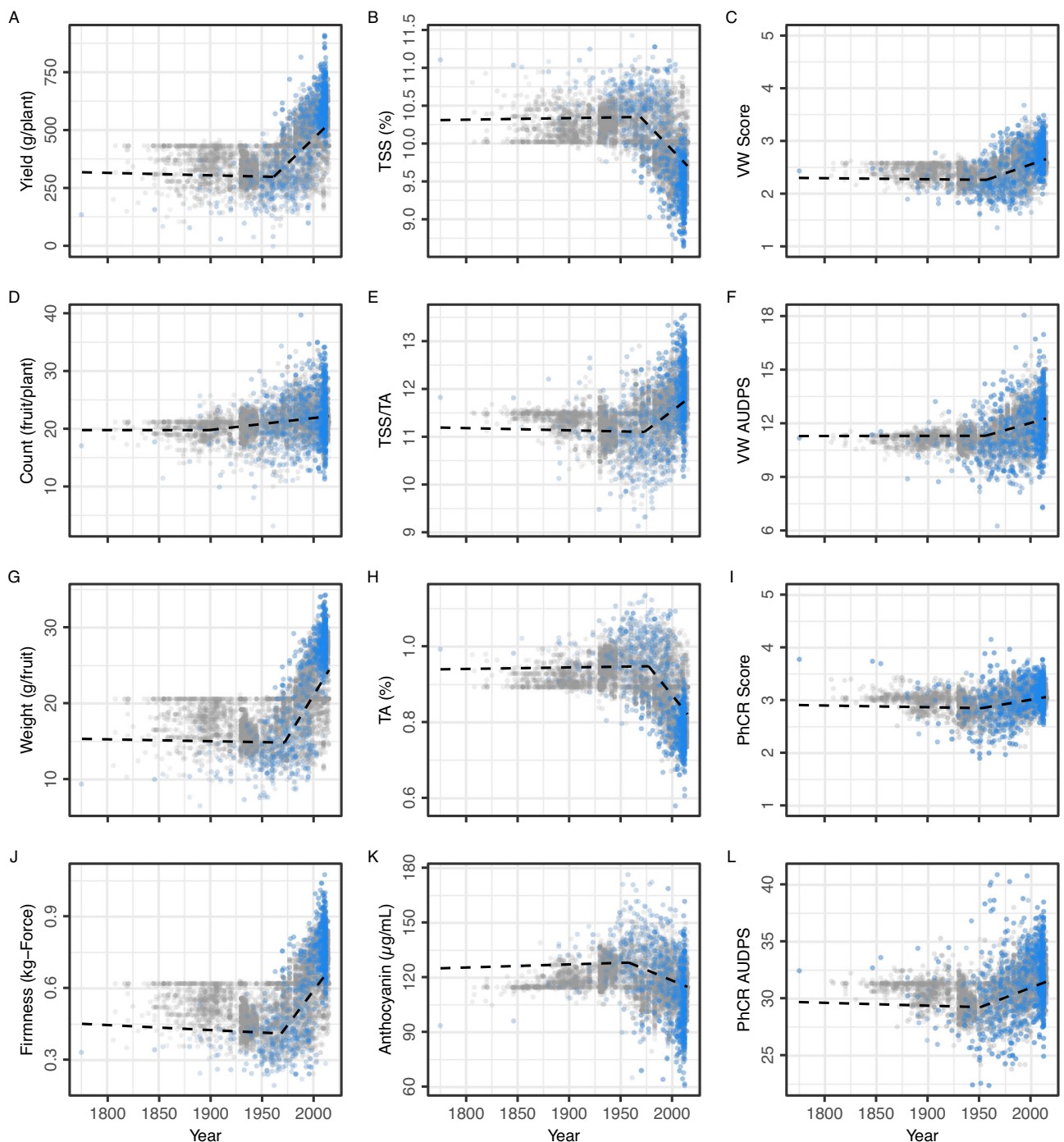

**Fig. 5 | Genomic prediction of breeding values.** Genomic-estimated breeding value (GEBV) estimates are shown for agriculturally important traits among individuals spanning the domestication history of strawberry (1775-present). GEBVs were estimated by single-step best linear unbiased prediction (ss-BLUP) from pedigree (**A**) and genomic (**G**) relationship matrices for 796 genotyped individuals (shown in blue) and 5646 non-genotyped individuals (shown in gray) with known birth years. The dashed lines depict the predicted values (population means) from piecewise linear regressions of GEBVs on birth years before and after change-point (CP) years (1775 to CP and CP to 2015). Change-point years ranged from 1943 to 1962 for different traits (see Table 2). Statistics are shown in the left-hand column of plots for fruit yield (**A**), count (**D**), weight (**G**), and firmness (**J**). Abbreviations for fruit quality traits displayed in the center column of plots (**B**, **E**, **H**, and **K**) are TSS = total soluble solids and TA = titratable acidity. Abbreviations for disease resistance traits displayed in the right-hand column of plots (**C**, **F**, **I**, and **L**) are Verticillium wilt (VW) resistance score and area under the disease pressure stairs (AUDPS) and Phytophthora crown rot (PhCR) resistance score and AUDPS. The ordinal resistance scores for both ranged from highly resistant (1) to highly susceptible (5). Source data are provided as a Source Data file.

population means shifted from unchanged to increasing or decreasing as a function of time (Fig. 5; Supplementary Figs. 2–3). Separate (piecewise) linear regressions applied to GEBVs for individuals born before and after change-point years showed that population means were unchanged from 1775 to the CP year, then sharply increased or decreased after the CP year for every trait (Fig. 5; Table 2). The sign of the post-CP year linear regression slope was a function of the direction of selection and additive genetic correlations among traits (Fig. 6).

**Table 2 | Genomic prediction of domestication-associated breeding value changes in strawberry**

| Statistic[a] | Trait[b] | CP Year[c] | Estimate | | | Change (%)[d] | | Change/Year (%/Year) | |
|---|---|---|---|---|---|---|---|---|---|
| | | | 1775 | CP | 2015 | 1775 to CP | CP to 2015 | 1775 to CP | CP to 2015 |
| $\hat{y}$ | Yield (g/plant) | 1961 | 467.71 | 450.94 | 680.56 | −3.59 | 50.92 | −0.02 | 0.95 |
| | Count | 1893 | 21.52 | 21.31 | 23.80 | −0.98 | 11.71 | −0.01 | 0.10 |
| | Weight (g/fruit) | 1971 | 20.86 | 20.12 | 29.74 | −3.56 | 47.79 | −0.02 | 1.09 |
| | Firmness (kg-force) | 1969 | 0.58 | 0.54 | 0.81 | −6.59 | 48.92 | −0.03 | 1.06 |
| | TSS (%) | 1969 | 9.81 | 9.86 | 9.21 | 0.43 | −6.55 | 0.00 | −0.14 |
| | TA (%) | 1978 | 0.89 | 0.89 | 0.77 | 0.70 | −14.14 | 0.00 | −0.38 |
| | TSS/TA | 1976 | 11.31 | 11.22 | 11.91 | −0.80 | 6.19 | 0.00 | 0.16 |
| | Anthocyanin (ng/mL) | 1959 | 117.84 | 120.45 | 107.13 | 2.22 | −11.06 | 0.01 | −0.20 |
| | VW Resistance Score | 1957 | 2.49 | 2.44 | 2.84 | −2.15 | 16.29 | −0.01 | 0.28 |
| | VW Resistance AUDPS | 1956 | 10.92 | 10.92 | 11.89 | 0.00 | 8.86 | 0.00 | 0.15 |
| | PhCR Resistance Score | 1953 | 3.00 | 2.94 | 3.15 | −2.07 | 7.16 | −0.01 | 0.12 |
| | PhCR Resistance AUDPS | 1952 | 30.39 | 29.92 | 32.14 | −1.53 | 7.42 | −0.01 | 0.12 |
| $\sqrt{(y-\hat{y})^2}$ | Yield (g/plant) | 1940 | 50.84 | 49.81 | 98.92 | −2.02 | 98.58 | −0.01 | 1.32 |
| | Count | 1934 | 1.19 | 1.39 | 2.45 | 16.93 | 76.44 | 0.11 | 0.95 |
| | Weight (g/fruit) | 1940 | 2.10 | 1.78 | 3.70 | −15.07 | 107.55 | −0.09 | 1.43 |
| | Firmness (kg-force) | 1939 | 0.06 | 0.06 | 0.10 | −4.05 | 73.00 | −0.02 | 0.96 |
| | TSS (%) | 1937 | 0.16 | 0.13 | 0.33 | −17.24 | 146.61 | −0.11 | 1.87 |
| | TA (%) | 1930 | 0.03 | 0.03 | 0.06 | 1.39 | 122.15 | 0.01 | 1.44 |
| | TSS/TA | 1897 | 0.19 | 0.20 | 0.40 | 5.21 | 101.88 | 0.04 | 0.86 |
| | Anthocyanin (ng/mL) | 1927 | 5.88 | 5.79 | 9.93 | −1.62 | 71.57 | −0.01 | 0.82 |
| | VW Resistance Score | 1932 | 0.13 | 0.13 | 0.20 | 0.89 | 61.60 | 0.01 | 0.75 |
| | VW Resistance AUDPS | 1930 | 0.24 | 0.27 | 0.90 | 13.83 | 230.60 | 0.09 | 2.72 |
| | PhCR Resistance Score | 1984 | 0.13 | 0.15 | 0.11 | 16.24 | −29.46 | 0.08 | −0.94 |
| | PhCR Resistance AUDPS | 1924 | 0.97 | 1.04 | 1.31 | 7.30 | 26.11 | 0.05 | 0.29 |

Genomic-estimated breeding values (GEBVs) for several agriculturally important traits were estimated by single-step pedigree-genomic BLUP among 6,419 strawberry hybrids developed between 1775 and 2015. The predicted values ($\hat{y}$) from linear regressions of GEBVs on birth years are shown for the year 1775, the estimated CP year for each trait, and the year 2015. Changes in predicted GEBV values ($\Delta\hat{y}$) are shown in the upper half of the table (see Fig. 5), whereas changes in GEBV residuals ($y − \hat{y}$) are shown in the lower half of the table (see Supplementary Fig. 2) for time periods before and after estimated CP years for each trait (1775 to CP and CP to 2015). Source data are provided as a Source Data file.

[a]Residuals ($y − \hat{y}$) from piecewise linear regressions of GEBVs on birth years were transformed to obtain positive values by taking the square root of the squared residuals ($\sqrt{(y − \hat{y})^2}$).

[b]*TSS* total soluble solids and *TA* titratable acidity. The acronyms for disease resistance phenotypes are Verticillium wilt (VW) resistance score and area under the disease pressure stairs (AUDPS) and Phytophthora crown rot (PhCR) resistance score and AUDPS, where resistance scores ranged from highly resistant (1) to highly susceptible (5).

[c]Change-point (CP) years were estimated by piecewise linear regressions of GEBVs on birth years for each trait.

[d]Percent change was estimated from the population means using (CP−1775)/1775 × 100 for 1775 to the CP year and (2015−CP)/CP × 100 for the CP year to 2015.

While a statistically supported CP year was not found for fruit count (Fig. 5), breeding in the CA population has significantly increased the number of fruit produced per plant since the 1950s with percent genetic gains in the 1454 to 3940% range (Table 1). Count was positively genetically correlated with yield and was under indirect selection, whereas yield, weight, and firmness were under direct selection (Figs. 5–6).

The estimated change-point years were conspicuous, concordant with breeding history, and pinpointed the origin of the strawberry Green Revolution in California to the 1950s when Royce S. Bringhurst and Victor Voth pioneered breeding and bare-root plant propagation improvements at UC Davis (Fig. 5; Table 2)[1,7,20,36]. As we showed earlier, the introduction of cultivars carrying the Wasatch *PF* gene doubled strawberry yields in California and had an impact on strawberry production in the US as profound as that of the dwarfing (short-stemmed) genes that underpinned the wheat Green Revolution[9,10,17,31,35]. Although widely unknown, their accomplishments chronologically paralleled and were technically comparable to those of Norman Borlaug and others that drove the influential Green Revolutions in wheat and other staple crops[15,16,23]. The societal impacts of the strawberry Green Revolution were modest by comparison, but influential nonetheless and symbolic of the contributions of plant breeding towards diversifying the year-round supply of fruits and vegetables, decreasing malnutrition, improving human health, and increasing food security[11,12].

**The inclusion of extinct individuals increased historical depth while sacrificing genomic prediction accuracy**

We applied ss-BLUP in a rearward-facing direction (back through the history of strawberry breeding), primarily to shed light on the timing, direction, and scale of Green Revolution-associated genetic changes (gains or losses) in the California population, as opposed to the forward-facing direction for which these methods were originally developed for genomic selection in animal and plant breeding[25,41–43]. The strength of our approach was that the domestication history of strawberry could be modeled on a more comprehensive scale than would have been possible from the genotypes and phenotypes of living individuals alone, albeit with greater uncertainty for ungenotyped individuals the further back in time we went (Fig. 5; Supplementary Fig. 2). While a significant fraction of the individuals needed to connect the past and present were extinct, their genetic relationships were known and could be estimated from pedigree records and connected through DNA-informed estimates of genetic relationships among 10- to 247-year-old individuals preserved in clonal genetic resource collections (Fig. 3). The weakness of our approach was that the inclusion of extinct individuals meant that their genetic relationships could only be estimated from pedigree records (family history), which varied in depth and completeness[24,25,44].

While the inclusion of extinct (ungenotyped) individuals (gray points in Fig. 5 and Supplementary Figs. 2–3) increased the historical

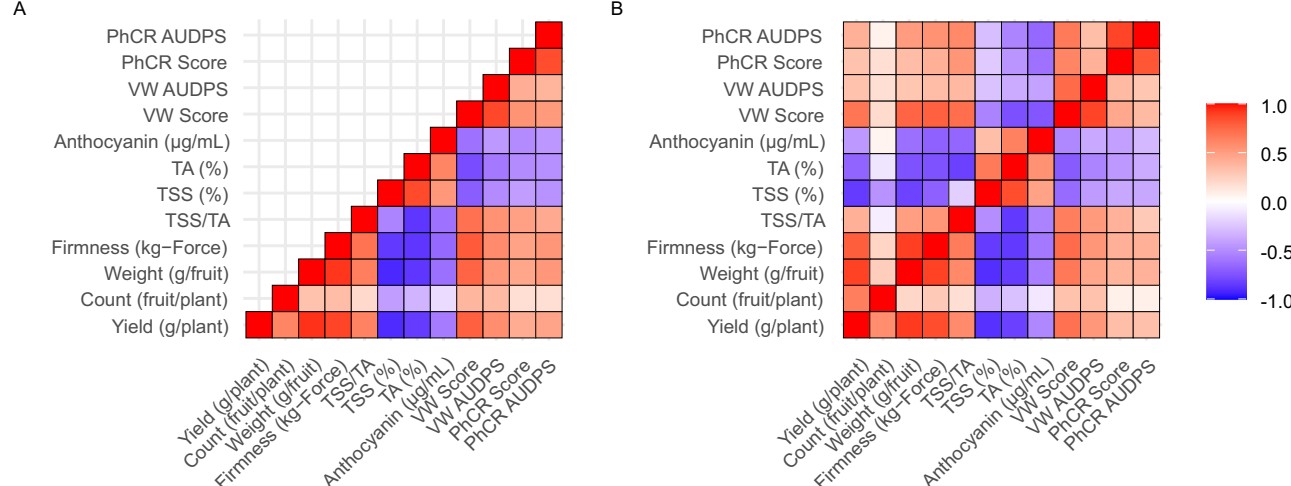

**Fig. 6 | Additive genetic correlations.** Additive genetic correlations among fruit and disease resistance traits were estimated from genomic-estimated breeding values (GEBVs) of 796 genotyped individuals and 5646 non-genotyped individuals with known birth years (see Fig. 5). **A** The lower triangle displays estimates for individuals with birth years between 1775 and 2015. **B** The upper triangle displays estimates for individuals originating before 1954, whereas the lower triangle displays estimates for individuals originating after 1954. Source data are provided as a Source Data file.

depth and completeness our analyses, GEBVs were more accurately estimated from genotyped than ungenotyped individuals (blue points in Fig. 5 and Supplementary Figs. 2–3). GEBVs with lower accuracy were more strongly shrunk to the population mean (Fig. 5). Accuracy decreased and shrinkage of GEBV estimates towards the population mean increased for ungenotyped individuals as the shallowness and disconnectedness of their pedigree records increased (Supplementary Fig. 2). This was primarily caused by the sparsity of information (shallowness and disconnectedness of pedigrees) for estimating the genetic relationships of older, ungenotyped individuals in the genealogy[24,25,38,44,45].

Our conclusions about historical genetic gains and losses in the California population were drawn from phenotypes directly observed among elite × elite and elite × exotic training population hybrids per se (Table 1; Fig. 4), not from ss-BLUP estimates of GEBVs that incorporated ungenotyped individuals (Table 2; Fig. 5; Supplementary Fig. 2). The inclusion of ungenotyped individuals (gray points in Fig. 5) decreased the steepness of the slopes of GEBV-on-birth year regressions in post-CP years because of increased shrinkage towards the population mean, thereby underestimating genetic gains and losses (Table 2; Supplementary Fig. 1). The exclusion of ungenotyped individuals sacrificed historical depth for increased accuracy of GEBV estimates (Supplementary Fig. 2). GEBVs estimated by G-BLUP per se from genotyped individuals only more accurately depict within- and between-year breeding value variation and ranges in the California population (blue points in Fig. 5 and Supplementary Figs. 2–3).

**Green Revolution breeding decreased perishability, sweetness, and acidity**
Our analyses show that a half-century of direct selection for increased fruit yield, weight, and firmness in the California population indirectly selected for increased fruit count and decreased total soluble solids (TSS), titratable acidity (TA), and anthocyanin concentration (ANC) (Figs. 5–6). Genetic gains from breeding for increased fruit firmness greatly improved shelf life (decreased perishability), which was critical for expanding the production of strawberries for export (Fig. 5; Table 2). GEBV means progressively increased for traits under direct selection (fruit yield, weight, and firmness) and progressively decreased for traits under indirect selection (TSS, TA, and ANC) from post-CP years onward (Fig. 5; Table 2). The trends uncovered by ss-BLUP analyses were consistent with the breeding history of the CA

population[2], production history of strawberry in the US (Fig. 1), additive genetic correlations among directly and indirectly selected traits (Fig. 6), and genetic gains or losses directly estimated from the phenotypic means of training population hybrids (Table 1).

We attributed the correlated responses to selection (TSS, TA, and ANC decreases) to negative additive genetic correlations caused by the pleiotropic effects of loci targeted by direct selection for fruit yield, weight, and firmness (Fig. 6). Despite TSS and TA decreases, our analyses show that TSS/TA (sugar-to-acid) ratios increased in post-CP years (Fig. 5; Table 2). The latter possibly increased perceived sweetness and partly counterbalanced the decrease in sweetness associated with the decrease in fruit sugar (TSS) concentrations from the 1960s onward (Figs. 5–6; Table 2). The increased yields of firm, less perishable fruit were, nevertheless, accompanied by decreased sweetness of strawberries produced on a large scale in California for year-round export to markets across North America.

Although population means for anthocyanin concentration steadily decreased after 1959 (the estimated CP year), the ANC ranges widened and individuals with the lowest and highest anthocyanin concentrations emerged in post-CP years (Fig. 5; Table 2). The trend observed for this trait exemplifies the trend observed for every trait, namely that Green Revolution breeding greatly increased phenotypic ranges among CA population individuals, in addition to increasing or decreasing population means over time (Fig. 5; Supplementary Figs. 2–3). The phenotypic range increases were attributed to transgressive segregation and genetic gains from selection (increased frequencies and accumulation of favorable alleles).

Green Revolution breeding improved the external physical appearance of the fruit in parallel with extending shelf life and increasing fruit yield, count, and weight in the California population (Fig. 7; Supplementary Fig. 4). The fruit shapes and sizes depicted in Fig. 7A–C for the wild parent and early hybrids were reproduced from the botanical line drawings of Duchesne[3], rescued and annotated by Staudt[46]. They highlight the variation in fruit morphology observed by Duchesne amongst the earliest, extinct spontaneous hybrids between the octoploid founders (*F. chiloensis* and *F. virginiana*)[46]. Surface defects, deformities, and asymmetrical shapes were common among those early hybrids and many heirloom and early Green Revolution cultivars (Fig. 7A–G)[1,46]. Such imperfections are uncommon among high-yielding modern cultivars from the California population, which are depicted in the terminal nodes of Fig. 7H, J. Our domestication

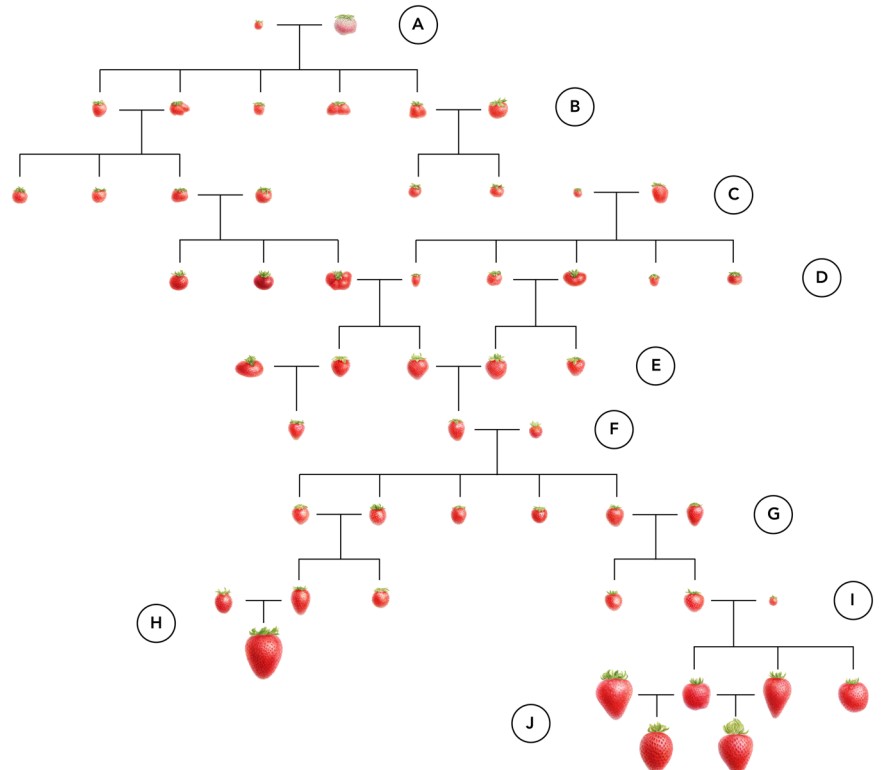

**Fig. 7 | Graphical abstract of strawberry domestication milestones (1715-present).** The fruit illustrations are watercolors developed by Sandra Doyle© from botanical illustrations of extinct ancestors and photographs of living specimens (https://www.sandra-doyle.co.uk/; commissioned by UC Davis and used with permission of the artist). Letters identify milestones in the domestication history of strawberry. Letters identify milestones in the domestication history of strawberry. **A** Spontaneous hybrids arise between *F. chiloensis* subsp. *chiloensis* and *F. virginiana* subsp. *virginiana* ecotypes under cultivation in western Europe (1715 and later). Fruit of the original *F. chiloensis* parent plant gifted by[104] to Antoine de Jussieu (Professor and Demonstrator of the Interior and Exterior of Plants at the King's Garden, Versailles, France) were described as "whitish red". **B** The earliest interspecific hybrids emerge and are exchanged and cultivated in the Garden of Versailles and other botanical gardens in western Europe (1715–1766). **C** Antoine Nicolas Duchesne discovers the interspecific hybrid origin of *F. × ananassa* and early cultivars begin emerging from artificial hybridization and selection among interspecific progeny (1766 and later). **D** Keen's Seedling, Downtown, and other iconic early cultivars emerge in western Europe, are widely disseminated and exchanged, and migrate to North America in the early 1800s and later. **E** Royal Sovereign, Nich Omher, and other iconic cultivars emerge in the late 1800s and early 1900s in North America and Europe. **F** Alfred Etter introgresses alleles from native *F. chiloensis* subsp. *pacifica* and *F. chiloensis* subsp. *lucida* ecotypes, pioneers strawberry breeding at the turn of the century in coastal California (1899–1927), and donates his genetic resources to the University of California in 1928. **G** Lassen, Fairfax, Shasta, and other early cultivars emerge from the California and other populations in North America (1924 onward). **H** Green Revolution short-day (photoperiod sensitive) cultivars emerge from the California population (1953 onward). **I** Royce S. Bringhurst initiates the development of photoperiod-insensitive *F. × ananassa* cultivars by introgressing alleles from an *F. virginiana* subsp. *glauca* ecotype native to Utah (1953–1980). **J** Green Revolution day-neutral (photoperiod-insensitive) cultivars emerge from the California population (1980 onward). Source data are provided as a Source Data file.

chronology highlights the introduction of genetic variation from North American ecotypes of *F. chiloensis* by Alfred F. Etter in the 1900s and *F. virginiana* by Royce S. Bringhurst in the 1950s (Fig. 7F, I) and the speed with which fruit weight and external physical appearance were improved by Green Revolution breeding (Supplementary Fig. 4)[20,21,31,47].

### Resistance to two widespread soil-borne pathogens broadly declined over the course of the Green Revolution

Verticillium wilt and Phytophthora crown rot are among the most widespread and devastating diseases of strawberry[48]. These diseases are excellent bellwethers of historical genetic gains from breeding because the genetic mechanisms underlying resistance are complex (quantitative) and both were initially reported in the early twentieth century and have caused plant death and yield losses in strawberry for more than a century[49–54]. Moreover, they were both virtually eliminated as problems in conventional strawberry production systems, along with other diseases caused by soil-borne pathogens, once methyl bromide fumigation was introduced in 1960 as the Green Revolution was getting underway[37,55,56].

We modeled how resistance to these soil-borne pathogens appears to have changed over the course of strawberry domestication by estimating ss-BLUP GEBVs for 923 training population hybrids previously phenotyped for resistance to Verticillium wilt and 434 training population hybrids previously phenotyped for resistance to Phytophthora crown rot (Fig. 5; Table 2)[52–54]. The hybrids in these training populations were genotyped using the 50K Axiom® SNP array and included genetically diverse samples of CA and non-CA population individuals[52–54]. The change-point years for disease resistance traits ranged from 1952 to 1957, the initial decade of the Green Revolution in California (Fig. 5; Table 2; Supplementary Figs. 2–3). We found that susceptibility to these diseases has increased since 1960, that strong resistance was more common in pre-CP than post-CP populations, and that the frequency of susceptible individuals has increased in post-CP populations (Fig. 5; Table 2).

The additive genetic correlations between yield and disease traits were antagonistic (the estimates were positive because disease susceptibility increases as disease score and AUDPS increase on the ordinal symptom rating scale used in our study; Fig. 6). The estimated CP years for resistance traits (1952–1957) coincided with the

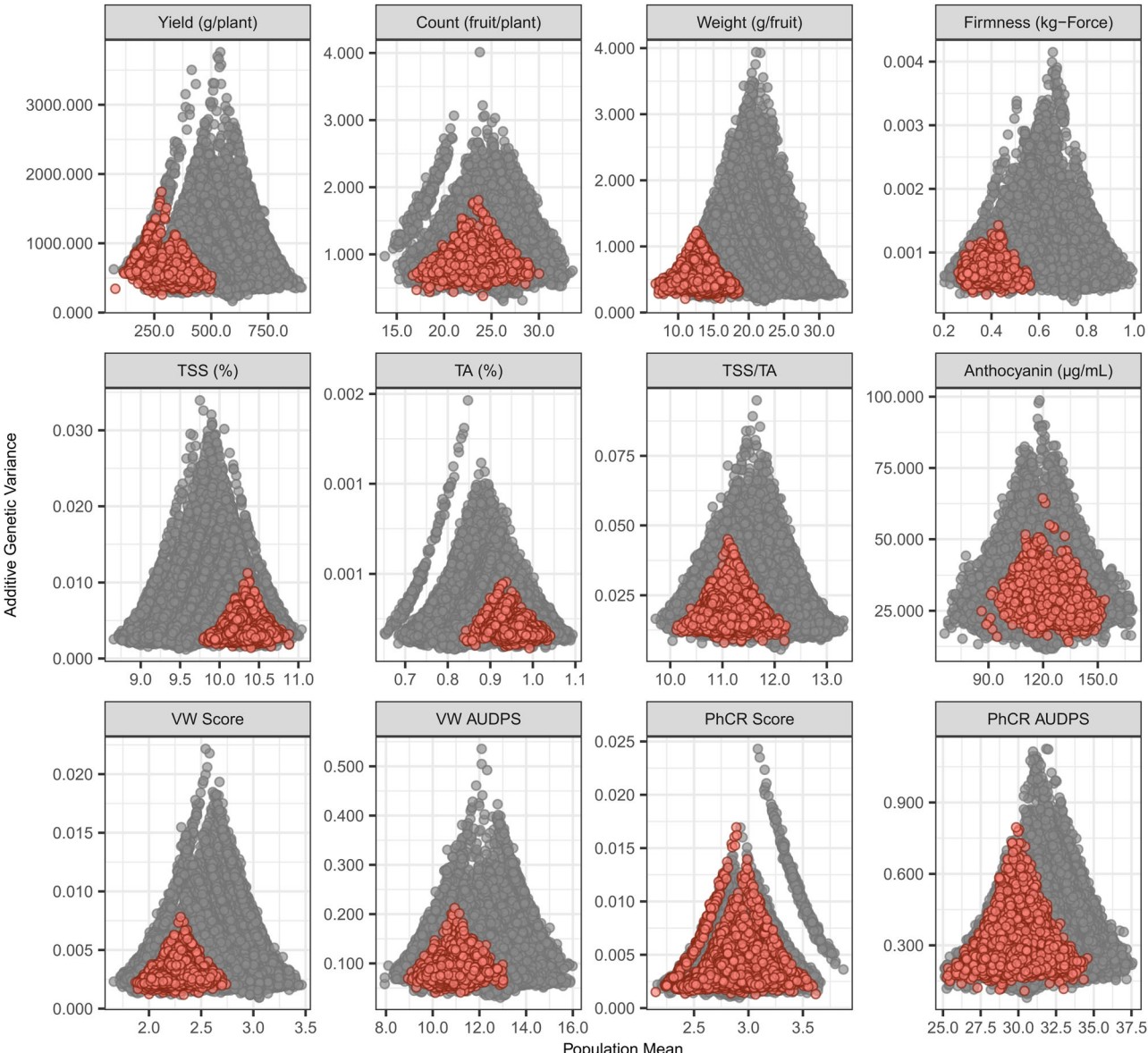

**Fig. 8 | Genomic prediction of additive genetic variance.** Genomic-predicted population means and additive genetic variances ($V_A$) were estimated for agriculturally important traits spanning the history of strawberry breeding (1775–2015). The statistics shown were estimated from 87,893 simulated segregating populations from crosses among parents originating before the approximate start of the Green Revolution (1775–1953; solid red circles) and 84,490 simulated segregating populations from crosses among parents originating after 1953 (solid gray circles).

Abbreviations for fruit quality traits shown in the middle row of plots are TSS = total soluble solids and TA = titratable acidity. Abbreviations for disease resistance traits shown in the lower row of plots are Verticillium wilt (VW) resistance score and area under the disease pressure stairs (AUDPS) and Phytophthora crown rot (PhCR) resistance score and AUDP. The ordinal resistance scores for both ranged from highly resistant (1) to highly susceptible (5). Source data are provided as a Source Data file.

introduction and widespread adoption of soil fumigation with methyl bromide[37,55,56]. Interestingly however, the most highly resistant individuals emerged in the earliest years of the Green Revolution before steadily declining over time (Fig. 5). Those individuals were selected in the decades immediately preceding the start of the Green Revolution in California. Even though GEBV ranges increased in post-CP years, GEBVs were strongly shrunk towards the within-year population means. While additive genetic variation for resistance increased in post-CP years, the shifts in disease score and AUDPS ranges were mostly towards greater susceptibility (Fig. 5). Notably, AUDPS ranges widened in both directions in post-CP years, with a trend towards greater AUDPS values as a whole (earlier onset of symptoms, faster progression of symptoms, or both) over time (Fig. 5; Table 2). As noted

earlier, hybrids with the smallest AUDPS values (most resistant) emerged in the early post-CP years.

### Green Revolution breeding increased additive genetic variation in the California population

One of the unanticipated findings from our study was that additive genetic variance ($\sigma_A^2$) increased for every trait from change-point years onward (Figs. 5 and 8; Supplementary Fig. 3). This was evident from the progressive increase in breeding value ranges (Fig. 5), triangle distributions of residuals from linear regressions of GEBVs on birth years (Supplementary Fig. 3), and genomic-estimated additive genetic variances (Fig. 8). We attributed the $\sigma_A^2$ increases to transgressive segregation, widening of phenotypic ranges associated with genetic gains

from selection, and increased homozygosity[5,20,31]. The $\sigma_A^2$ increases cannot be attributed to the introduction of novel alleles from non-California population individuals (migration), which ceased in the late 1980s when the California population experienced a breeding bottleneck, aside from the apparent introduction of novel alleles from one to a few individuals of unknown origin in the early 1990s[2,5].

The estimated change-point years from piecewise linear regressions of GEBV residuals on birth years pinpointed the origins of the $\sigma_A^2$ increases to the formative years of the UC Davis strawberry breeding program (Supplementary Fig. 3; Table 2). We identified three fairly distinct periods in the evolution of the California population. First were the formative years (1927–1950) where genetic variation was introduced from diverse founders, transgressive phenotypes began emerging, and early breeding breakthroughs were achieved (Fig. 5; Table 2). Second were the early Green Revolution years (1950–1990) where several agriculturally important traits were substantially improved and novel alleles were introduced. Third were the late Green Revolution years (1990-present) where steady genetic gains for fruit yield, weight, and firmness were achieved and the population became closed. The alleles present in the mid-twentieth century generations of the California population were the genetic foundation on which the Green Revolution was built (Fig. 5; Table 2) and on which several public and private sector strawberry breeding programs were either built or improved[2,7,9,10,20,21,57,58].

We used whole-genome regression methods[29,59,60] to predict $\sigma_A^2$ by simulating 87,893 segregating populations using training population hybrids as parents (Fig. 8). These analyses shed further light on the $\sigma_A^2$ increases associated with Green Revolution breeding. We observed triangle population mean × $\sigma_A^2$ distributions for every trait (Fig. 8), the classic pattern observed for traits where genetic gains from breeding have been substantial[29,59,60]. Crosses between post-CP parents were predicted to generate significantly more $\sigma_A^2$ than crosses between pre-CP parents (gray and coral points, respectively, in Fig. 8). Although predicted population means were strongly shifted upwards for traits under direct selection (fruit yield, weight, and firmness) and downwards for traits under indirect selection (TSS, TA, and ANC) in post-CP years, several post-CP crosses had predicted population means that exceeded the upper tails of the TSS, TA, and ANC distributions for pre-CP crosses (Fig. 8).

The $\sigma_A^2$ increases observed in the California population were presumably caused by transgressive segregation and the accumulation of favorable alleles, which widened phenotypic ranges[61,62]. Naturally, $\sigma_A^2$ estimates were greatest for crosses between post-CP parents from opposite tails of the phenotypic distributions (Fig. 8). The predicted population means estimated for these simulated segregating populations (x-axes in Fig. 8) further highlight the historical genetic gains and losses associated with Green Revolution breeding in strawberry (Table 1). Counterintuitively perhaps, our analyses show that the California population, which has experienced the greatest breeding-associated declines in nucleotide diversity and heterozygosity reported thus far[5], harbors more additive genetic variance than founder populations as a consequence of the accumulation of favorable alleles in coupling, unmasking, and selection against deleterious recessive alleles (a significant component of genetic diversity), and transgressive segregation (Figs. 5 and 8; Supplementary Fig. 3).

## Discussion

The little-known California strawberry Green Revolution documented here chronologically paralleled the wheat and rice Green Revolutions[16,22,23], transformed strawberry production in the US, created a highly dependable, more accessible, and less perishable supply of fresh strawberries in North America and beyond, and enabled dependable year-round strawberry production[20,21]. Strawberries were only one or two generations earlier seasonally available, substantially lower yielding, and more perishable (Figs. 1 and 4–5)[20,21,31]. As a consequence of increased demand, population growth, and genetic gains from Green Revolution breeding, strawberries have since become one of the most widely consumed fruits in the world (https://www.fao.org/faostat/) and an increasingly important source of vitamin C and other nutrients and functional food with "multiple health benefits beyond basic nutrition"[63,64].

Genetic gains from breeding for increased yield, in concert with chemically intensive production practices, drove the production increases needed to meet the year-round demand for fresh strawberries in North America (Fig. 1; https://www.fao.org/faostat/), principally through expansion in California (https://www.ers.usda.gov/). The timing and trends of the Green Revolution yield increases reported here for strawberry (Fig. 5) are uncannily similar to the well-known and widely reported yield increases from single-cross breeding in corn[13,14].

Strawberry production has increased at nearly identical linear rates in the US and Europe since 1961, the approximate start of the Green Revolution in California (Fig. 1; https://www.fao.org/faostat/). Since 1961, the annual production increase has been slightly greater in Europe (26,782 tonne/year; $R^2 = 0.915$) than the US (22,215 tonne/year; $R^2 = 0.918$) with a cumulative increase of 1.6B kg in Europe and 1.3B kg in the US. Strikingly, the production increase in Europe was achieved by a 398% increase in the area harvested and 1% decrease in yield, whereas the production increase in the US was achieved by a 2,755% increase in yield and 17% decrease in the area harvested (Fig. 1). If the UN-FAO production statistics are accurate, the US and Europe solved the problem of meeting increased demand in markedly different ways, both of which appear to be unsustainable (Fig. 1). The soil fumigation practices that were necessary to meet demand and achieve the genetic gains for intrinsic yield reported here are unsustainable[55,56,65]. The continual expansion of land under cultivation, which has been necessary to meet increased demand in many parts of the world, is equally unsustainable and increasingly constrained by urbanization and production on increasingly marginal lands, and limits crop rotation schemes that are vital for maintaining soil health and crop productivity, particularly in prime coastal areas where strawberries are produced on a large scale[11,12,66,67].

We restricted our survey of strawberry yield and production trends to the US and Europe because of their long breeding and production histories, pivotal roles in strawberry domestication, and availability of production statistics from the inception of UN-FAO record keeping for strawberry[1,20,21,68–70]. UN-FAO statistics show that strawberry yields have not increased in Europe since 1961, were lower in the US than Europe in the 1960s, identical in the US and Europe by 1970 before diverging over the course of the California Green Revolution, and were 53,440 kg/ha greater in the US than Europe by 2021 (Fig. 1; https://www.fao.org/faostat/). The lack of a yield increase in Europe over the last 60 years seems implausible. Despite the uncertain accuracy of the reported yield differences between the US and Europe (Fig. 1), they have undoubtedly been caused by a multitude of economic, societal, and scientific factors, most importantly differences in cultivar preferences, production practices, pesticide, and inorganic fertilizer inputs, shelf-life requirements, farm-to-market transportation distances, consumer preferences, and marketing and distribution practices[69,70].

Senger et al.[70] stressed that breeding priorities in Europe have traditionally included "higher yields and yield stability, lower production costs, and better product quality" and that, "in the past, breeders focused mainly on yield and production costs, but product quality is now a high priority." This suggests that differences in breeding priorities are insufficient to account for the apparent absence of yield increases in Europe (Fig. 1), and implies that cultivars with improved yields have not been widely adopted, that improved production practices have not increased yields, that biotic and abiotic stresses have masked genetic gains, or a combination of these factors and others.

Our conclusions about historical genetic gains were necessarily limited to what we could learn from UN-FAO production statistics (Fig. 1) and direct estimates of phenotypic differences among elite × elite and elite × exotics hybrids observed in common garden experiments (Table 1; Fig. 4). The elite parents for our study were exclusively drawn from California population individuals preserved in the UC Davis Strawberry Germplasm Collection as January 1, 2015, with ancestry tracing back through the century-long history of breeding at UC Berkeley (1924–1945) and UC Davis (1952-present)[2,5,10,20,21,71]. The parents we selected for hybrid development originated subsequent to the circa 1987 California population bottleneck we previously identified[2,5]. While the impact of the California strawberry Green Revolution documented here has been substantial and does not appear to have been widely replicated, many other public and private breeding programs have contributed to the commercial success and global expansion of strawberry production inside and outside of California[58,68–70]. We had no basis, however, for determining their contributions (genetic gains), which have not been documented by their originators either.

Genetic gains from artificial selection, founder effects, and population bottlenecks predictably decrease allelic variation within domesticated plant and animal populations[30,72,73]. While such decreases are a natural consequence of breeding success (genetic gains), and well-founded concerns have been raised about the negative effects of breeding-caused genetic erosion on sustainability and global food security[30,72,73], those decreases are primarily caused by the accumulation of favorable and elimination of unfavorable alleles, apart from the unwanted fixation of unfavorable alleles in selective sweeps and persistence of unfavorable alleles in genomic regions recalcitrant to recombination[5,74–76]. While discourses on genetic erosion and the preservation of genetic diversity appropriately focus on the negative impacts of breeding on genetic diversity[77], they often downplay or overlook the profound increases in additive genetic variation created by transgressive segregation and the emergence of novel phenotypes[18,78–80], as exemplified by the extreme phenotypes and $\sigma_A^2$ increases reported here (Figs. 4–5 and 8; Supplementary Fig. 3). Those increases, in hindsight, were completely predictable and appear to have been largely but not exclusively driven by transgressive segregation[81,82]. The principal factors that increase additive genetic variation within domesticated plant populations, including the California population, include: the introgression of novel favorable alleles from elite and exotic sources (migration); identification, characterization, and utilization of large-effect mutations that improve agriculturally important traits (e.g., *PF* in strawberry and *SELF-PRUNING* and many others in tomato); the identification and selection of transgressive segregates (hybridization, recombination, and selection); increased inbreeding without the loss of alleles within or between populations; and between-population partitioning of allelic variation[78–82].

Our analyses of historical genetic gains for resistance to Verticillium wilt and Phytophthora crown rot in strawberry suggest that breeding for increased resistance to these diseases was either ineffective, neglected, or relaxed following the introduction of soil fumigation with methyl bromide in 1960[51–54]. The genetic foundation for the strawberry Green Revolution built by Royce S. Bringhurst at the University of California, Davis in the 1950s intersected with improvements in bare-root plant propagation and methyl bromide-suppression of diseases caused by soil-borne pathogens, a perfect storm for the Green Revolution where breeders could select for yield and other agriculturally important traits critical for large-scale production free of the burden and challenge of simultaneously selecting for resistance to multiple soil-borne pathogens[7,20,21,36,37,55,56,58]. That freedom, however, appears to have translated into more than a half-century of relaxed selection for increased resistance to diseases caused by soil-borne pathogens[37,51–55,83,84]. Our analyses suggest that favorable alleles found in Verticillium wilt and Phytophthora crown rot-resistant individuals discovered in the 1960s and earlier appear to have been under-utilized but definitely not lost. Those alleles are preserved in cultivars and other individuals maintained in gene banks that are accessible for delivering more sustainable strawberry production solutions[52,53,69,70,83,84].

The California strawberry Green Revolution could not have been achieved without the protection provided by methyl bromide, a much maligned and banned molecule that historically sustained production in more than a hundred agriculturally important plants afflicted by diseases caused by soil-borne pathogens and other pests[55,65,85]. Our look back on the breeding history that underpinned the strawberry Green Revolution suggests that the most important outcome of the breeding that drove productivity upward in California was undoubtedly the development of elite genetic resources that have been extensively used in breeding worldwide and supply a solid foundation for addressing breeding challenges going forward[5,7,20,21,31,57,58]. The most pressing challenge is the development of cultivars resistant to a broad spectrum of diseases that have either recently emerged or for which historical genetic gains have been uneven or inadequate[35,52–54,83,84,86–90]. The path forward will almost certainly entail the production of strawberries and many other fruits and vegetables in a world without soil fumigants[55,56,65]. As the insightful and balanced analysis by Charles C. Mann suggests[23], society as a whole and communities with diametrically opposed world views (at least historically), face the daunting challenge of simultaneously protecting the planet and feeding an ever-growing population.

## Methods

### Parent selection, plant material, and mating design for the ss-BLUP genomic prediction study

The parents of training population hybrids developed for our genomic prediction study spanned the domestication range from wild ancestors to modern cultivars. Seventy-five full-sib families were developed by manual emasculation and pollination using an incomplete factorial (14 × 16) mating design among 27 elite and three exotic parents. We randomly selected 560 hybrid offspring from a field-grown population of 2800 hybrid offspring from these families to construct a training population for the ss-BLUP genomic prediction study. Twenty-seven 'elite' parents (asexually propagated individuals) were selected for the development of 405 elite × elite training population hybrids. Of the 27 elite parents, 13 were classified as photoperiod sensitive (short-day flowering) and 14 were classified as photoperiod-insensitive (day-neutral flowering). Their photoperiod sensitivities were verified through three years of field observation. They originated between 1988 and 2011 in the University of California, Davis (UC Davis) Strawberry Breeding Program and included several historically and commercially important cultivars publicly released by UC Davis.

The 27 elite parents are themselves hybrids with varying degrees of genetic relatedness and heterozygosity. They were genotyped with a 50K Axiom® SNP array and were known from DNA marker-informed estimates of genetic relationship, genealogical analyses, population structure analyses, and genome-wide analyses of nucleotide diversity to belong to a single genetically distinct population, identified herein as the California population[2,5,39]. The genealogy of the California population has been fully reconstructed and integrated with the genealogies of cultivars and other individuals with public pedigree records[2]. The individuals in this population share multiple common ancestors, as do many other *F. × ananassa* populations around the globe, and have become progressively more inbred over time as a consequence of consanguineous matings, artificial selection, population bottlenecks, and selective sweeps[2,5]. We used insights gained from this knowledge to select parents that were predicted to sample genetic variation as broadly as possible among photoperiod-sensitive (short-day flowering) and photoperiod-insensitive (day-neutral flowering) individuals in the California population.

Three exotic individuals (Puget Reliance, Oso Flaco, and Del Norte) were selected as parents for the development of 132 elite × exotic and eight exotic × exotic hybrids for the training population. Puget Reliance (PI664321) is a short-day *F. × ananassa* cultivar released in 1994 for fresh and processing production in the Pacific Northwest (USPP9310P). Oso Flaco (55C023P001) is a short-day hybrid developed in 1955 between the heirloom *F. × ananassa* cultivar Lassen (36C003P001 developed in 1936) and an extinct *F. chiloensis* subsp. *lucida* ecotype originally collected from the Guadalupe-Nipomo Dunes near Oso Flaco Lake, Guadalupe, California by Royce S. Bringhurst. Del Norte (PI551753) is a short-day *F. chiloensis* subsp. *lucida* ecotype originally collected from coastal Washington. These individuals provided a progressively more exotic series of parents, from the cultivar Puget Reliance to the heirloom cultivar × wild ecotype hybrid Oso Flaco to the wild ecotype Del Norte. We developed 54 elite × Del Norte, 19 elite × Oso Flaco, 59 elite × Puget Reliance, and eight exotic × exotic hybrids.

Oso Flaco and other individuals, identified by 10-digit UC Davis identification (UC Davis ID) numbers in a year-family-individual format (e.g., 55C023P001), are preserved in the UC Davis Strawberry Germplasm Collection. Puget Reliance and Del Norte, the individuals identified by plant introduction (PI) numbers, were originally acquired as bare-root plants from the United States Department of Agriculture (USDA) National Plant Germplasm System (NPGS) National Clonal Germplasm Repository, Corvallis, Oregon, USA (https://www.ars.usda.gov/pacific-west-area/corvallis-or/national-clonal-germplasm-repository/). They were subsequently maintained and increased by asexual propagation at UC Davis.

Of the 574 parents and hybrids in the training population, 48% originated from elite day-neutral × elite day-neutral, 25% originated from elite day-neutral × elite short-day, and 22% originated from elite day-neutral × exotic short-day crosses. We used a SNP marker on the 50K Axiom® array (AX-184937335) found to be in linkage disequilibrium with the *PERPETUAL FLOWERING* (*PF*) locus to predict *PF* locus genotypes[31–33]. The day-neutral parents we selected were determined to be heterozygous for the dominant allele (*PF*), whereas the short-day parents were determined to be homozygous for the recessive allele (*pf*). Hence, elite day-neutral × elite day-neutral families segregated 3 *PF_* : 1 *pfpf* and elite day-neutral × elite short-day and elite day-neutral × exotic short-day families segregated 1 *PFpf* : 1 *pfpf*. Using genotypes predicted by the *PF*-associated SNP marker, 205 were predicted to be *pfpf* (short-day), whereas 337 were predicted to be *Pf_*, of which 231 where heterozygous (*PFpf*) and 106 were homozygous (*PFPF*).

### Asexual propagation of parents and hybrids

Clones (bare-root plants) of the parents and hybrids analyzed in our studies were asexually propagated from stolons in a low-elevation nursery from foundation genetic stocks preserved in the UC Davis Strawberry Germplasm Collection at Wolfskill Experiment Orchard, Winters, CA (38.53° N, 121.97° W; 41 m). Clones were annually harvested from the field in December or January, stored at 0° C for approximately three months, and transplanted in April to a commercial high-elevation nursery (Cedar Point Nursery, Dorris, CA; 41.97° N, 121.92° W; 1,293 m). Clones were produced using standard industry production practices in collaboration with Cedar Point Nursery. The harvest was carefully timed in each year to ensure plants had the appropriate number of chilling hours needed for optimal production in short-day or day-neutral flowering environments[36]. Clones produced for yield trials in Oxnard (a typical short-day production environment in CA) were harvested in September, temporarily stored in the dark at 4° C, and transplanted to fruit production fields within one-week of harvest. Clones produced for yield trials in Santa Maria and Prunedale (typical day-neutral production environments in CA) were harvested in October, temporarily stored in the dark at 4° C, and transplanted to

fruit production fields within one-week of harvest. Finally, clones of training population parents and hybrids produced for the genomic prediction study were harvested in October, temporarily stored in the dark at 4° C, and transplanted to fruit production fields in Salinas, CA (a typical day-neutral environment) within one-week of harvest.

### Pedigree database

We developed a genealogical database with pedigree records (parent-offspring trios) for 15,649 *F. × ananassa* individuals. This database incorporated 8,851 pedigree records for *F. × ananassa* individuals from a previously described genealogical database[2]. The expanded database includes pedigree records for every cultivar (asexually propagated hybrid individual) developed at the University of California, Davis (UC Davis) since inception, 560 training population hybrids developed for our ss-BLUP genomic prediction study, every non-UC Davis cultivar with readily available public pedigree records, and ancestors of these individuals tracing back to 1775. This genealogy documents the history of cultivated strawberry breeding in the US and Europe, the epicenters of early breeding in the eighteenth and nineteenth centuries[1,2].

### SNP genotyping

To obtain DNA variant-informed estimates of genetic relationships for our ss-BLUP analyses, 3821 individuals with pedigree records were genotyped with a 50K Axiom® SNP array[39]. The genotyped individuals included the ss-BLUP training population parents and hybrids described above. SNP data from several previous studies[2,5,39,52,53,91] were collated with SNP data for individuals genotyped for the present study. Young leaf tissue samples were collected into 1.1 mL tubes, freeze-dried in a Benchtop Pro (VirTis SP Scientific, Stone Bridge, NY), and ground using stainless steel beads in a Mini 1600 (SPEX Sample Prep, Metuchen, NJ). Genomic DNA (gDNA) was extracted from powdered leaf samples using the E-Z 96® Plant DNA Kit (Omega BioTek, Norcross, GA, USA) according to manufacturer's instructions. To enhance the quality of the DNA and reduce polysaccharide carry-through, the protocol was modified with a Proteinase K treatment, a separate RNase treatment, an additional spin, and heated incubation steps during elution. DNA quantification was performed using Quantiflor dye (Promega, Madison, WI) on a Synergy HTX (Biotek, Winooski, VT).

SNP genotyping with the 50K Axiom® SNP Array[39] was performed by Affymetrix (Santa Clara, CA) on a GeneTitanTM HT Microarray System using gDNA samples that passed quality and quantity control standards. SNP genotypes were automatically called with the Axiom® Analysis Suite software (v1.1.1.66, Affymetrix, Santa Clara, CA). Samples with call rates below 90% were dropped. Quality metrics output by the Axiom® Analysis Suite and custom R scripts were utilized to exclude SNPs with minor allele frequency (MAF) greater than 0.05% and less than 10.0% missing data were included with 48,177 high-quality bi-allelic and subgenome-specific SNPs.

### Genetic gain field study design and phenotyping

To estimate genetic gains, 573 hybrids were phenotyped over two years in Salinas, CA for yield (g/plant), count (fruit/plant), weight (g/fruit), firmness (kg-Force), TSS (%), TA (%), TSS/TA, and ANC (µg/mL). The parents and hybrids were grown on a commercial farm in Salinas, CA (36.62° N, −121.54° W, 46 m) over the 2017–18 and 2018–19 growing seasons. The field was pre-plant flat-fumigated with a chloropicrin-based fumigant (Pic-Clor 60; 560 kg/ha) and sealed with a totally impermeable film tarp for one-week post-fumigation. Once the tarps were removed, fields were prepared for planting by building 30.0 cm high × 75.0 cm wide raised beds with 120.0 cm of spacing between beds center-to-center. Drip irrigation lines were installed before covering the beds with black plastic mulch. Hybrids were grown in four-plant plots arranged in a randomized complete blocks experiment design with four complete blocks (replications). Low-elevation produced bare-root plants (6900/year) were transplanted in early

November of each year through small planting holes spaced 30.0 cm apart within and between rows (equivalent to a density of 45,000 plants/hectare). These experiments covered approximately 0.3 ha/year. They were irrigated, fertilized, and sprayed with pesticides as needed and managed according to the production practices of Garcia Farms at Spence Ranch, Salinas, CA.

Fruit were harvested from each plot once per week for 11 or 13 successive weeks from the beginning of April to the end of June in the 2017–18 and 2018–19 seasons, respectively. The harvest period was purposefully chosen to avoid the confounding effects of photoperiod sensitivity. The last harvest in both years was one-week past the summer solstice (June 21) when daylengths reached their maximum (14.7 h) and short-day hybrids typically cease flowering at the latitudes in our test environment. The short-day hybrids in our experiment produced fruit through the summer solstice from flowers produced approximately four weeks earlier. Fruit yield (g/plant), count (number of fruit/plant), and weight (g/fruit) were recorded at each harvest (123,012 phenotypic observations were collected and analyzed for these traits).

We sampled three fruit/plant from the sixth and twelfth harvest each year for fruit quality trait phenotyping. Fruit firmness (maximum resistance kg-force) was assessed on whole fruit using a TA.XT plus Texture Analyzer with a TA-53 3 mm puncture probe (Stable Micro Systems Ltd., Goldaming, United Kingdoms). Fruit samples were frozen at −20° C in Whirl-Pak®Homogenizer Blender Filter Bags (Nasco, Fort Atkinson, WI, USA) for quantifying titratable acidity (TA; %), total soluble solid content (TSS; ° BRIX), and total anthocyanin concentration (ANC; μg/mL). TA percentages were quantified with a Metrohm Robotic Titrosampler System from 1–5 mL of the defrosted homogenized fruit juice (Metrohm AG, Herisau, Switzerland). SSC was measured from approximately 200 μL of juice on an RX-5000α-Bev Refractometer (ATAGO Co. Ltd., Tokyo, Japan). Total anthocyanin concentration was measured from a 25 μL sample of juice in 200 μL 1% HCl in methanol by reading absorption at a wavelength of 520 nm on a Synergy HTX plate reader equipped with Gen5 software (Molecular Devices, San Jose, California, USA). A standard curve ($y = sx + i$) was calculated for quantifying AC using a dilution series of pelargonidin (Sigma–Aldrich, St. Louis, MI, USA) from zero to 300 μg/mL in 50 μg/mL increments, where $y$ were absorption readings for the pelargonidin dilution series, $s$ was the slope, $x$ was the concentration of pelargonidin in the dilution series, and $i$ was the intercept. ANC was estimated by $(A - i)/s$, where $A$ was the absorption reading. In this study, 25,630 phenotypic observations were recorded and analyzed for TSS, TA, ANC, and firmness.

Lastly, we analyzed 52,900 phenotypic observations for Verticillium wilt and Phytophthora crown rot resistance among 923 and 434 training population hybrids, respectively, observed in previous studies[52,53]. GEBVs were estimated by ss-BLUP for disease symptom score and area under the disease pressure stairs (AUDPS) score for both diseases[92]. AUDPS scores were calculated using the R package *agricolae* (https://cran.r-project.org/web/packages/agricolae/index.html) from a time-series of five to eight disease symptom scores recorded over several weeks. These data are publicly available[52,53].

### Single-step pedigree-genomic best linear unbiased prediction of breeding values

Estimated marginal means (EMMs) for training population parents and hybrids were estimated for each trait within and between years using the R package *emmeans*[93]. Linear mixed models (LMMs) were constructed and analyzed using the R package *lme4*[94], where hybrid was the only fixed effect and block (B), year (Y), hybrid × year (H × Y), and the residuals were random effects for the across year analysis. Variance components for random effects were estimated using REML with the R package *lme4*[94]. To estimate broad-sense heritability on a clone-mean basis ($\hat{H}^2 = \hat{\sigma}_G^2/\hat{\sigma}_P^2$), LMM analyses were repeated with hybrids as random effects, where $\hat{\sigma}_G^2$ is the among hybrid variance, $\hat{\sigma}_P^2 = \hat{\sigma}_G^2 + \hat{\sigma}_{G \times Y}^2/y + \hat{\sigma}_E^2/ry$ is the phenotypic variance on a clone-mean

basis, $\hat{\sigma}_{G \times Y}^2$ is the hybrid × year variance, $\hat{\sigma}_E^2$ is the residual variance, $y$ is the number of years, and $r$ is the harmonic mean number of replications. The among hybrid variance estimated from clones has the same expected causal genetic variances as identical twins, specifically 100% of the additive and non-additive genetic variances among clones or twins[62].

Genetic gains or losses ($\Delta G = EMM_1 - EMM_2$) were estimated for each trait using linear contrasts between EMMs for pairs of hybrids with the most extreme phenotypes (statistics were estimated using *emmeans*). First, we tested the null hypothesis of no difference between the minimum and maximum EMMs among 405 elite × elite hybrids ($H_0$: $EMM_1 - EMM_2 = 0$), where $EMM_1$ was the maximum and $EMM_2$ was the minimum EMM for each trait. Second, we tested the null hypothesis of no difference between the elite × elite hybrid with the maximum EMM and elite × wild hybrid with the minimum EMM ($H_0$: $EMM_1 - EMM_2 = 0$) for each trait. The elite × elite and elite × wild hybrids in these linear contrasts differed. $t$-statistics and the probability of a greater $t$-statistic by chance were estimated for each contrast. Genetic change (gain or loss) percentages were estimated by $(EMM_1 - EMM_2)/EMM_2 \times 100$. The *emmeans* R package[93] was used to calculate EMMs, contrasts, and $p$-values adjusting for two tests per trait.

Genomic-estimated breeding values (GEBVs) were estimated for 15,649 individuals for each trait using single-step pedigree-genomic best linear unbiased prediction (ss-BLUP)[24,25]. This was accomplished by building a unified additive genetic relationship matrix ($H$) that integrated the numerator relationship matrix ($A$) for 15,649 individuals with pedigree records (extinct and living) and the genomic relationship matrix ($G$) for 3821 individuals genotyped with the 50K Axiom® SNP array[38,95–97]. The genealogical database included every cultivar and several hundred other hybrids with public pedigree records and their ancestors tracing back to 1775, the earliest date for which pedigree records exist in strawberry[1,2]. The pedigree relationship matrix (A), a Bayesian prior for the genomic relationship matrix (G)[25], was estimated for 15,649 individuals in the genealogical database using the *AGHmatrix::Amatrix()* function in the R package *AGHmatrix* (https://cran.r-project.org/web/packages/AGHmatrix/index.html)[95,98]. The genomic relationship matrix (G) was estimated for the 3821 genotyped individuals using the *AGHmatrix::Gmatrix()*. The unified additive genetic relationship matrix ($H$)[97,99] was estimated for 15,649 individuals using the *AGHmatrix:Hmatrix()*[98]. The A, G, and H matrices used in the SS-BLUP analyses have been provided in Source Data file.

We used the function *mmer()* in the R package *sommer* to estimate GEBVs for genotyped and non-genotyped individuals from the H matrix (https://cran.r-project.org/web/packages/sommer/index.html)[99–101]. The linear mixed model for this analysis was

$$\mathbf{y} = \mathbf{1}\mu + \mathbf{Z}\mathbf{g}_H + \epsilon \tag{1}$$

where $\mathbf{y}$ are the EMMs for training population hybrids, $\mu$ is the population mean, $\mathbf{Z}$ is the incidence matrix, $\mathbf{g}_H$ are GEBVs for individuals in the unified additive genetic relationship matrix ($\mathbf{H}$), $\mathbf{g}_H \sim N(0, \mathbf{H}\sigma_{\mathbf{g}_H}^2)$, and $\epsilon$ are the residuals with $\epsilon \sim N(0, \mathbf{I}\sigma_\epsilon^2)$. Additive genetic correlations were estimated using the Pearson correlation coefficient ($\hat{r}_G$) among GEBVs for all possible pairs of traits. GEBV accuracy was estimated for each hybrid in the ss-BLUP analysis by $1 - \sqrt{PEV_i/\hat{\sigma}_A^2}$, where $PEV_i$ is the prediction error variance of the $i$-th hybrid and $\hat{\sigma}_A^2$ is the genomic additive genetic variance estimated from (1).

### Change-point analyses of genomic-estimated breeding values over time

The birth years were known for 6419 and unknown for 9198 individuals used in the ss-BLUP analysis. The birth year range was 1775 to 2015. We used GEBVs for individuals with known birth years as dependent

variables and birth year as the independent variable in time-series change-point analyses to estimate change-point (CP) years[40]. We estimated the slope before and after each change-point year. These analyses were performed using the R package *mcp*[102]. The predicted GEBV values from those linear regressions estimated the population mean for each year. The predicted GEBVs ($\hat{y}$) for 1775, the CP year, and 2015 were used to estimate changes in population means (genetic gains) between 1775 and the CP year and between the CP year and 2015, e.g., the genetic gain from the CP year to 2015 was estimated by $(\hat{y}_{CP} - \hat{y}_{2015})/\hat{y}_{CP} \times 100$. Observed and predicted GEBVs from piecewise linear regressions were plotted using the R package *ggplot2*[103]. The change in additive genetic variance for each trait was estimated by the residuals ($y - \hat{y}$) from these analyses. The residuals were transformed to obtain positive values by taking the square root of the squared residuals, $\sqrt{(y - \hat{y})^2}$. The transformed GEBV residuals were regressed on birth year using the time-series linear regression approach described above to search for and estimate change-point years. Hence, a positive slope from linear regression of the transformed GEBV residuals on birth year indicated an increase in the additive genetic variance, whereas a negative slope indicated a decrease in the additive genetic variance.

### Genomic prediction of additive genetic variance in simulated segregating populations

We estimated genomic-predicted additive genetic variances ($V_A$) for simulated segregating populations (200 individuals/population) arising from crosses among prospective parents in the study population using the R package *PopVar*[29]. VA estimates were produced for 3403 crosses among parents born before 1953 and 84,490 crosses among parents born after 1953. This year was chosen as a logical break point for comparing $V_A$ estimates among prospective parents originating before and after the start of the Green Revolution. We used 50K Axiom® array SNP genotypes and a reference genetic map developed for the cultivar Camarosa[39] as input in the R package *PopVar* to estimate population means ($\hat{\mu}$) and genetic variances ($\hat{V}_A$) for each simulated population.

### Modern hybrid field study design and phenotyping

To estimate full-season yields of modern Green Revolution hybrids grown in coastal environments where strawberries are commercially produced on a large scale in California, 116 short-day hybrids were tested and phenotyped in a typical short-day production environment (Oxnard, CA), and 72-day-neutral hybrids were tested and phenotyped in typical day-neutral production environments (Santa Maria and Prunedale, CA). Thirty-one short-day and 33-day-neutral cultivars were tested over the 2015–16 and 2016–17 growing seasons, whereas 85 short-day and 39-day-neutral hybrids were tested in the 2015–16 growing season only. The California population hybrids in these experiments included several commercially important cultivars originating between 1997 and 2012, e.g., Albion (US Patent and Trademark Office Plant Patent USPP16228P3), Monterey (USPP19767P2), Fronteras (US20150230374P1), UCD Royal Royce (USPP16/501,374), UCD Moxie (USPP16/501,376), UCD Valiant (USPP16/501,375), UCD Victor (USPP16/501,372), and others.

The individual hybrids (entries) in this study were grown in one to four 20-plant plots in augmented randomized complete block experiment designs in 2015–16 or two to four 20-plant plots in augmented randomized block experiment designs in 2016–17 using the management practices, bed configurations, plastic mulches, planting densities, planting dates, irrigation, fertilization, and pesticide application decisions and schedules, and harvest decisions and schedules applied by our cooperators on their commercial farms in Oxnard, Santa Maria, and Prunedale, CA. Fruit was harvested once or twice per week over the growing season on each farm. We recorded the number and yield of marketable fruit from each plot at each harvest. Fruit

weight (g/fruit) was calculated by dividing fruit yield (g/plot/harvest) by the number of fruit/plot/harvest. The number of harvests/environment ranged from 46 to 52. Approximately 45,066 phenotypic observations were collected and analyzed for marketable fruit yield and count across harvests and environments.

### Statistical analyses of the modern hybrid field study

EMMs for short-day and day-neutral cultivars tested in the modern hybrid field study were estimated using the R package *emmeans*[93]. Within and across environment LMMs were analyzed using the R package *lme4*[94], where hybrid was the only fixed effect and block (B), Year (Y), location (L), year × location (Y × L), hybrid × year (H × Y), hybrid × location (H × L), and hybrid × year × location (H × Y × L), and residual were random effects in the across year analysis. Variance components for random effects were estimated using REML with the R package *lme4*[94].

EMMs were estimated for hybrids within and across growing seasons (2015–16 and 2016–17) using *emmeans*. We used *t*-tests to estimate differences between the EMMs for specific hybrids or combinations of the 31 short-day and 33-day-neutral hybrids tested in both growing seasons. First, we tested the null hypothesis of no difference between the minimum and maximum EMMs for short-day or day-neutral hybrids (H₀: $\text{EMM}_1 - \text{EMM}_2 = 0$), where $\text{EMM}_1$ is the maximum and $\text{EMM}_2$ is the minimum. Second, we tested the null hypothesis of no difference between short-day and day-neutral hybrid EMMs (H₀: $\text{EMM}_1 - \text{EMM}_2 = 0$), where $\text{EMM}_1$ is the EMM among 31 short-day and $\text{EMM}_2$ is the mean among 33-day-neutral hybrids. Third, we tested the hypothesis of no difference between EMMs for short-day and day-neutral hybrids with the highest yields from each group. Finally, we tested the null hypothesis of no difference in the yield/harvest between the 31 short-day and 33-day-neutral hybrids tested in both growing seasons, where the yield/harvest was estimated by dividing the cumulative marketable fruit yield by the number of harvests.

To estimate broad-sense heritability on a clone-mean basis:

$$\hat{H}^2 = \hat{\sigma}_G^2 / \hat{\sigma}_P^2, \tag{2}$$

LMM analyses were repeated with hybrids as random effects, where $\hat{\sigma}_G^2$ is the among hybrid variance,

$$\hat{\sigma}_P^2 = \hat{\sigma}_G^2 + \hat{\sigma}_{G\times Y}^2/y + \hat{\sigma}_{G\times L}^2/l + \hat{\sigma}_{G\times Y\times L}^2/ly + \hat{\sigma}_E^2/rly \tag{3}$$

is the phenotypic variance on a clone-mean basis, $\hat{\sigma}_{G\times Y}^2$ is the hybrid × year variance, $\hat{\sigma}_{G\times L}^2$ is the hybrid × location variance, $\hat{\sigma}_{G\times Y\times L}^2$ is the hybrid × year × location variance, $\hat{\sigma}_E^2$ is the residual variance, $y$ is the number of years, $l$ is the number of locations, and $r$ is the harmonic mean number of replications. The genetic variance among hybrids estimated from clones is equivalent to the genetic variance among identical twins, which captures 100% of the additive and non-additive genetic variances[62].

### Statistical analyses of yield and production trends in the US and Europe (1961–2021)

We downloaded raw data from the United Nations Food and Agricultural Organization (UN-FAO) database for strawberry production (kg), area harvested (ha), and yield (kg/ha) in the United States and Europe from 1961 to 2021 (https://www.fao.org/faostat/en/#data). UN-FAO began compiling data for strawberry in 1961. The raw data for Europe were flagged as "official figures", "unofficial figures", or "imputed values". The raw data for the US were exclusively flagged as "official figures". We limited our analyses to official figures only. We calculated production and area harvested data for Europe by summing "official figures" data for 43 countries flagged under the "Europe" category in the UN-FAO database: Albania, Austria, Belgium, Belgium-

Luxembourg, Bulgaria, Croatia, Cyprus, Czechia, Czechoslovakia, Denmark, Estonia, Finland, France, Georgia, Germany, Greece, Hungary, Ireland, Italy, Kazakhstan, Latvia, Lithuania, Luxembourg, Malta, Netherlands, North Macedonia, Norway, Poland, Portugal, Republic of Moldova, Romania, Serbia, Serbia and Montenegro, Slovakia, Slovenia, Spain, Sweden, Switzerland, Türkiye, Ukraine, and the United Kingdom of Great Britain and Northern Ireland. Yield (kg/ha) data were calculated for Europe by dividing production (kg) by area harvested (ha) data for individual years.

To model production, area harvested, and yield changes over time, we regressed these variables on the year of production from 1961 to 2021 using the *stats::lm()* function in R. Linear regression equations were estimated by $y = \beta_0 + \beta_1 x$, where $y$ is the dependent variable, $\beta_0$ is the intercept, $\beta_1$ is the regression coefficient, and $x$ is the year (independent variable). The predicted values for each variable ($\hat{y}$) were calculated for each year between 1961 and 2021. We estimated the percentage change in each variable over the 60-year study period using $(\hat{y}_{2021} - \hat{y}_{1961})/\hat{y}_{1961} \times 100$, where $\hat{y}_{1961}$ is the predicted value for 1961 and $\hat{y}_{2021}$ is the predicted value for 2021.

The prediction equations from linear regressions of production (billion kg/year) on year were

$$\hat{y} = \hat{\beta}_0 + \hat{\beta}_1$$
$$= -(4.355 \times 10^{10}) + (2.222 \times 10^{7})X \qquad (4)$$

for the US and

$$\hat{y} = -(5.233 \times 10^{10}) + (2.678 \times 10^{7})X \qquad (5)$$

for Europe, where $\hat{\beta}_0$ is the intercept, $\hat{\beta}_1$ is the slope, and $X$ is the year. The slopes for both were linear, statistically significant, and highly predictive of the increase in strawberry production over the last 60 years. The production increase has been 4.6 million kg/year greater in Europe (26,780,000 kg/year; F-statistic = 644.8; $p \leq 0.0001$; $R^2 = 0.915$) than the US (22,220,000 kg/year; F-statistic = 673.2; $p \leq 0.0001$; $R^2 = 0.918$).

The prediction equations from linear regressions of yield (kg/ha) on year were

$$\hat{y} = -2,032,000 + 1,037X \qquad (6)$$

for the US and

$$\hat{y} = 14,081 - 1.5X \qquad (7)$$

for Europe. The slope for yield was not significantly different from zero in Europe (F-statistic = 0.002; $p = 0.96$; $R^2 = 0.0$), whereas the slope was linear, statistically significant, and highly predictive of yield in the US (F-statistic = 4459; $p \leq 0.0001$; $R^2 = 0.987$). Using the predicted values for 1961 (1557 kg/ha) and 2021 (63,777 kg/ha), yields were predicted to have increased by 3996% in the US between 1961 and 2021. The predicted values for yield in Europe were virtually identical in 1961 (11,140 kg/ha) and 2021 (11,050 kg/ha).

The prediction equations from linear regressions of area harvested (ha) on year were

$$\hat{y} = 149,786 - 64.8X \qquad (8)$$

for the US and

$$\hat{y} = -4,157,095 + 2,136X \qquad (9)$$

for Europe. The slope for area harvested was not significantly different from zero in the US (F-statistic = 3.335; $p = 0.073$; $R^2 = 0.054$), whereas the slope was linear, statistically significant, and highly predictive of

the area harvested in Europe (F-statistic = 196.8; $p \leq 0.0001$; $R^2 = 0.765$).

## Reporting summary
Further information on research design is available in the Nature Portfolio Reporting Summary linked to this article.

## Data availability
Data supporting the findings of this work are available within the paper and its Supplementary Information files. A reporting summary for this article is available as a Supplementary Information file. Additional data related to this study are available upon request. Source data are provided with this paper.

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

## Acknowledgements

We thank several California strawberry growers for providing the farms and production infrastructure needed for our modern hybrid field studies: Thomas AmRhein and Hillary Thomas (Naturipe, Prunedale, CA), Dave Murray (Good Farms, Oxnard, CA), Greg France (Mar Vista Farms, Santa Maria, CA), Dave Peck (Manzanita Berry Farms, Santa Maria. CA), Jose Garcia (Spence Ranch, Salinas, CA), Liz Ponce and Scott Scholer (Lassen Canyon Nursery, Macdoel, CA), and Michael Fahner (Cedar Point Nursery, Dorris, CA). Our studies would not have been possible without their collaboration and participation. We thank Cindy Lopez, Nayeli

Valencia, Eduardo Garcia, Bruce Campopiano, and Randi Famula for their assistance with plant propagation, hybrid development, field experiments, and laboratory analyses. This research was supported by grants to SJK from the United States Department of Agriculture (https://doi.org/10.13039/100000199) National Institute of Food and Agriculture (NIFA) Specialty Crops Research Initiative (#2017-51181-26833 and #2022-51181-38328), California Strawberry Commission (https://doi.org/10.13039/100006760), and the University of California, Davis (https://doi.org/10.13039/100007707).

## Author contributions

S.J.K. and G.S.C. conceived, designed, and planned the modern hybrid field studies. S.J.K., G.S.C., and M.J.F. conceived, designed, and planned the genetic gain and ss-BLUP training population hybrid field studies. M.J.F. conceived and performed the ss-BLUP analyses and change-point analyses. M.J.F., G.S.C., and D.D.A.P. conducted the field experiments. D.D.A.P assembled and curated the pedigree database. M.J.F., D.D.A.P., and S.J.K. performed data analyses. S.J.K. and M.J.F. wrote the manuscript. S.J.K., M.J.F., D.D.A.P, and G.S.C. revised the manuscript.

## Competing interests

The authors declare no competing interests.
