## [Peer Review File · Nature Communications]

Genetic Gains Underpinning a Little-Known Strawberry Green RevolutionReviewers' Comments:

Reviewer #1:

Remarks to the Author:

General comments:

1) This manuscript reports on an exhaustive analysis of breeding gains in strawberry, and introduces application of the term "Green Revolution" to strawberry. This article will certainly be of interest to all in the strawberry world, and will attract broad interest in the plant breeding world. The manuscript is very concisely and clearly written, and to my eye is ready for publication. That being said, I am unable to comment usefully on the nuts and bolts of the statistical analysis. I am confident that these authors know what they are doing in the conduct of this analysis; however, I trust that publication of this manuscript will also be subject to the approval of a reviewer with the necessary statistical expertise.

2) The focus is on the role of the University of California Davis (UCD) breeding program, which has been a major player in strawberry breeding for yield gain. This focus is fine, and is reflective of the magnitude of the UCD contribution as well as the ready availability to these authors of the utilized data. However, I think it would be appropriate to at least acknowledge the existence and contributions of the other major strawberry breeding programs in the U.S., which have also contributed to strawberry yield increases through breeding, notably the University of Florida program, the USDA Beltsville program, smaller programs at Michigan State and Cornell, and private breeding companies such as Driscolls.

Specific comments:

Figure 1 seems incomplete without appropriately placed depiction of a "white" Chilean strawberry (see attached image).

Line 132. Spelling: "upen" should be "upon".

Lines 587-589: Wording is confusing. What is meant here by "earlier"?

A white Chilean strawberry. Pink areas are exposed to direct sunlight.

Reviewer #2:

Remarks to the Author:

This paper makes a case for large genetic gain and increased genetic variance as a consequence of the breeding efforts in strawberry, and especially North American strawberry. The authors have an intelligent approach to support their case. They have created a large population of parent lines and hybrids as a training population for genomic prediction of a relevant population of strawberries spanning a period of 1775 till 2012. They integrate their knowledge on pedigree relations between a wide collection of strawberry genotypes, SNP information and phenotypic records to predict genomic breeding values and genomic variances. By regressing genomic breeding values, better 'genome predicted means' and genome predicted variances on birth / release year of cultivars / genotypes they show that from an estimated change point onwards yield and important yield components for strawberry have undergone an impressive genetic improvement.

Overall, I think the approach chosen by the authors is sound. Still, I found a number of aspects in the quantitative elaboration hard to follow. A first issue is the composition of the training set of genotypes. A number of crosses is made between elite x elite parents and elite x exotic parents, 27 elite and 3 exotic parents in an incomplete factorial. The 27 parents for elite x elite crosses produce 432 hybrids, 13 elite parents are photoperiod sensitive, 14 elite parents are photoperiod insensitive. The 27 elite parents are themselves hybrids with varying genetic relatedness between them. The 3 exotic parents are well known and amply described. With the three exotic parents various numbers of crosses are made with elite parents and also with other exotic parents. The total training population for a genomic prediction model is composed of the offspring produced by all above crosses. It appears to me that the training populations differ slightly between phenotypic traits.

My question(s), can the authors be more clear on which crosses are made between which parents, with which motivation, and how things change from one trait to another? Can they also show / visualize and quantify the genetic composition of the training set? Can they do the same for the set of genotypes for which they want to predict? Can they show the genetic relations that exist between genotypes in the training set and genotype in the set to be predicted? Why is the training set adequate for the prediction task defined by the authors? Are there measures / indicators that show the quality of the genomic prediction models? I acknowledge that the authors refer to other papers for such information, but I think the current paper should be more self-contained on this point.

Just like the genetic part, also the phenotypic part of the training set is not fully clear to me. The phenotypic data come from different locations and years. Why is this set of phenotypic data representative of the conditions for which the authors want a genomic prediction? What is the target population of environments? Is it the current conditions? Is it the conditions for the coming years? How much genotype by location interaction is there, how much genotype by year interaction, how much genotype by location by year interaction? Are these interactions taken into account in any way? Were the environmental conditions between 1775 and now homogeneous? Probably not, so how do we know that past cultivars would not perform better under past conditions than current conditions and would have a genetic advantage in earlier times? The spectrum of diseases will have changed over time, and so will measures to control those diseases. Such changes complicate estimates for genetic gain over the long period over which the authors claim to have information. I found the amount of sampling of environmental conditions rather limited to make such strong conclusions on genetic revolutions in strawberry as the authors do.

This brings me to a final major point, can we get some diagnostic insights about the quality of the genomic predictions as these genomic prediction models underlie the main argument of the paper? I couldn't find any numbers in the paper that show to me that the genomic predictions are reliable. Can some cross validation accuracies and mean squared errors of prediction be given? It is true that genomic heritabilities looked high, but this may be a consequence of some extremely bad genotypes that inflate the heritability by inducing a strong contrast between genotypes that is going to dominate within-genotype differences with a high heritability as a consequence. Especially for disease

resistances such a phenomenon may occur.

A minor point is the reporting of genetic gain in percentages at many points in the paper. When the reference level to which a percentage is calculated is not given, it is hard to interpret such a percentage.

Reviewer #3:

Remarks to the Author:

Feldmann et al's MS presents the change with generations of selection of strawberry plant traits, in particular yield and yield components, fruit quality (soluble sugars, acidity, anthocyanins and ratios), and resistance to diseases. The study is based on phenotypic data collected in one location over two years for a large collection of genotypes. Presented results show rapid genetic progress of all yield components, accompanied by a decrease in soluble content and in disease resistance. It is argued that a major cause of yield increase is the loss of photoperiodic response, which allows longer plant cycle. A main difficulty is that the genetic material used in this study is not clear to me. The 'Method' section mentions 560 hybrid offspring of 30 parents either photoperiod-sensitive or photoperiod insensitive (line 789-). However, Fig 3 presents 6,419 hybrids with known date of release, whereas another paragraph of the 'Method' section mentions 15,649 individuals estimated using ssBLUP. It would be indispensable that the genetic material is better described (possibly with a table), which genotypes were used for what, and which presented values are measured, which are estimated. How were dates of released of offsprings or of hybrids described by ssBLUP known? This is crucial to understand Fig. 3, and is not clear enough.

I had a second difficulty, the objective of the study was not fully clear for me, beyond the fact that strawberry yields increased as much, or more, than those of cereals. Indeed, many questions arise from this study, which are not addressed:

- Was this progress accompanied by an increased genotype x environment interaction and year of release x environment interaction? The analysis of such interactions may shed light on the origins of the very large variability observed at Fig. 3. The protocol used in experiments (one site only) does not allow analysing this point, but this analysis could be performed, at least in part, if yield progress presented in Fig 2 was analysed in smaller areas than the whole USA. For instance, Lobell et al (2021, Nature food) analysed yields in counties over several years, and derived GxE interaction based on these datasets.
- A striking feature of Fig. 3 is the very large variability for nearly all traits, in particular yield components. Some genotypes released recently had yield and yield components equal or lower than those in 1850. This is acknowledged at the end of the result section, but would merit to be analysed more deeply. Can this be linked to inaccurate estimation of the date of release, or of the confounding effect of adding offspring in the analysis? As the figure 3 stands, the general pattern for all studied genotype, denoted with the dashed line, was obviously not followed by many genotypes. It would be indispensable to analyse which genotypes presented a response to breeding and which did not, together with hypotheses that may explain this fact. Perhaps some breeding schemes gave a priority of quality in relation to yield? This would show if a multi-variable study was carried out.
- Although correlations between traits are presented in Fig. 4, this analysis is too superficial as it is carried out over the whole population. I would expect that those genotypes that did not clearly respond to breeding may show different correlations between traits than those with high response (see above, for instance quality vs yield). The construction of a 'phenotypic space', i.e. the structure of correlations within the studied population would be indispensable
- Fig. 5 is difficult to interpret as it is: again, its analysis is too superficial. It is clear that non-photoperiodic genotypes end up with higher yield than photoperiodic ones, but curves have markedly different shapes: were yields of photoperiodic genotypes similar to non-photoperiodic ones at the same age? Are differences due to the cycle duration, and presumably leaf area? Do differences apply to all traits mentioned in Fig. 3? A more detailed analysis would be necessary, so the reader can understand why non photoperiodic genotypes have better yields.

Minor points

- In many paragraphs, the style is not that for a scientific journal. It is understandable that authors are enthusiastic with their results and with the genetic progress of strawberry, but a more factual writing would greatly help the reader to understand the results
- There are many repetitions, which need a careful editing
- The introduction is way too long
- The first paragraph of the result section has a strange status : authors do not provide a quantitative assessment of data presented in Fig. 1, which looks like a figure for an extension journal rather than a demonstration, which is not brought by this paragraph.

Reviewer #1

Reviewer #1 (Remarks to the Author):

General comments:

1) This manuscript reports on an exhaustive analysis of breeding gains in strawberry, and introduces application of the term “Green Revolution” to strawberry. This article will certainly be of interest to all in the strawberry world, and will attract broad interest in the plant breeding world. The manuscript is very concisely and clearly written, and to my eye is ready for publication. That being said, I am unable to comment usefully on the nuts and bolts of the statistical analysis. I am confident that these authors know what they are doing in the conduct of this analysis; however, I trust that publication of this manuscript will also be subject to the approval of a reviewer with the necessary statistical expertise.

RESPONSE

We appreciate the positive comments about our study. Other reviewers raised statistical questions that are addressed below.

2) The focus is on the role of the University of California Davis (UCD) breeding program, which has been a major player in strawberry breeding for yield gain. This focus is fine, and is reflective of the magnitude of the UCD contribution as well as the ready availability to these authors of the utilized data. However, I think it would be appropriate to at least acknowledge the existence and contributions of the other major strawberry breeding programs in the U.S., which have also contributed to strawberry yield increases through breeding, notably the University of Florida program, the USDA Beltsville program, smaller programs at Michigan State and Cornell, and private breeding companies such as Driscolls.

RESPONSE

We appreciate that others have contributed to genetic gains and the global success of strawberry; however, we are not aware of any publicly available data or peer-reviewed publications that document the achievements of other breeding programs in the US or anywhere else for that matter. One of the motivations for our study was to document the domestication history of strawberry using genetic resources (the California population) that played a central role in the phenomenal expansion of strawberry production in the US (California accounts for 88-90% of US production) and more broadly in the growth of strawberry production worldwide. The genetic resources used in our study are directly connected to the century-long history of strawberry breeding at the University of California (1924-present) and most importantly to genetic gains underpinning the Green Revolution documented in our paper. The reviewer acknowledged this but wanted us to acknowledge the “existence and contributions” of other strawberry breeding programs. We understand the point; however, we did not and do not have access to genetic resources or data associated with other breeding programs, several of which are privately held and have never been publicly described or documented. Moreover, the accomplishments of those breeding programs have not been documented by the scientists that have had access to the genetic material and information. We should not be expected to speculate about the “existence and contributions” of other breeding programs to strawberry improvement but appreciate the point the reviewer made. To address this, we added the following paragraph to the discussion (shown in red). We did this as concisely as possible to manage the word count (the new text added 88 words).

The inferences in our study were limited to phenotypes of California population hybrids observed in coastal California environments and genetic resources (clones of hybrid individuals) directly connected to the century-long history of strawberry breeding at UC Berkeley (1924-1945) and UC Davis (1952-present). While other public and private breeding programs

have certainly contributed to genetic gains and the global expansion of strawberry production, their contributions to yield increases across markets and geographies remain undocumented, anecdotal, and speculative.

REVIEWER COMMENT. Figure 1 seems incomplete without appropriately placed depiction of a “white” Chilean strawberry (see attached image).

RESPONSE

The fruit image shared by the reviewer is not an accurate representation of the size and shape of the fruit documented in the botanical illustrations that we used and cited in our paper. Those illustrations were produced by Duchesne, the botanist that discovered the hybrid species. The artist we worked with (Sandra Doyle) used the original artwork of Duchesne to create the watercolors shown in our paper (with fruit shown in red in the original version we submitted). There is ambiguity in the historical records about fruit color. We had the artist repaint the fruit of that parent whitish red not white (for the reasons explained below) and have provided the revised figure.

The reviewer is specifically referencing the *F. chiloensis* plants imported from Chile to France by Frézier in 1714 that became parents of the original spontaneous interspecific hybrids (*F. chiloensis* x *F. virginiana*) that Duchesne discovered (his illustrations were black and white drawings). The fruit of that parent is depicted in the uppermost row (A) of Fig. 1 (Fig. 3 in the revised paper). Frézier (1717) gifted a plant of his original Chilean strawberry to Antoine de Jussieu, “Professor and Demonstrator of the Interior and Exterior of Plants at the King's Garden”, Versailles, France and described the fruit as “whitish red”. That original plant (individual) is extinct so the best we could do is ask the artist to depict the ‘whitish-red’ color of the fruit using the original black and white drawings of Duchesne as model. The artist (Sandra Doyle) painted a new watercolor that we used to represent the Frézier *F. chiloensis* parent, which is shown in the in the upper row (A) of the revised version of Fig. 3. We dug into historical documents and literature to address the point about color. We wanted to share the information we extracted from various publications to support the decision we made to respond to the point raised by the reviewer about fruit color.

- Frézier (1717) described the fruit of his original Chilean strawberry plant as “whitish red”.
- Bailey (1894) stated, apparently incorrectly (P295), that “About 1712, a second species of strawberry reached Europe. This is the *Fragaria Chiloensis*, brought from Chili to Marseilles by Capt. Frézier. It reached England in 1727. It is a stout, thick-leaved shaggy plant which bore a large globular or somewhat pointed late dark colored fruit.” We concluded that plants reaching England from France early on may have been descendants of crosses to *F. virginiana* and were red, not white or whitish red.
- Finn et al. (2013) stated that the Frézier Chilean strawberry plant was “white-fruited”; however, they did not cite Frézier or another scientific authority. We propose sticking with the actual description made by the person that collected the plants in Chile in 1714.
- Gambardella et al. (2016) muddles the picture even more by arguing that the original Frézier plant was not a land race, which is what Finn et al. (2013) and other authors may have assumed, but a wild ecotype. Gambardella et al. (2016) described the fruits of land races as being “large (about 35 mm long) and pale red, pink or white in color, with large, dark-colored achenes” and the fruits of wild ecotypes as being “red colored” and “significantly smaller (22 mm long on average)”. They quoted a translation of Frézier (1717) that described strawberries cultivated in Chile as whitish red in color, which only muddled their claim that the Frézier plant was wild and was not the “white strawberry cultivated by the Mapuche people” (the land race).

You can see the ambiguity in the literature. The color of the fruit of the original Frézier plant is immaterial to the findings in our paper and a problem that cannot be solved without evidence from the original plants, which no longer exist. The most accurate and prudent approach for us here was to cite Frézier (1717) and depict a whitish red fruit in Fig. 3. To that end, we added the following statements in the text and caption for Fig. 3 (shown in red here).

TEXT ADDED. Frézier described fruit of the Chilean *F. chiloensis* parent of the earliest *F. × ananassa* hybrids as “whitish red” (as depicted in Fig. 3A).

TEXT ADDED TO THE FIGURE 3 CAPTION. The color of fruit of the original *F. chiloensis* parent plant gifted by Frézier (1717) to Antoine de Jussieu (Professor and Demonstrator of the Interior and Exterior of Plants at the King's Garden, Versailles, France) was described as “whitish red”.

Reviewer #2 (Remarks to the Author):

This paper makes a case for large genetic gain and increased genetic variance as a consequence of the breeding efforts in strawberry, and especially North American strawberry. The authors have an intelligent approach to support their case. They have created a large population of parent lines and hybrids as a training population for genomic prediction of a relevant population of strawberries spanning a period of 1775 till 2012. They integrate their knowledge on pedigree relations between a wide collection of strawberry genotypes, SNP information and phenotypic records to predict genomic breeding values and genomic variances. By regressing genomic breeding values, better 'genome predicted means' and genome predicted variances on birth / release year of cultivars / genotypes they show that from an estimated change point onwards yield and important yield components for strawberry have undergone an impressive genetic improvement.

Overall, I think the approach chosen by the authors is sound. Still, I found a number of aspects in the quantitative elaboration hard to follow. A first issue is the composition of the training set of genotypes. A number of crosses is made between elite x elite parents and elite x exotic parents, 27 elite and 3 exotic parents in an incomplete factorial. The 27 parents for elite x elite crosses produce 432 hybrids, 13 elite parents are photoperiod sensitive, 14 elite parents are photoperiod insensitive. The 27 elite parents are themselves hybrids with varying genetic relatedness between them. The 3 exotic parents are well known and amply described. With the three exotic parents various numbers of crosses are made with elite parents and also with other exotic parents. The total training population for a genomic prediction model is composed of the offspring produced by all above crosses. It appears to me that the training populations differ slightly between phenotypic traits.

My question(s), can the authors be more clear on which crosses are made between which parents, with which motivation, and how things change from one trait to another? Can they also show / visualize and quantify the genetic composition of the training set? Can they do the same for the set of genotypes for which they want to predict? Can they show the genetic relations that exist between genotypes in the training set and genotype in the set to be predicted? Why is the training set adequate for the prediction task defined by the authors? Are there measures / indicators that show the quality of the genomic prediction models? I acknowledge that the authors refer to other papers for such information, but I think the current paper should be more self-contained on this point.

RESPONSE

We agree with the reviewer that some of the details were a bit obtuse and that the reader needs to refer to previously published work to understand some of these details and find answers to these questions. The specific questions raised by Reviewer #2 are addressed below (using numbers we added).

One of the challenges we faced with our description of some of this information was the word limit in the main text (< 4,000 words). However, we included extensive information on the genetic material in our study in the on-line methods (no word limit). We went completely through our on-line methods and the text and added information to address the suggestions of this reviewer.

1. My question(s), can the authors be more clear on which crosses are made between which parents, with which motivation, and how things change from one trait to another?

RESPONSE

We revised the text to better explain the motivations for the mating and study designs and the sources of phenotypic data for fruit versus disease resistance traits. The confusion about “how things change from one trait to another” stems from our inclusion of analyses for two disease resistance traits. We used two previously published phenotypic datasets for this and cited the respective papers (both from our laboratory). The disease resistance data were collected from highly controlled field experiments done in Davis, CA using plants artificially inoculated with soil-borne pathogens. Separate experiments with biological replicates (clones) of different training population hybrids were conducted for each pathogen over two years. There was significant but not complete overlap between hybrids screened for both pathogens. Those data were collected in disease resistance screening experiments that were completely independent of the yield trial experiments. The latter were conducted under conditions designed to ELIMINATE the confounding effects of diseases caused by soil-borne pathogens. It is IMPOSSIBLE for us to convey the effort required to obtain the high-quality phenotypes and results reported in our paper. Fruit biomass and quality traits were collected from disease-free on-farm testing experiments, whereas disease traits were collected from independent field experiments designed to repeatably and reliably quantify resistance to soil-borne pathogens.

Our ss-BLUP analyses of disease traits provided insights and an important perspective on the breeding history of strawberry. The Green Revolution we describe was achieved by using chemical production practices that eliminated pressure from diseases caused by soil-borne pathogens. The diseases we studied have affected the course of history in strawberry. We improved the text by more clearly and adequately explaining the datasets we used, why we used them, and how and why they differ.

2. Can they also show / visualize and quantify the genetic composition of the training set? Can they do the same for the set of genotypes for which they want to predict? Can they show the genetic relations that exist between genotypes in the training set and genotype in the set to be predicted? Why is the training set adequate for the prediction task defined by the authors?

RESPONSE

We addressed these questions by including results from a principal component analysis of the genomic relationship matrix. Fig. 2 was added to visualize the genetic composition of the training datasets (individuals with genotypes and phenotypes) and prediction datasets (individuals with genotypes but NOT phenotypes). Fig. 2 shows the first two principal components from a principal component analysis of the genomic relationship matrix for 1,406 genotyped individuals used in our genomic prediction analysis. The elite x elite (training) and

elite x exotic (testing) hybrids are identified in that figure. We did our best to explain the details in the text and on-line methods.

The reviewer's suggestions made it clear to us that we needed to more clearly and adequately explain that we produced breeding value estimates for genotyped and ungenotyped individuals. To show and discuss this, we revised the original GEBV x birth year figure (Fig. 4 in the revised paper) and the associated supplementary figures (see Supplementary Fig. S1-S2 in the Supplementary Information PDF). The three figures have the same x-axes and show statistics for genotyped individuals in blue and ungenotyped individuals in gray. This was an important improvement because this also enabled us to show that breeding values were more accurately estimated from genotyped than ungenotyped individuals. The data for genotyped individuals were sufficient for us to tell the scientific story, which was not clear in the original manuscript. We added text to explain this and discuss the results we observed when ungenotyped individuals were included and excluded from the ss-BLUP analysis. We added a figure showing the accuracy of breeding value estimates (requested by the reviewer) and a section describing the accuracy analyses and results (see Supplementary Fig. S1).

The ungenotyped individuals included in our analyses were either extinct or transient (never preserved) but are integral to the genealogy of strawberry. We used pedigree-based estimates of their genetic relationships in the ss-BLUP analysis because they enabled us to show where they fit in domestication history of strawberry. As we explained in the original and revised manuscripts (better in the latter), the purpose of the ss-BLUP analysis was to study and reconstruct historical trends (domestication-associated genetic changes) not to predict the performance of future hybrids (= genomic selection). We strengthened our explanation of this in the revised paper.

Just like the genetic part, also the phenotypic part of the training set is not fully clear to me. The phenotypic data come from different locations and years. Why is this set of phenotypic data representative of the conditions for which the authors want a genomic prediction? What is the target population of environments? Is it the current conditions? Is it the conditions for the coming years? How much genotype by location interaction is there, how much genotype by year interaction, how much genotype by location by year interaction? Are these interactions taken into account in any way? Were the environmental conditions between 1775 and now homogeneous? Probably not, so how do we know that past cultivars would not perform better under past conditions than current conditions and would have a genetic advantage in earlier times? The spectrum of diseases will have changed over time, and so will measures to control those diseases. Such changes complicate estimates for genetic gain over the long period over which the authors claim to have information. I found the amount of sampling of environmental conditions rather limited to make such strong conclusions on genetic revolutions in strawberry as the authors do.

3. Why is this set of phenotypic data representative of the conditions for which the authors want a genomic prediction?

RESPONSE

We collected data on the training population from a single location (Salinas, CA) over two years with multiple harvests of fruit from the onset of production through the summer solstice (June 21). The training population included photoperiod sensitive (pfpf) and insensitive (PF_) hybrids. The former produce fruit through the solstice but cease flowering around the solstice. We harvested through the summer solstice to avoid the confounding effect of photoperiod sensitivity on yield (a point we clearly documented in the paper).

Salinas is a typical coastal California strawberry production environment and the only such environment in California where we have the research infrastructure needed to conduct such an experiment. The other factor that dictated the environment design for our study was cost. These experiments are extremely expensive. The cost of strawberry production is approximately \$227,336 per hectare. We estimate that the expenses associated with conducting the on-farm experiments reported in our paper were in the \$450,000-550,000 range. This estimate includes the on-farm experiments (with modern cultivars = hybrids) that we conducted in Oxnard, CA with photoperiod sensitive hybrids (pfpf) and Santa Maria and Prunedale, CA with photoperiod insensitive hybrids (Pf_). We are confident that the data reported in our paper are sufficient to draw the conclusions we did about the genetic gains that drove the Green Revolution in California. As noted in the original paper, “we collected and analyzed 250,702 phenotypic observations from 257 harvests (46 to 52 harvests/environment) of fruit of 762 hybrids grown on coastal California farms from Oxnard to Prunedale, CA”. We have provided more details on these numbers for the training population experiment (Salinas) and the modern hybrid experiments (Oxnard, Santa Maria, and Prunedale). The latter were FULL SEASON harvests of a smaller number of hybrids (compared to the training population). Those experiments were done on commercial farms that represent the full range of environments and production conditions relevant to our study, predictions, and conclusions.

There is a subtle aspect of the question asked by the reviewer that we were careful to explain in the original paper, but which obviously needed further clarification. We did not apply genomic prediction in the genomic selection sense. Experts in the field (including this reviewer) apply genomic prediction in the *genomic selection* sense where they are using predictions from current genotyped and phenotyped individuals to estimate the breeding values of future genotyped individuals and selecting the latter based on breeding values alone (= genomic selection). With genomic selection, genotypes and phenotypes of training population individuals (current) are used to predict the phenotypes of selection candidates (future individuals) from genotypes alone. This is NOT how we used genomic prediction. We used the genotypes and phenotypes of training population individuals, genotypes of living individuals, and genealogies of both to model PAST genetic changes using genomic prediction approaches. Our approach was explained in the introduction and discussion; however, we recognize that the newness and uniqueness of the ideas we presented needed improvement. We have done our best to do that in the paper.

4. What is the target population of environments? Is it the current conditions? Is it the conditions for the coming years?

RESPONSE

We provided yield data for hybrids tested in every important coastal production environment where strawberries are commercially grown in California. We did our best to clearly state that in the original paper; however, we have gone through the paper to ensure that this is clear.

Our TARGET environments were those found in coastal California where strawberries are currently produced and have been produced since the start of the Green Revolution.

There is no doubt that production practices have changed dramatically over the last 300 years. There is no way for us to replicate how strawberries were grown 60 years ago let alone in the 1800s or 1900s. We were not trying to predict the FUTURE (“coming years”) at all. We modeled genetic changes from PAST to PRESENT and are confident that we accurately portrayed the genetic changes that have been achieved by breeding since 1775. We used hybrids between modern cultivars (elite parents) and wild and nearly wild individuals (exotic parents) to model those genetic changes under CURRENT environmental conditions. We are

confident that the exotic parents and elite x exotic hybrids in our study would perform worse in PAST conditions, e.g., without soil fumigation and other input production practices.

We did our best to make it clear that our analyses were limited to California population hybrids tested in coastal California environments. These are the environments where 88-90% of strawberries are produced in the US, which is why we have argued that the genetic gains reported in our paper drove the expansion of strawberry production in the US. The yield and production increases reported by the UN-FAO fully support our conclusions. Our results and conclusions provide the solid quantitative estimates of the magnitude and direction of genetic changes from breeding in California.

We are confident that the yields (and genetic gains) estimated in our test environments accurately represent yields across our target environments (see the answer to the next question for further support).

5. How much genotype by location interaction is there, how much genotype by year interaction, how much genotype by location by year interaction? Are these interactions taken into account in any way?

RESPONSE

We added statistics and text to answer these questions and address these points. Yes, we accounted for genotype x environment interactions in our statistical analyses. We added estimates of the ratio of genotype-by-environment variance to phenotypic variance to Table 1. They show that genotype-by-environment interactions only accounted for 2-4% of the phenotypic variance for fruit yield, count, weight, and firmness (these traits that were the primary focus of our paper and most important Green Revolution traits). Most importantly, rank changes (cross-over interactions) were not observed among hybrids. Our heritability estimates showed that non-genetic sources of variation did not affect our inferences or conclusions (Table 1).

We developed a new figure to illustrate the absence of genotype-by-environment interactions in the modern hybrid study (see Fig. 7 revised). That figure clearly shows the consistent rankings of hybrids across years and locations and illustrates the profound difference in yield between photoperiod sensitive and insensitive hybrids. We retained the original figure (which showed individual harvest data and cumulative yields) but moved that to supplementary information (see Supplementary Fig. S3). We added estimates of the ratio of genotype-by-environment variance to phenotypic variance for the modern hybrid study to the text, which ranged from 10.6 to 15.6% for yield.

6. Were the environmental conditions between 1775 and now homogeneous? Probably not, so how do we know that past cultivars would not perform better under past conditions than current conditions and would have a genetic advantage in earlier times?

RESPONSE

We are confident that past cultivars would not perform better under past than current conditions. There is a wealth of literature to back this up (we cited the most relevant and seminal sources). We should be expected to speculate on the “genetic advantage” of past cultivars grown in past conditions. We are not sure what the reviewer meant by “past cultivars” or past conditions. We had no way of knowing or duplicating past conditions anyway. We have presented compelling scientific evidence for the genetic changes achieved by breeding in strawberry and stand behind our conclusions. The exotic parents of the hybrids we tested were thoughtfully selected to be progressively more wild (less domesticated). We unequivocally

showed that the fruit yield and weight increases achieved by breeding relative to exotic (past) sources have been substantial in strawberry.

7. The spectrum of diseases will have changed over time, and so will measures to control those diseases. Such changes complicate estimates for genetic gain over the long period over which the authors claim to have information. I found the amount of sampling of environmental conditions rather limited to make such strong conclusions on genetic revolutions in strawberry as the authors do.

RESPONSE

We stand behind our claim that genetic gains from breeding in the CA population (at the University of California, Davis) drove the Green Revolution in California. We stressed in the paper that the yield and production increases observed in the US were the result of genetic gains (improved cultivars) *and* improved production practices. The genetic gain estimates from our common garden experiment are completely aligned with the UN-FAO data documented in the paper. We agree that a larger study would have been valuable; however, we have confidence in our inferences and conclusions. We did the absolute best we could to estimate genetic gains within the limits of the physical and financial resources we had available to produce data.

We partly agree that the spectrum of diseases has changed over time; however, the two diseases we covered in our paper were discovered nearly a century ago and are still problems in strawberry. They were the **ONLY** diseases caused by soil-borne pathogens that were widespread, consistently problematic, and documented in strawberry **BEFORE** the start of the Green Revolution. As explained in the paper and our response to reviewers, our yield trials were conducted under conditions designed to prevent diseases caused by soil-borne pathogens so that we could accurately estimate genetic gains for intrinsic yield. We would prefer to keep the results for the diseases in the paper and believe it is valuable element of the genetic gain story in strawberry.

This brings me to a final major point, can we get some diagnostic insights about the quality of the genomic predictions as these genomic prediction models underlie the main argument of the paper? I couldn't find any numbers in the paper that show to me that the genomic predictions are reliable. Can some cross validation accuracies and mean squared errors of prediction be given? It is true that genomic heritabilities looked high, but this may be a consequence of some extremely bad genotypes that inflate the heritability by inducing a strong contrast between genotypes that is going to dominate within-genotype differences with a high heritability as a consequence. Especially for disease resistances such a phenomenon may occur.

RESPONSE

The reviewer is correct, we omitted accuracy metrics in the original submission and should have provided that information. We have rectified that by adding a figure showing accuracy estimates (Supplementary Fig. S1), adding accuracy estimates in Supplementary Data 11, and a new section to address the accuracy points raised by the reviewer. Supplementary Fig. S1 shows the accuracy = $1 - \sqrt{\text{PEV} / \text{VA}}$ for each hybrid in our analysis. Supplementary Data 11 provides the underlying GEBV and accuracy estimates shown in Supplementary Fig. S1.

One of the *main arguments* in our paper was that genetic gains from breeding drove a Green Revolution in California. This was supported by genetic gains directly estimated from the *observed phenotypes* of the hybrids we studied (Table 1), not from breeding value estimates (Fig. 4; Table 2). The genomic prediction analyses were not needed or used to make that argument. The other *main argument* in our paper was that substantial genetic variation exists in

strawberry and has not been eroded breeding. The ss-BLUP analyses and genomic prediction of additive genetic variances supplied the data and evidence for the latter.

The parents of the hybrids in our study were purposefully selected to sample extant genetic variation in the population that has been the primary source of cultivars and other genetic resources that drove the Green Revolution in California (and hence the US). Those parents included historically and commercially important Green Revolution cultivars, several of which were used as parents and phenotyped in diverse coastal California environments where strawberries are commercially produced.

We used genomic prediction to model domestication-associated genetic changes (both observed and predicted) in strawberry and show that significant genetic variation exists in strawberry (the phenotypes alone were sufficient to show this too).

Reviewer #3 (Remarks to the Author):

Feldmann et al's MS presents the change with generations of selection of strawberry plant traits, in particular yield and yield components, fruit quality (soluble sugars, acidity, anthocyanins and ratios), and resistance to diseases. The study is based on phenotypic data collected in one location over two years for a large collection of genotypes. Presented results show rapid genetic progress of all yield components, accompanied by a decrease in soluble content and in disease resistance. It is argued that a major cause of yield increase is the loss of photoperiodic response, which allows longer plant cycle.

A main difficulty is that the genetic material used in this study is not clear to me. The 'Method' section mentions 560 hybrid offspring of 30 parents either photoperiod-sensitive or photoperiod insensitive (line 789-). However, Fig 3 presents 6,419 hybrids with known date of release, whereas another paragraph of the 'Method' section mentions 15,649 individuals estimated using ssBLUP. It would be indispensable that the genetic material is better described (possibly with a table), which genotypes were used for what, and which presented values are measured, which are estimated. How were dates of released of offsprings or of hybrids described by ssBLUP known? This is crucial to understand Fig. 3, and is not clear enough.

I had a second difficulty, the objective of the study was not fully clear for me, beyond the fact that strawberry yields increased as much, or more, than those of cereals. Indeed, many questions arise from this study, which are not addressed:

- Was this progress accompanied by an increased genotype x environment interaction and year of release x environment interaction? The analysis of such interactions may shed light on the origins of the very large variability observed at Fig. 3. The protocol used in experiments (one site only) does not allow analysing this point, but this analysis could be performed, at least in part, if yield progress presented in Fig 2 was analysed in smaller areas than the whole USA. For instance, Lobell et al (2021, Nature food) analysed yields in counties over several years, and derived GxE interaction based on these datasets.
1. REVIEWER COMMENT. Was this progress accompanied by an increased genotype x environment interaction and year of release x environment interaction? The analysis of such interactions may shed light on the origins of the very large variability observed at Fig. 3.

RESPONSE

Genotype x environment interactions were minimal and cross-over genotype x environment interactions were non-existent in our study. As stated earlier, we added statistics and text to support this.

The origin of the “very large variability observed at Fig. 3” was described in detail in our original manuscript. We explained that phenotypic data for individuals in the historically, scientifically, and commercially important California population were non-existent. We further explained that we purposefully selected the parents of the hybrids to broadly sample extant genetic variation in the California population. We strengthened our explanations of these points in the manuscript.

The genetic variation displayed in Fig. 3 was exactly what we hoped to uncover in our study and, as our heritability estimates and other statistics show, have nothing to do with genotype x environment interactions. The “very large variability” showed that our parent selection strategy achieved the goals of adequately sampling and accurately quantifying genetic variation in the California population and demonstrating that significant genetic variation has persisted in this population. The dispersion of GEBV estimates we reported was not caused by genotype x environment interactions or “year of release x environment interaction”, but by additive genetic variation (see Fig. 6; Supplementary Fig. S2). We checked our paper front to back to ensure we clearly and thoroughly explained these points.

2. REVIEWER COMMENT. The protocol used in experiments (one site only) does not allow analysing this point, but this analysis could be performed, at least in part, if yield progress presented in Fig 2 was analysed in smaller areas than the whole USA. For instance, Lobell et al (2021, Nature food) analysed yields in counties over several years, and derived GxE interaction based on these datasets.

RESPONSE

We understand that the scale and scope of our strawberry study was nowhere close to what can be done in maize, soybean, wheat, and other grain crops where massive public databases exist. Such data simply do not exist for strawberry. We produced and analyzed 250,000 phenotypic observations to support our inferences and conclusions.

We appreciate that studies in maize and soybean like those referenced by the reviewer (Lobell et al. 2014, 2021) provide deep insights into interactions between genotypes and environments, but this has no relevance to our study. Our statistics, heritability estimates, and cumulative marketable fruit yield plots (Table 1, Fig. 7, and Supplementary Fig. S3) show that genotype x environment interactions had little to no impact on conclusions. The reviewer is asking us to answer questions about genotype x environment interactions in strawberry when the data from our experiments directly show that they were unimportant for the hybrids and environments sampled in our study.

The Lobell et al. (2014, 2021) studies are brilliant and interesting but far beyond the scale and scope of anything that can be done in strawberry or that needs to be done to answer the questions addressed in our paper. Lobell et al. (2014) had free access to “field-scale yield data on nearly one million maize and soybean fields in the United States.” We had to produce 100% of the data for our study at a cost of approximately a half million dollars for the field experiments and phenotyping components alone.

- A striking feature of Fig. 3 is the very large variability for nearly all traits, in particular yield components. Some genotypes released recently had yield and yield components equal or lower than those in 1850. This is acknowledged at the end of the result section, but would merit to be analysed more deeply. Can this be linked to inaccurate estimation of the date of release, or of the confounding effect of adding offspring in the analysis? As the figure 3

stands, the general pattern for all studied genotype, denoted with the dashed line, was obviously not followed by many genotypes. It would be indispensable to analyse which genotypes presented a response to breeding and which did not, together with hypotheses that may explain this fact. Perhaps some breeding schemes gave a priority of quality in relation to yield? This would show if a multi-variable study was carried out.

3. REVIEWER COMMENT. A striking feature of Fig. 3 is the very large variability for nearly all traits, in particular yield components. Some genotypes released recently had yield and yield components equal or lower than those in 1850. This is acknowledged at the end of the result section, but would merit to be analysed more deeply. Can this be linked to inaccurate estimation of the date of release, or of the confounding effect of adding offspring in the analysis?

RESPONSE

As explained above, the "very large variability for nearly all traits" is EXACTLY what we hoped to find and see in our study. We observed "very large variability" for every trait and discussed the scientific importance of that finding. Our conclusions were supported by exceptionally high heritability estimates (discussed in greater detail in the revised paper). The comments made by the reviewer suggested to us that the history and composition of the study population must not have been explained with enough clarity in the original paper. We added and rearranged text to address this deficiency as documented below.

- First, the reviewer used the terms "genotypes released" and "release dates". We reported the birth dates of individuals from pedigree records to establish the chronology. The variability displayed in the figure has nothing to do with inaccuracy of "release date". The birth years of the individuals used to illustrate chronological changes in population means and genetic variances were KNOWN from pedigree records.
- Second, 100% of the "genotypes" analyzed in our study were asexually (clonally) propagated hybrid individuals. Strawberry genetic resources (hybrid individuals) are maintained as clones. These include named and 'released' cultivars and, in our unique germplasm collection, hundreds of other asexually propagated hybrid individuals that were never *released* as cultivars. Our study revolved around the historically, scientifically, and commercially important collection of such individuals that have been preserved at the University of California, Davis (the CA population). This population has been the source of the most commercially successful and impactful strawberry cultivars ever developed. The clonal genetic resources (hybrid individuals) phenotyped in our study included important sources of genetic variation for agriculturally important traits that have been preserved in the germplasm collection at the UC Davis, e.g., fruit quality and disease resistance. We purposefully sampled the broadest range of individuals in that collection to develop a comprehensive picture of extant genetic variation in the CA population, something that had not been adequately done over the century-long history of breeding in the CA population.
- Third, there was no "confounding effect of adding offspring in the analysis". The parents in our genomic prediction study were hybrid individuals, as were their offspring (strawberry cultivars are hybrids of hybrids). A strawberry population is analogous to a human population or any domesticated animal population where parents = generation t hybrids and offspring = generation $t + 1$ hybrids. Only a small fraction of the hundreds of thousands of offspring (hybrids) observed since 1924 in the CA population have been "released" as cultivars (77 to be exact).

- Although correlations between traits are presented in Fig. 4, this analysis is too superficial as it is carried out over the whole population. I would expect that those genotypes that did not clearly

respond to breeding may show different correlations between traits than those with high response (see above, for instance quality vs yield). The construction of a 'phenotypic space', i.e. the structure of correlations within the studied population would be indispensable

RESPONSE

With all due respect, we disagree that our analyses were “too superficial” and strongly disagree with the supposition that genetic correlations caused by the pleiotropic effects of genes under selection could be different in genotypes that did not “respond to breeding” (whatever that means) than in genotypes that did. On the contrary, the genetic correlations reported in our paper clearly show and reflect the exact biological patterns and relationships among these traits in strawberry. We cited previous studies, all of which are completely aligned with our findings.

The reviewer claimed that “genotypes that did not clearly respond to breeding may show different correlations between traits than those with high response”. There is not a shred of evidence for this in strawberry. Several authors, including us, have shown that selection for increased fruit weight and firmness was associated with decreases in sugar and acid concentrations. We stand behind our original analyses and conclusions, which perfectly fit with the patterns shown in the genetic gain figure and with findings in previous studies.

- Fig. 5 is difficult to interpret as it is: again, its analysis is too superficial. It is clear that non-photoperiodic genotypes end up with higher yield than photoperiodic ones, but curves have markedly different shapes: were yields of photoperiodic genotypes similar to non-photoperiodic ones at the same age? Are differences due to the cycle duration, and presumably leaf area? Do differences apply to all traits mentioned in Fig. 3? A more detailed analysis would be necessary, so the reader can understand why non photoperiodic genotypes have better yields.

RESPONSE

These comments told us that something must be missing in our narrative and online methods, so we went back to the drawing board. We found the briefness of narrative (keeping under 4,000 words) challenging. We added text to the narrative and on-line methods to add more detail and granularity to the statistics behind Fig. 5.

1. It is clear that non-photoperiodic genotypes end up with higher yield than photoperiodic ones, but curves have markedly different shapes: were yields of photoperiodic genotypes similar to non-photoperiodic ones at the same age? Are differences due to the cycle duration, and presumably leaf area?

RESPONSE

To answer the last question, the differences are caused by sexual reproductive cycle length and growing degree days, which are greater for day-neutral than short-day cultivars. We were only trying to illustrate in the paper that the introgression of a single gene from a wild ancestor (PF) and 60 years of breeding in the CA population created cultivars with phenomenal yields that are DOUBLE those of short-day cultivars.

We disagree that the “curves have markedly different shapes” (this referenced what is now Supplementary Fig. S3). To the contrary, the shapes of the curves are strikingly similar but have different widths, heights, and steepness for reasons explained in the paper. Our data are from commercial on-farm experiments and perfectly illustrate the production cycles used in California to produce 88-90% of the strawberries grown in the US (see Fig. 7 and Supplementary Fig. 3). The curves shown in Supplementary Fig. 3 represent the actual (real world) growth and harvest cycles of strawberries. They are different because the planting, production, and harvest cycles

are different in the three production districts in California, starting with Oxnard (south), Santa Maria (central), and Prunedale (north). These three districts (target environments) cover 100% of the commercial production in California. We added text to clearly explain this.

The widths and heights of the cumulative marketable fruit yield curves shown in Supplementary Fig. S3 perfectly capture strawberry production patterns in California. We tested short-day cultivars in the south zone and day-neutral cultivars in the central and north zones because that is where they are commercially grown. Short-day cultivars grown in the north zone STOP flowering in June, typically by the summer solstice, which means that their yields go to zero. This is about the time of the year when yields start picking up for day-neutral cultivars in the central and north zones.

We don't understand the controversy here or what the reviewer meant by "a more detailed analysis". We are simply reporting full-season, on-farm data to demonstrate the yield differences between short-day and day-neutral cultivars to support our point that the introduction of competitive day-neutral (photoperiod insensitive) doubled yields and that those played a critically important part in the Green Revolution.

The results from these full-season on-farm experiments were designed to provide readers with benchmarks for comparison to our common garden experiment (in Salinas) with nearly 500 hybrids (used for genomic prediction). The genomic prediction training population included both photoperiod sensitive and insensitive hybrids and was only harvested and phenotyped through the summer solstice to avoid the confounding effects of day-length sensitivity. As noted above, photoperiod-sensitive cultivars cease flowering in June, typically by the summer solstice. The fruit harvested by the summer solstice (June 21) was set by flowers produced after May 21. The summer solstice has been a consistently reliable cut-off date for studying photoperiod-sensitive and insensitive genotypes (hybrids) in common garden experiments. Our study designs were carefully planned out to take this critical factor into account.

Minor points

- In many paragraphs, the style is not that for a scientific journal. It is understandable that authors are enthusiastic with their results and with the genetic progress of strawberry, but a more factual writing would greatly help the reader to understand the results

RESPONSE

We respectfully disagree that our writing style "is not suited for a scientific journal". To the contrary, our scientific writing and *storytelling* were guided by advice published in a Nature (<https://www.nature.com/articles/d41586-019-00546-7>) article titled "Essential elements for high-impact scientific writing". We used an active voice, avoided the "passive of modesty", and focused on telling our story rather than chronologically downloading facts. We appreciate that some readers prefer factual, chronological scientific writing. To that end, we went completely through the paper to try to eliminate overly enthusiastic statements and adverbs that were not needed and to ensure that we effectively communicated the important facts. We hope the editor will find that our writing and storytelling faithfully follows the advice in the Nature article and is worthy of publication in Nature Communications.

- There are many repetitions, which need a careful editing

RESPONSE

We carefully searched for repetitive statements and revised the paper accordingly. We did our best to eliminate redundant and repetitive statements.

- The introduction is way too long

RESPONSE

We respectfully disagree. The introduction was less than 700 words long. Specific suggestions were not provided by the reviewer to eliminate information or reduce the introduction. We welcome suggestions from the editor that could improve our paper by shortening the introduction; however, we found it difficult to identify text to eliminate that would not detract from the information needed in the introduction.

- The first paragraph of the result section has a strange status : authors do not provide a quantitative assessment of data presented in Fig. 1, which looks like a figure for an extension journal rather than a demonstration, which is not brought by this paragraph.

RESPONSE

We did not expect a reaction like this, calling the first paragraph “a strange status” and suggesting that this “looks like a figure for an extension journal” (that stung). These comments caused us to reflect on what was missing in our message and how we might have better communicated the information shown in that figure. We agree with the reviewer that the value and importance of the first paragraph was unclear. We decided that that paragraph was not the best way to start the paper. To address this, we removed the first section (paragraph) and integrated the important points from that section into the section where we described genetic gains underpinning the Green Revolution. The fruit shown in the figure and the caption of the figure illustrate qualitative phenotypic changes associated with the quantitative genetic gains reported in our paper.

The comment about us not providing a “quantitative assessment of data presented in Fig. 1” was confusing because that figure provides visual (qualitative) evidence for fruit morphology changes to complement the quantitative changes that were rigorously documented in the paper. Our feeling was that a picture was worth a thousand words. Fig. 1 (now Fig. 3) is a carefully researched and assembled depiction of the domestication history of strawberry from inception through the Green Revolution (the caption clearly documented that). We added statements in the text that hopefully bring the purpose and value of this figure out more clearly for the reader. Numerous excellent papers discuss genetic gains for yield in maize (for example) that show images of a single pair of parents and their hybrid offspring that are far cruder and less detailed than what we are illustrating in revised Fig. 3, e.g., see Fig. 1 in Schnable and Springer (2013), one of the seminal reviews we cited, which shows the parent-hybrid figure for maize that we mentioned (this exact figure and many renditions of it have appeared in numerous seminal papers on heterosis in maize). The development of Fig. 3 in our paper (previously Fig. 1) required a painstaking effort by us and the artist to carefully and accurately document morphological changes in the fruit. We drew upon a vast collection of historic documents, many obscure and difficult to find. Sandra Doyle (the artist) created beautiful artwork (52 individual watercolors) that accurately depicts the size and shape variation among wild and domesticated strawberry individuals, from the early founders to modern cultivars. We would like to retain the figure but understand if the editors want us to remove it. The text we moved and edited is shown in red below.

Fig. 3 illustrates the changes in fruit weight, shape, symmetry, and physical appearance spanning the history of *F. × ananassa* domestication (1714-present) and the recurring importance of small-fruited wild relatives as sources of genetic variation from inception through the early years of the Green Revolution (A, C, F, and I). We depicted fruit of the earliest *F. chiloensis* parent (original Frézier clones) as whitish red (Fig. 3A), as described by Frézier (1717). Surface defects, deformities, and asymmetrical shapes were prevalent among early

hybrids and heirloom cultivars (A-E in Fig. 3) (Darrow 1966; Staudt 2003). The terminal nodes and branches in the chronology (H and J in Fig. 3) highlight the magnitude and speed of the fruit morphology changes that accompanied the 'Green Revolution' genetic gains reported here (Table 1).

RESPONSE

We appreciate that our study had limitations, which are discussed here and were added to the paper. First, our analyses were limited to genetic resources developed and preserved at the University of California, Davis. Second, we did not and do not have access to genetic resources associated with other breeding programs. We are not even sure if such collections exist for other breeding programs because they have never been publicly documented. We have no way of knowing what others have preserved in undocumented clonal genetic resource collections associated with breeding programs. Third, the genetic resources used in our study (the 'CA' population) are connected to the century-long history of strawberry breeding at the University of California (1924-present) and most importantly to genetic gains underpinning the Green Revolution documented here (1952-present). We appreciate that others have contributed to genetic gains and the global success of strawberry; however, those contributions have not been documented. We had no way of knowing or documenting the genetic gains achieved by other public or private institutions. Fourth, unlike maize and other grain crops with well-documented data resources critical for estimating historic genetic gains, public yield data are lacking for strawberry. Finally, our reconstruction of the genealogy and documentation of the breeding history of the CA population was essential for estimating genetic gains and modeling the domestication history of strawberry. We have no way of knowing if the necessary pedigree and genotypic data exist for other populations.

Reviewers' Comments:

Reviewer #1:

Remarks to the Author:

The authors have satisfactorily responded to my original review comments.

Reviewer #3:

Remarks to the Author:

The new version of this paper is more clear, many of my concerns about the genetic material are now addressed with the inclusion of Fig. 2 (answer to reviewer 2's point 2) and sentences that explain the composition of the training set. The date associated with each line is now better described. Although I am not a specialist of the methods used in this MS for genomic prediction, the clarification and added elements allowed understanding the approach in this study.

However, I still have concerns.

1. The first concern is about the genotype x environment interaction, as raised by reviewer 2 and myself. I appreciate the addition of Figure 7, and statements about GxE based on Table 1. However, except if I am mistaken, the statement that GxE was small relies on measurements over two years in one site. This is far from enough to answer reviewer 2's questions 3 (representativity), 4 (target population of environments), and my own questions about GxE.

GxE calculated in one site over two years is not sufficient to back the statement in line 190 of a low GxE, so this statement cannot be left as it is. This would have needed an analysis across more sites or, at least, a study of environmental conditions during in the two experiments, compared with those in the target population of environments, namely the coastal California. Hence, GxE and considerations about the representativity of results remain a flaw of this study.

I agree with authors that reviewer 2's points 6 and 7 are difficult to address. However, this should be discussed, together with the extent of GxE in presented experiments.

2. A second concern is about Fig 3. Images of fruit shape cannot be considered as sufficient evidences to back the statements in lines 284-. A quantitative assessment over the relevant populations would be expected here. Furthermore, no indication is given in the method section on how this data was collected and analysed. Hence, Figure 3 should be either better documented or deleted

3 The point I raised on trait correlations was partially addressed with the comparison of correlations before and after 1954. A First problem is that this figure is hardly commented so the reader cannot appreciate to what extent correlations were conserved before and after 1954. Second, I would have expected a broader analysis in subpopulations, in particular how they possibly differ between hybrids having contrasting response to photoperiod.

4. The writing was improved, but often remains too general. In particular, the beginning of the discussion section, commenting the yield progress in USA and elsewhere, could be shortened and better documented (for example, in no part of the MS it is discussed that other breeding programmes focused on quality, rather than yield). It would be desirable that the discussion dedicates more space to the study itself, in particular its limits.

Minor points

Line 231. The sentence 'with the latter two problem solved' (about anonymous hybrids) is difficult to follow. How were these problems solved?

Line 822-. What is meant when authors state that strawberry breeding other than UC Berkeley and UC Davis have results remain undocumented, anecdotal and speculative ? I understand that scientific papers have been published out of these two universities.

Reviewer #4:

Remarks to the Author:

I have been invited by the Editor to review the revised manuscript and response letter, with the specific task of evaluating whether the authors have adequately addressed the comments from Reviewer #2. Consequently, my primary focus has been on the comments and questions raised by Reviewer #2, as well as the authors' responses. In the following evaluation, the text in black represents the original comments from Reviewer #2, while the text in blue represents my comments, taking into account the authors' responses to Reviewer #2.

Comment #1: It appears to me that the training populations differ slightly between phenotypic traits. My question(s), can the authors be more clear on which crosses are made between which parents, with which motivation, and how things change from one trait to another? Can they also show / visualize and quantify the genetic composition of the training set? Can they do the same for the set of genotypes for which they want to predict? Can they show the genetic relations that exist between genotypes in the training set and genotype in the set to be predicted? Why is the training set adequate for the prediction task defined by the authors? Are there measures / indicators that show the quality of the genomic prediction models?

In the initial submission, there was some confusion regarding the different crosses made to study two disease resistance traits. In the revised manuscript, the authors have improved the clarity and comprehensibility of the text by providing a more detailed and coherent explanation of the datasets used, their purpose, and the distinctions between them.

The authors have incorporated a PCA analysis (Fig 2) to illustrate the genetic relationship between the training datasets (comprising individuals with genotypes and phenotypes) and the prediction datasets (comprising individuals with genotypes but WITHOUT phenotypes). The overlapping regions between the training and prediction sets signify the genetic associations between them, providing justification for the suitability of the training set for the prediction task. The authors have elaborated on these details in the text and in the online methods section.

In the revised manuscript, the authors have incorporated $\text{accuracy} = 1 - \sqrt{\text{PEV} / \text{VA}}$ as a metric to show the performance of their genomic prediction models. They applied this calculation to 6,419 hybrids with known birth years spanning from 1775 to 2015 using ss-BLUP (Fig S1). Nevertheless, I have some questions concerning this supplementary figure, as indicated in my comments regarding the response to Comment #3.

Comment #2: Just like the genetic part, also the phenotypic part of the training set is not fully clear to me. The phenotypic data come from different locations and years. Why is this set of phenotypic data representative of the conditions for which the authors want a genomic prediction? What is the target population of environments? Is it the current conditions? Is it the conditions for the coming years? How much genotype by location interaction is there, how much genotype by year interaction, how much genotype by location by year interaction? Are these interactions taken into account in any way? Were the environmental conditions between 1775 and now homogeneous? Probably not, so how do we know that past cultivars would not perform better under past conditions than current conditions and would have a genetic advantage in earlier times? The spectrum of diseases will have changed over time, and so will measures to control those diseases. Such changes complicate estimates for genetic gain over the long period over which the authors claim to have information. I found the amount of sampling of environmental conditions rather limited to make such strong conclusions on genetic revolutions in strawberry as the authors do.

The authors have clarified in the revision that their research is primarily focused on environments within coastal California, mainly due to cost constraints. It is reasonable to assume that the experimental variation among these coastal areas is not substantial, which might explain why genotype x environment interactions only accounted for a very small portion of the phenotypic variance. However, as acknowledged by the authors, it is undeniable that production practices have undergone significant changes over the past 300 years, and it is practically impossible to replicate how strawberries were cultivated 60 years ago, let alone in the 1800s or 1900s. Given these confounding factors, the inferences and conclusions heavily rely on prediction accuracy. However, the accuracy of

"genomic prediction" for the studied traits appears to be relatively low, ranging from 0.2 to 0.4 (refer to Fig S1). Further discussion on this matter would be beneficial in justifying the chosen "prediction strategy".

Comment #3: This brings me to a final major point, can we get some diagnostic insights about the quality of the genomic predictions as these genomic prediction models underlie the main argument of the paper? I couldn't find any numbers in the paper that show to me that the genomic predictions are reliable. Can some cross validation accuracies and mean squared errors of prediction be given? It is true that genomic heritabilities looked high, but this may be a consequence of some extremely bad genotypes that inflate the heritability by inducing a strong contrast between genotypes that is going to dominate within-genotype differences with a high heritability as a consequence. Especially for disease resistances such a phenomenon may occur.

The authors have integrated $\text{accuracy} = 1 - \sqrt{\text{PEV} / \text{VA}}$ as a metric to assess the performance of their genomic prediction models. This calculation was applied to various traits of 6,419 hybrids, each with known birth years spanning from 1775 to 2015, using ss-BLUP, as presented in Figure S1. I have several questions and suggestions concerning this figure.

- 1) In Fig S1, the definitions of the gray-colored dots and the blue-colored dots are not clear.
- 2) It would be beneficial to provide an explanation for the occurrence of negative accuracy values (below zero).
- 3) Trait heritability often correlates with prediction accuracy. It appears that the average prediction accuracy for these traits was relatively low (ranging from 0.2 to 0.4). Does this imply that the application of the strategy to 'predict' the performance of historical cultivars was not successful? An elucidation of these results would be insightful.
- 4) As suggested by Reviewer #2, conducting a cross-validation could be valuable for assessing the prediction models.

Response to Referee Comments and Questions: Second Round of Reviews

GENERAL RESPONSE TO REFEREE COMMENTS AND QUESTIONS

Key Result #1. We showed that genetic gains from breeding for a few domestication traits at the University of California, Davis underpinned the expansion of strawberry production in California from the 1950s onward. The genetic gains for yield in this nearly 100-year-old breeding program have been substantial and are not well known even inside the strawberry research community. Our goal was to document the historical impact of this breeding program on strawberry production in the US and emphasize the unsustainability of the strawberry Green Revolution that transpired over the last 60 years in California. We stressed that these genetic gains were achieved because of the crop protection afforded by fumigation with an environmentally unfriendly ozone layer-depleting substance (methyl bromide).

Key Result #2. We showed that the origin of the Green Revolution traces to the early breeding and genetics work of Royce S. Bringhurst, an obscure pioneer (outside of strawberry circles) with technical achievements on par with Norman Borlaug, the iconic figure of the wheat Green Revolution. His work was instrumental in building the strawberry industry from the ground up in California. The primary purpose of the pedigree-genomic prediction approach we applied (ss-BLUP) was to *model historic trends* by identifying when breeding began to have an impact and showing how population means and breeding value ranges changed over time. We used piecewise linear regressions of genomic-estimated breeding value on birth year and time-series change-point analyses to illustrate the impact of breeding over time and estimate when genetic changes began to occur in our study population (the California population). Reviewer #4 raised legitimate concerns about the accuracy of our ss-BLUP predictions, which we acknowledge and address in depth below.

Key Result #3. We documented one of Bringhurst's most important achievements: the introduction of photoperiod insensitive cultivars that ultimately doubled yields and transformed strawberry production. He did this in part by introgressing a single gene from a wild relative and selecting for improved performance among perpetual flowering individuals (strawberries are harvested every day of the year in California because of his work and this gene). We showed that the photoperiod insensitive cultivars developed with this gene doubled yields in California. Although this was anecdotally known in a small scientific circle of strawberry specialists, the impact of this work has previously not been rigorously documented or widely communicated. To us this is a remarkable achievement worthy of communication to a wider audience. This is a prime and not widely known example of a domestication gene that has profoundly affected the production of a plant worldwide.

Key Result #4. We showed that phenotypic ranges and additive genetic variation have increased despite a decrease in allelic diversity in the California population. We explained the reasons behind this and discussed our findings in the context of genetic erosion. We pointed out, for example, that discourses on genetic erosion nearly always focus on the negative impacts of modern breeding on 'genetic diversity' and ignore or downplay the positive impacts on 'genetic variation'. We cited and discussed seminal studies on both sides of the issue.

Explaining the Present-Day Yield Difference Between the US and Europe. We emphasized that the yield trend differences between the US and Europe (reported by the UN-FAO) are difficult to comprehend and reconcile. We have done our absolute best to objectively identify and discuss the factors that could be behind those differences. One of the referees was concerned that we did not document the success of other breeding programs and another referee was concerned that we did not state that the present-day yield differences between the US and Europe were caused by European breeding prioritizing quality and US breeders prioritizing quantity. The quantity versus quality claim is inaccurate and overgeneralizes what has transpired in North America and Europe. Several public breeding programs in North America have developed cultivars with improved consumer fruit quality. Those include the University of Florida, USDA-ARS in Corvallis, OR and Beltsville, MD, and Agriculture and Agri-Food Canada in Nova Scotia. Two recent reviews of European breeding programs have emphasized yield as a top priority (Mezzetti et al. 2018; Senger et al. 2022). Senger et al. (2020) stated that “breeding objectives include higher yields and yield stability, lower production costs, and better product quality (Capocasa et al., 2008; Cellon et al., 2018)” and that, “in the past, breeders focused mainly on yield and production costs, but product quality is now a high priority (Mezzetti et al., 2016; Verma et al., 2017).”

Different Fruit Quality Perspectives. The UC breeding program has greatly improved fruit quality from the producer perspective (e.g., reduced perishability, longer shelf-life, and improved external appearance), which has been at the expense of fruit quality from the consumer perspective (Fig. 4-5). Reviewer #2 harshly criticized Fig. 3. We developed this figure in collaboration with a highly skilled artist, Sandra Doyle (<https://www.sandra-doyle.co.uk/about>), to illustrate that the changes and improvements in external physical appearance of strawberries over the course of domestication and to depict the improvement in external physical appearance that accompanied the genetic gains for yield in the UC breeding program from the 1950s onward. We added additional clarifying statements in the manuscript and developed Supplementary Fig. S4 and Supplementary Table 3 to document the literature, botanical illustrations, photographs, and other sources of the images that we provided to Sandra to serve as models for the watercolors she produced for Fig. 3 (both were added to the Supplementary Information PDF). The external physical appearance of the early UC cultivars (even in the 1970s) was inferior to that of the modern cultivars that dominate production today. Fig. 3 accurately depicts the cat-faced and misshapen fruit common in ancient and heirloom cultivars and early UC cultivars and the speed of improvement from the 1950s onward.

Breeding Priorities are Not the Only Factor Behind the Yield Trend Differences Between the US and Europe. We added “breeding priorities” to our list of possible causes of yield differences between the US and Europe; however, the two review articles we cited (Mezzetti et al. 2018; Senger et al. 2022) stated that breeding for yield was and continues to be a high priority in Europe (yield was the first trait identified as important in both reviews). We made compelling arguments that *several* factors collectively explain the yield differences between the US and Europe over the last 60 years (Fig. 1). We added “breeding priorities” to that list, but made the case above that the genetic gains we reported were not a simple matter of North American breeders focusing on quantity (yield) and European breeders focusing on quality.

Documenting the Success of Other Breeding Programs. We cannot reference or discuss the success of other breeding programs because peer-reviewed publications of those success do not exist. We have no doubt whatsoever that the breeding successes (genetic gains) achieved by other programs could be documented, but they haven't. We should not be expected to speculate on undocumented genetic gains or undocumented priorities in other breeding programs. We are confident that the genetic gains we reported for the UC breeding program are accurate and that those genetic gains largely drove the yield and production increases observed in the US since 1960 (as documented by UN-FAO). We acknowledge that high input production practices and soil fumigation with methyl bromide have been equally responsible for the yield and production increases realized in the US.

We have done our absolute best to address and respond to the criticisms and concerns of the referees. The comments and criticisms of the referees are shown in BLACK. The comments made by Referee #2 that were included by Referee #4 are highlighted in RED. Our responses are highlighted in BLUE.

REVIEWER COMMENTS

Reviewer #1 (Remarks to the Author):

The authors have satisfactorily responded to my original review comments.

Reviewer #3 (Remarks to the Author):

The new version of this paper is more clear, many of my concerns about the genetic material are now addressed with the inclusion of Fig. 2 (answer to reviewer 2's point 2) and sentences that explain the composition of the training set. The date associated with each line is now better described. Although I am not a specialist of the methods used in this MS for genomic prediction, the clarification and added elements allowed understanding the approach in this study. However, I still have concerns.

1. The first concern is about the genotype x environment interaction, as raised by reviewer 2 and myself. I appreciate the addition of Figure 7, and statements about GxE based on Table 1. However, except if I am mistaken, the statement that GxE was small relies on measurements over two years in one site. This is far from enough to answer reviewer 2's questions 3 (representativity), 4 (target population of environments), and my own questions about GxE.

GxE calculated in one site over two years is not sufficient to back the statement in line 190 of a low GxE, so this statement cannot be left as it is. This would have needed an analysis across more sites or, at least, a study of environmental conditions during in the two experiments, compared with those in the target population of environments, namely the coastal California.

Hence, GxE and considerations about the representativity of results remain a flaw of this study. I agree with authors that reviewer 2's points 6 and 7 are difficult to address. However, this should be discussed, together with the extent of GxE in presented experiments.

RESPONSE. The statement we made on L190 clearly applies to *the data we observed*. We did not state here or anywhere else that genotype by environment interactions were unimportant in our target environments. We did state, however, that genotype-by-environment interactions were negligible in our genomic prediction experiment.

RESPONSE. We previously discussed $G \times E$ from on-farm trials of modern cultivars across environments where strawberries are commercially grown on a large scale in California (L701-705). Those experiments were conducted in southern, central, and northern CA production districts over two years. The contribution of $G \times E$ variance to the phenotypic variance was slightly greater in those experiments than in our single location \times two year genomic prediction experiment; however, as we clearly showed, cross-over $G \times E$ interactions were negligible in those on-farm experiments too (Fig. 7). We stand by our conclusion that the key findings in our paper (articulated above) were not affected by $G \times E$ interactions. Most importantly, we are 100% confident that if these experiments were repeated, we would get the same results and reach the same conclusions even if the magnitude of the differences between cultivars varied across environments (non-crossover $G \times E$) or if the ranks of some of the cultivars changed across environments (cross-over $G \times E$). We observed minor rank changes among cultivars across environments. The winners have always been in the upper 5-10% of the cultivars across environments and the losers have always been in the lower 5-10% of the cultivars across environments (Fig. 7; Supplemental Table S1 & S2).

RESPONSE. We disagree with the statement that “GxE and considerations about the representativity of results remain a flaw of this study”. The referee does not specifically state what those considerations are or should be and does not explain how $G \times E$ would impact or fundamentally change any of the four key results in our paper (outlined at the beginning of our response to referees). Our heritability estimates clearly show that our results are highly reproducible. We have done our best to temper our conclusions; however, we do not agree with the referee that “our results are unrepresentative and consequently that our study is flawed” (we are paraphrasing here). It is true that we only conducted the larger hybrid study in a single location (Salinas), which happens to be the only coastal CA strawberry production environment where the University of California, Davis has field research facilities. We tested smaller numbers of modern hybrids on farms across the environments relevant to our conclusions. Here is an excerpt from our response to the initial reviews.

RESPONSE. “As noted in the original paper, “we collected and analyzed 250,702 phenotypic observations from 257 harvests (46 to 52 harvests/environment) of fruit of 762 hybrids grown on coastal California farms from Oxnard to Prunedale, CA”. We have provided more granularity on these numbers for the training population experiment (Salinas) and the modern hybrid experiments (Oxnard, Santa Maria, and Prunedale). The latter were FULL SEASON harvests of a smaller number of hybrids (compared to the training population) by commercial growers that reflect the full range of environments and production conditions relevant to our study, predictions, and conclusions.”

RESPONSE. We stand by our position that we have provided phenotypic data from the full range of coastal environments where strawberries are commercially grown and that these data justify our interpretations of the key results reported in our paper. We are confident that our key results and conclusions will stand the test of time. We have not observed significant $G \times E$

interactions in our target environments. We only know of one study that addresses $G \times E$ for domestication traits in strawberries across diverse environments in North America (Mathey et al. 2017). They reported a nearly complete lack of $G \times E$ for several traits among several genotypes (cultivars) phenotyped in California, Oregon, New Hampshire, and Michigan. It would be naive for us to suggest on the basis of Mathey et al. (2017) and our on-farm results that $G \times E$ is unimportant in strawberry. That is not what we have argued or claimed at all. We strongly disagree that our data are insufficient to draw the conclusions we did about historic genetic gains.

2. A second concern is about Fig 3. Images of fruit shape cannot be considered as sufficient evidences to back the statements in lines 284-. A quantitative assessment over the relevant populations would be expected here. Furthermore, no indication is given in the method section on how this data was collected and analysed. Hence, Figure 3 should be either better documented or deleted

RESPONSE. We do not agree with the statement that Fig. 3 does not back up “the statements in lines 284-”. We are not sure what line number the referee meant to put behind the dash. Regardless, we stand behind every statement made from L284-300. The harsh criticism of this figure is perplexing to us. The development of this figure was a significant undertaking. The phenotypes illustrated in that figure were drawn from an extensive collection of documents and published articles and photographs of fruit of wild ecotypes and ancient, heirloom, and modern cultivars. We worked closely with a world class botanical illustrator to ensure that the artwork accurately captured size, shape, and external physical appearance variation of fruit. Her artwork perfectly captured the phenotypes shown in the original source images we provided to her. We agree with the referee that the artwork shown in Fig. 3 needed to be “better documented”. We developed Supplementary Fig. 4 and Supplementary Table 3 to document the literature, botanical illustrations, and other materials used by the artist to develop Fig. 3. These have been added.

3 The point I raised on trait correlations was partially addressed with the comparison of correlations before and after 1954. A First problem is that this figure is hardly commented on so the reader cannot appreciate to what extent correlations were conserved before and after 1954. Second, I would have expected a broader analysis in subpopulations, in particular how they possibly differ between hybrids having contrasting response to photoperiod.

RESPONSE. We agree that the caption for Fig. 5 (additive genetic correlation heat maps) was inadequate. We revised the caption and pointed the reader to the GEBV estimates in Fig. 4 which were used to estimate the statistics shown in Fig. 5.

RESPONSE. The statistics shown in Fig. 4 and 5 were estimated from short-day and day-neutral hybrids tested together through the summer solstice in both years of our genomic prediction study (to avoid the confounding effects of photoperiod sensitivity). There were no differences in the correlations between short-day and day-neutral hybrids (nor would you expect differences). We and others have observed genetic correlations with the same signs in diverse populations, environments, and studies.

4. The writing was improved, but often remains too general. In particular, the beginning of the discussion section, commenting the yield progress in USA and elsewhere, could be shortened and better documented (for example, in no part of the MS it is discussed that other breeding programmes focused on quality, rather than yield). It would be desirable that the discussion dedicates more space to the study itself, in particular its limits.

RESPONSE. We disagree with several points here. We earnestly tried to discuss the limitations of our study and did our best to reinforce that discussion. We eliminated a paragraph that included a discussion about maize that we agree was too general. Our discussion is not perfect, but was written using the expert advice and guidance from two outstanding sources of information: PLoS and Nature. Here are excerpts from PLoS and Nature on writing discussion and conclusion sections.

- PLoS. “The discussion informs readers about the larger implications of your study based on the results. Highlighting these implications while not overstating the findings can be challenging, especially when you’re submitting to a journal that selects articles based on novelty or potential impact. Regardless of what journal you are submitting to, the discussion section always serves the same purpose: concluding what your study results actually mean.”
- Nature. “The discussion is where you can be more reflective. Here you can make more complex interpretations in light of other work, or discuss caveats and future directions.”

RESPONSE. Taking the PLoS and Nature advice into account, we strove to: (a) objectively discuss the larger implications of our study on the basis of *our results* and what was previously known; (b) avoid overstating the implications of our findings; (c) interpret what our results actually mean; (d) discuss the caveats of our approach and study limitations; and (e) interpret our findings “in light of other work” (put them in a broader context).

RESPONSE. The “yield progress” we reported for the US and Europe was based on data compiled by the UN (Fig. 1; Supplementary Data 1). We *directly reported* UN statistics. The reviewer apparently wants us to “better document” why the yield increases reported by the UN have been substantially different between the US and Europe by stating that “other breeding programmes focused on quality, rather than yield”. As noted above, this is an inaccurate generalization and would require wild speculation on our part. We exhaustively searched the literature and are not aware of any peer-reviewed articles or other documentation to support the claim that “other breeding programmes focused on quality, rather than yield”. To the contrary, the two review articles we cited (Mezzetti et al. 2018; Senger et al. 2022) argue otherwise.

RESPONSE. There isn’t a single peer-reviewed article on genetic gains in other strawberry breeding programs for any trait that we can cite either. As we previously pointed out, the genetic gains for yield in other breeding programs have not been documented. Mezzetti et al. (2018) documented *production* increases and trends in different geographic regions but did not document genetic gains for yield and other traits. We cited this review paper and the Senger et al. (2020) review paper because they thoroughly cite previous breeding studies, none of which document historic genetic gains or the impact of genetic gains on production increases in Europe or the US. There is no evidence in the literature to support the claim that the yield

differences observed between California and Europe exist because European breeders focused on quality not quantity. To the contrary, Mezzetti et al. (2018) stated that "... Diamanti et al. [28] suggested that 12 criteria be considered for the acceptance of the new genotypes in the market." Mezzetti et al. (2018) then went on to identify the 12 criteria and yield was the first criteria they listed: "1. crop productivity (i.e. yield) that must be maintained or increased to guarantee widespread farmer acceptance." Finally, Senger et al. (2020), a review authored by a diverse and accomplished group of European scientists working on strawberry and other berries, stated that: "Breeding objectives include higher yields and yield stability, lower production costs, and better product quality (Capocasa et al., 2008; Cellon et al., 2018). In the past, breeders focused mainly on yield and production costs, but product quality is now a high priority (Mezzetti et al., 2016; Verma et al., 2017)."

Minor points

Line 231. The sentence 'with the latter two problem solved' (about anonymous hybrids) is difficult to follow. How were these problems solved?

RESPONSE. We cited two papers from our laboratory in that sentence [2,5] that *fully document* how we reconstructed the breeding history of the CA population and estimated genetic relationships using pedigree and genome-wide SNP profiles. We reworded these sentences to improve the text.

Line 822-. What is meant when authors state that strawberry breeding other than UC Berkeley and UC Davis have results remain undocumented, anecdotal and speculative ? I understand that scientific papers have been published out of these two universities.

RESPONSE. The statements we made (L816-822) were added to our earlier revision of the manuscript because we were criticized for failing to document the contributions of other public and private sector breeding programs to genetic gains in strawberries. We improved the wording and added further documentation to clarify the statements made on L816-822. As we explained then and repeat here, we cannot document the goals, achievements, and genetic gains made by other public breeding programs or any private sector breeding program because: (a) they have not done that (published peer-reviewed papers on the topic) themselves; (b) the data needed to do that are not publicly available; and (c) the genetic resources needed to do that are not publicly available.

Reviewer #4 (Remarks to the Author):

I have been invited by the Editor to review the revised manuscript and response letter, with the specific task of evaluating whether the authors have adequately addressed the comments from Reviewer #2. Consequently, my primary focus has been on the comments and questions raised by Reviewer #2, as well as the authors' responses. In the following evaluation, the text in black represents the original comments from Reviewer #2, while the text in blue represents my comments, taking into account the authors' responses to Reviewer #2.

RESPONSE. Thank you for reading and reviewing our manuscript. Your time and energy are appreciated.

Comment #1: It appears to me that the training populations differ slightly between phenotypic traits. My question(s), can the authors be more clear on which crosses are made between which parents, with which motivation, and how things change from one trait to another? Can they also show / visualize and quantify the genetic composition of the training set? Can they do the same for the set of genotypes for which they want to predict? Can they show the genetic relations that exist between genotypes in the training set and genotype in the set to be predicted? Why is the training set adequate for the prediction task defined by the authors? Are there measures / indicators that show the quality of the genomic prediction models?

In the initial submission, there was some confusion regarding the different crosses made to study two disease resistance traits. In the revised manuscript, the authors have improved the clarity and comprehensibility of the text by providing a more detailed and coherent explanation of the datasets used, their purpose, and the distinctions between them.

The authors have incorporated a PCA analysis (Fig 2) to illustrate the genetic relationship between the training datasets (comprising individuals with genotypes and phenotypes) and the prediction datasets (comprising individuals with genotypes but WITHOUT phenotypes). The overlapping regions between the training and prediction sets signify the genetic associations between them, providing justification for the suitability of the training set for the prediction task. The authors have elaborated on these details in the text and in the online methods section. In the revised manuscript, the authors have incorporated $\text{accuracy} = 1 - \sqrt{\text{PEV} / \text{VA}}$ as a metric to show the performance of their genomic prediction models. They applied this calculation to 6,419 hybrids with known birth years spanning from 1775 to 2015 using ss-BLUP (Fig S1). Nevertheless, I have some questions concerning this supplementary figure, as indicated in my comments regarding the response to Comment #3.

RESPONSE. See below.

REVIEWER #2. Comment #2: Just like the genetic part, also the phenotypic part of the training set is not fully clear to me. The phenotypic data come from different locations and years. Why is this set of phenotypic data representative of the conditions for which the authors want a genomic prediction? What is the target population of environments? Is it the current conditions? Is it the conditions for the coming years? How much genotype by location interaction is there, how much genotype by year interaction, how much genotype by location by year interaction? Are these interactions taken into account in any way? Were the environmental conditions between 1775 and now homogeneous? Probably not, so how do we know that past cultivars would not perform better under past conditions than current conditions and would have a genetic advantage in earlier times? The spectrum of diseases will have changed over time, and so will measures to control those diseases. Such changes complicate estimates for genetic gain over the long period over which the authors claim to have information. I found the amount of sampling of environmental conditions rather limited to make such strong conclusions on genetic revolutions in strawberry as the authors do.

REVIEWER #4. The authors have clarified in the revision that their research is primarily focused on environments within coastal California, mainly due to cost constraints. It is reasonable to assume that the experimental variation among these coastal areas is not substantial, which might explain why genotype x environment interactions only accounted for a

very small portion of the phenotypic variance. However, as acknowledged by the authors, it is undeniable that production practices have undergone significant changes over the past 300 years, and it is practically impossible to replicate how strawberries were cultivated 60 years ago, let alone in the 1800s or 1900s. Given these confounding factors, the inferences and conclusions heavily rely on prediction accuracy. However, the accuracy of "genomic prediction" for the studied traits appears to be relatively low, ranging from 0.2 to 0.4 (refer to Fig S1). Further discussion on this matter would be beneficial in justifying the chosen "prediction strategy".

RESPONSE. Reviewer #2 and #4 raised legitimate concerns about the accuracy of the genomic predictions we reported using ss-BLUP. We agree that the accuracy of prediction of individual genomic-estimated breeding values (GEBVs) was low for many individuals, primarily the older, extinct, and ungenotyped individuals, and that our "inferences and conclusions heavily rely on prediction accuracy". We acknowledge that improved accuracy would have produced less noisy GEBV \times birth year regressions (smaller residuals); however, our inferences about trends were completely aligned with yield and production trends and breeding history. The change points we estimated, for example, were perfectly aligned with the narrative history of breeding in California and the California population and with UN-FAO documented yield increases.

RESPONSE. We added statements about the pitfalls or weaknesses of the ss-BLUP approach. We acknowledge that our ss-BLUP approach stretched the limits of genomic prediction methodology; however, we did not make any grandiose claims from the ss-BLUP analyses. We did state that they pinpointed the origin of the strawberry Green Revolution in California to the 1950s and 60s, that breeding has produced significant transgressive variation for several traits, and that the changes in the population means and ranges are perfectly aligned with breeding history and the effects of direct and indirect selection on traits with well known antagonistic correlations (Fig. 4-5).

RESPONSE. We used ss-BLUP to model historic trends (changes in population means and ranges over time). We did not apply prediction to the problem of identifying selection candidates as you would do in genomic selection. We are confident that the ss-BLUP approach accurately approximated the changes in population means over time (Fig. 4). They were not perfect by any means because the least accurate GEBVs predicted (were shrunk to) the population mean. The caveat with our ss-BLUP analysis is that the further back in time we went and the shallower the genealogy got, the less accuracy we had and the greater the shrinkage of GEBVs to the population mean.

RESPONSE. We regressed GEBVs on birth year and used time-series change-point analyses to visualize what was hypothesized from the narrative breeding history and production statistics (changes in yield and production over time). The trends observed in those analyses were perfectly aligned with historic documentation, breeding records, and UN-FAO production statistics. The gray highlighted points in Fig. 4, S1, and S2 are individuals with pedigree records that were not genotyped (which was the bulk of the least accurate estimates). The GEBVs for many of those individuals were strongly shrunk to the population mean. Although individuals with shallow genealogies dragged the accuracy down, they greatly expanded the analysis to extinct individuals and earlier years in the domestication of strawberry.

RESPONSE. We acknowledge that our ss-BLUP analysis did not perfectly reconstruct the early breeding history (before 1945). We are confident that it accurately reconstructed the modern breeding history (after 1945). The residuals from the GEBV by birth year regressions (shown in Supplemental Fig. S2) capture both genetic variation (signal) and errors (noise). Supplementary Fig. S1 shows that the GEBVs of ungenotyped individuals were less accurately estimated. As accuracy decreased, the magnitude of shrinkage to the population mean increased, which is evident from Fig. 1 (GEBV by birth year regressions) and plots of the residuals (Supplemental Fig. S2). The bottom line is that if we dropped ungenotyped individuals from the analysis (gray points), which were estimated with lower accuracy, our inferences about genetic gains and losses would not change because those inferences were made from observed phenotypes not GEBVs (Table 1). If GEBV x birth year regressions were fit through genotyped individuals only (the blue points), the slopes would have gotten steeper (e.g., more positive for yield and more negative for TSS), which strengthens our inferences and conclusions about historic trends. We acknowledge that the precision of our population mean estimates would increase if we dropped the ungenotyped individuals, most of which are extinct.

REVIEWER #2. Comment #3: This brings me to a final major point, can we get some diagnostic insights about the quality of the genomic predictions as these genomic prediction models underlie the main argument of the paper? I couldn't find any numbers in the paper that show to me that the genomic predictions are reliable. Can some cross validation accuracies and mean squared errors of prediction be given? It is true that genomic heritabilities looked high, but this may be a consequence of some extremely bad genotypes that inflate the heritability by inducing a strong contrast between genotypes that is going to dominate within-genotype differences with a high heritability as a consequence. Especially for disease resistances such a phenomenon may occur.

REVIEWER #4. The authors have integrated $\text{accuracy} = 1 - \sqrt{\text{PEV} / \text{VA}}$ as a metric to assess the performance of their genomic prediction models. This calculation was applied to various traits of 6,419 hybrids, each with known birth years spanning from 1775 to 2015, using ss-BLUP, as presented in Figure S1. I have several questions and suggestions concerning this figure. 1) In Fig S1, the definitions of the gray-colored dots and the blue-colored dots are not clear.

RESPONSE. The definitions of the points are in the figure captions. They are the same as in Fig 4, S1, and S2. The blue points are genotyped hybrids, whereas the gray points are ungenotyped hybrids in the test set.

REVIEWER #4. 2) It would be beneficial to provide an explanation for the occurrence of negative accuracy values (below zero).

RESPONSE. These points effectively reflect model guesses that will vary around the model mean, similar to 0s. They are individuals in the testing population who could have distinctly disconnected pedigrees from the rest of the material. As we have said in the article and in our previous response, the quality of the pedigree and the recentness of pedigree connections plays a strong role in the accuracy.

REVIEWER #4. 3) Trait heritability often correlates with prediction accuracy. It appears that the average prediction accuracy for these traits was relatively low (ranging from 0.2 to 0.4). Does this imply that the application of the strategy to 'predict' the performance of historical cultivars was not successful? An elucidation of these results would be insightful.

RESPONSE. As noted above, the prediction accuracies of GEBVs tended to be lower for ungenotyped than genotyped individuals and to be lower for older than younger individuals because of differences in the depths of the individual genealogies. Our heritabilities were estimated from the phenotypes of training population hybrids and were high. The prediction accuracies for those individuals were high (see our response to the cross-validation question below). The lower prediction accuracies were for ungenotyped and unphenotyped individuals with pedigree records.

REVIEWER #4. 4) As suggested by Reviewer #2, conducting a cross-validation could be valuable for assessing the prediction models.

RESPONSE. We have done that analysis for a companion paper on genomic selection that we are currently preparing for submission. The table below shows the excellent accuracies we got for every trait using the training population (genotyped and phenotyped individuals) for cross-validation. The issue is not the accuracy of our GEBV estimates for the training population or genotyped individuals (blue points in Fig. 4 and Supplementary Fig. S1), but the lower accuracy of the ungenotyped and unphenotyped individuals (gray points in Fig. 4 and Supplementary Fig. S1). The cross-validation accuracies for resistance to *Phytophthora* crown rot and *Verticillium* wilt were previously reported in the papers we cited (Pincot et al. 2020; Jimenez et al. 2023).

Cross-validation estimates of genomic predictive ability ($r(\bar{y}, \hat{G})$) for different genetic models and traits, where the \bar{y} are expected marginal means (EMMs) and \hat{G} are genomic-estimated breeding values (GEBVs). GEBVs were estimated using G-BLUP. Cross-validation was performed by randomly splitting hybrids into subsets of 80% for training and 20% for validating prediction models. Validation hybrids were randomly sampled without stratification. UCD = elite x elite and non-UCD = elite x exotic hybrids.

Population subset	Model	Yield (g/plant)	Count (fruit/plant)	Weight (g/fruit)	Firmness (kg-Force)	TSS (%)	TA (%)	TSS/TA	Anthocyanin ($\mu\text{g/mL}$)
All	G BLUP	0.75	0.65	0.83	0.82	0.51	0.67	0.44	0.58
UCD	G BLUP	0.54	0.59	0.49	0.61	0.25	0.54	0.41	0.57
Non-UCD	G BLUP	0.85	0.70	0.84	0.75	0.54	0.62	0.53	0.63

Response to Reviewer Comments and Criticisms: First Round of Reviews 29 June 2023

Reviewer #1

Reviewer #1 (Remarks to the Author):

General comments:

1) This manuscript reports on an exhaustive analysis of breeding gains in strawberry, and introduces application of the term “Green Revolution” to strawberry. This article will certainly be of interest to all in the strawberry world, and will attract broad interest in the plant breeding world. The manuscript is very concisely and clearly written, and to my eye is ready for publication. That being said, I am unable to comment usefully on the nuts and bolts of the statistical analysis. I am confident that these authors know what they are doing in the conduct of this analysis; however, I trust that publication of this manuscript will also be subject to the approval of a reviewer with the necessary statistical expertise.

We appreciate the positive comments about our study. Statistical experts did raise questions, which we address below.

2) The focus is on the role of the University of California Davis (UCD) breeding program, which has been a major player in strawberry breeding for yield gain. This focus is fine, and is reflective of the magnitude of the UCD contribution as well as the ready availability to these authors of the utilized data. However, I think it would be appropriate to at least acknowledge the existence and contributions of the other major strawberry breeding programs in the U.S., which have also contributed to strawberry yield increases through breeding, notably the University of Florida program, the USDA Beltsville program, smaller programs at Michigan State and Cornell, and private breeding companies such as Driscolls.

RESPONSE

We appreciate that others have contributed to genetic gains and the global success of strawberry; however, we are not aware of any publicly available data or peer-reviewed publications that document the achievements of other breeding programs in the US or anywhere else for that matter. One of the motivations for our study was to document the domestication history of strawberry using genetic resources (the California population) that played a central role in the phenomenal expansion of strawberry production in the US (California accounts for 88-90% of US production) and more broadly in the growth of strawberry production worldwide. The genetic resources used in our study are directly connected to the century-long history of strawberry breeding at the University of California (1924-present) and most importantly to genetic gains underpinning the Green Revolution documented in our paper. The reviewer acknowledged this but wanted us to acknowledge the “existence and contributions” of other strawberry breeding programs. We understand the point; however, we did not and do not have access to genetic resources or data associated with other breeding programs, several of which are privately held and have never been publicly described or documented. Moreover, the accomplishments of those breeding programs have not been documented by the scientists that have had access to the genetic material and information. We should not be expected to speculate about the “existence and contributions” of other breeding programs to strawberry improvement, but appreciate the point the reviewer made. To address this, we added the following paragraph to the discussion (shown in red). We did this as concisely as possible to manage the word count (the new text added 88 words).

The inferences in our study were limited to phenotypes of California population hybrids observed in coastal California environments and genetic resources (clones of hybrid individuals)

directly connected to the century-long history of strawberry breeding at UC Berkeley (1924-1945) and UC Davis (1952-present). While other public and private breeding programs have certainly contributed to genetic gains and the global expansion of strawberry production, their contributions to yield increases across markets and geographies remain undocumented, anecdotal, and speculative.

REVIEWER COMMENT. Figure 1 seems incomplete without appropriately placed depiction of a “white” Chilean strawberry (see attached image).

RESPONSE

The fruit image shared by the reviewer is not an accurate representation of the size and shape of the fruit documented in historical records and the botanical illustrations we cited. Those illustrations were produced by Duchesne, the botanist that discovered the hybrid species. The artist we worked with (Sandra Doyle) used the original artwork of Duchesne to create the watercolors shown in our paper (with fruit shown in red in the original version we submitted). There is ambiguity in the historical records about fruit color. We had the artist repaint the fruit of that parent whitish red not white (for the reasons explained below) and have provided the revised figure.

The reviewer is specifically referencing the *F. chiloensis* plants imported from Chile to France by Frézier in 1714 that became parents of the original spontaneous interspecific hybrids (*F. chiloensis* x *F. virginiana*) that Duchesne discovered (his illustrations were black and white drawings). The fruit of that parent is depicted in the uppermost row (A) of Fig. 1 (Fig. 3 in the revised paper). Frézier (1717) gifted a plant of his original Chilean strawberry to Antoine de Jussieu, “Professor and Demonstrator of the Interior and Exterior of Plants at the King's Garden”, Versailles, France and described the fruit as “whitish red”. That original plant (individual) is extinct so the best we could do is ask the artist depict a ‘whitish red’ color of the fruit she used as a model (from the original black and white drawings of Duchesne). The artist (Sandra Doyle) painted a new watercolor that we used to represent the Frézier *F. chiloensis* parent, which is shown in the in the upper row (A) of the revised version of Fig. 3. We dug into historical documents and literature to address the point about color. We wanted to share the information we extracted from various publications to support the decision we made to respond to the point raised by the reviewer about fruit color.

- Frézier (1717) described the fruit of his original Chilean strawberry plant as “whitish red”.
- Bailey (1894) stated, apparently incorrectly (P295), that: “About 1712, a second species of strawberry reached Europe. This is the *Fragaria Chiloensis*, brought from Chili to Marseilles by Capt. Frézier. It reached England in 1727. It is a stout, thick-leaved shaggy plant which bore a large globular or somewhat pointed late dark colored fruit.” We concluded that plants reaching England from France early on may have been descendants of crosses to *F. virginiana* and were red, not white or whitish red.
- Finn et al. (2013) stated that the Frézier Chilean strawberry plant was “white-fruited”; however, they did not cite Frézier or another scientific authority. We propose sticking with the actual description made by the person that collected the plants in Chile in 1714.
- Gambardella et al. (2016) muddles the picture even more by arguing that the original Frézier plant was not a land race, which is what Finn et al. (2013) and other authors may have assumed, but a wild ecotype. Gambardella et al. (2016) described the fruits of land races as being “large (about 35 mm long) and pale red, pink or white in color, with large, dark-colored achenes” and the fruits of wild ecotypes as being “red colored” and “significantly smaller (22 mm long on average)”. They quoted a translation of Frézier (1717) that described

strawberries cultivated in Chile as whitish red in color, which only muddled their claim that the Frézier plant was wild and was not the “white strawberry cultivated by the Mapuche people” (the land race).

You can see the ambiguity and confusion in the literature. The color of the fruit of the original Frézier plant is immaterial to the findings in our paper and a problem that cannot be solved without phenotypes or other evidence from the original plants, which no longer exist. The most accurate and prudent approach for us here was to cite Frézier (1717) and depict a whitish red fruit in Fig. 3. To that end, we added the following statements in the text and caption for Fig. 3 (shown in red here).

TEXT ADDED. Frézier described fruit of the Chilean *F. chiloensis* parent of the earliest *F. × ananassa* hybrids as “whitish red” (as depicted in Fig. 3A).

TEXT ADDED TO THE FIGURE 3 CAPTION. The color of fruit of the original *F. chiloensis* parent plant gifted by Frézier (1717) to Antoine de Jussieu (Professor and Demonstrator of the Interior and Exterior of Plants at the King's Garden, Versailles, France) was described as “whitish red”.

Reviewer #2 (Remarks to the Author):

This paper makes a case for large genetic gain and increased genetic variance as a consequence of the breeding efforts in strawberry, and especially North American strawberry. The authors have an intelligent approach to support their case. They have created a large population of parent lines and hybrids as a training population for genomic prediction of a relevant population of strawberries spanning a period of 1775 till 2012. They integrate their knowledge on pedigree relations between a wide collection of strawberry genotypes, SNP information and phenotypic records to predict genomic breeding values and genomic variances. By regressing genomic breeding values, better 'genome predicted means' and genome predicted variances on birth / release year of cultivars / genotypes they show that from an estimated change point onwards yield and important yield components for strawberry have undergone an impressive genetic improvement.

Overall, I think the approach chosen by the authors is sound. Still, I found a number of aspects in the quantitative elaboration hard to follow. A first issue is the composition of the training set of genotypes. A number of crosses is made between elite x elite parents and elite x exotic parents, 27 elite and 3 exotic parents in an incomplete factorial. The 27 parents for elite x elite crosses produce 432 hybrids, 13 elite parents are photoperiod sensitive, 14 elite parents are photoperiod insensitive. The 27 elite parents are themselves hybrids with varying genetic relatedness between them. The 3 exotic parents are well known and amply described. With the three exotic parents various numbers of crosses are made with elite parents and also with other exotic parents. The total training population for a genomic prediction model is composed of the offspring produced by all above crosses. It appears to me that the training populations differ slightly between phenotypic traits.

My question(s), can the authors be more clear on which crosses are made between which parents, with which motivation, and how things change from one trait to another? Can they also show / visualize and quantify the genetic composition of the training set? Can they do the same for the set of genotypes for which they want to predict? Can they show the genetic relations that exist between genotypes in the training set and genotype in the set to be predicted? Why is the training set adequate for the prediction task defined by the authors? Are there measures / indicators that show the quality of the genomic prediction models? I acknowledge that the

authors refer to other papers for such information, but I think the current paper should be more self-contained on this point.

We agree with the reviewer that some of the details were a bit obtuse and that the reader needs to refer to previously published work to understand some of these details and find answers to these questions. We have broken the specific questions down to address these points.

One of the challenges we faced with our description of some of this information was the word limit in the main text (< 4,000 words). However, we included extensive information on the genetic material in our study in the on-line methods (no word limit). We have completely gone through our on-line methods and the text and added information to address this.

We have a companion paper developed for submission to Genetics (not yet submitted) titled “Heterosis, the Domestication Syndrome, and Islands of Diversity in the Genomes of Highly Inbred Modern Strawberry Hybrids” that includes in depth analyses of genetic relationships of the genotypes of our training population. An as yet unpublished companion paper is not much help here because, as the reviewer stated, they would appreciate additional information to make the genetic gain paper more self-contained. We have a dilemma because the genetic relationship analyses presented in the heterosis companion paper are critical to the heterosis and domestication syndrome studies described in that paper. We have proposed a solution that hopefully will not prevent us from publishing our genetic relationship figure in the heterosis paper. We developed a figure showing the first two principal components of the 1,406 genotyped individuals used in our genomic prediction analysis. The elite x elite (training) and elite x exotic (testing) hybrids are identified in that figure. We added brief statements in the text to describe why this training population is appropriate for our analysis.

1. My question(s), can the authors be more clear on which crosses are made between which parents, with which motivation, and how things change from one trait to another?

We revised the text to better explain the motivations for the mating and study designs and the sources of phenotypic data for fruit versus disease resistance traits. The confusion about “how things change from one trait to another” stems from our inclusion of analyses for two disease resistance traits. We used two previously published phenotypic datasets for this and cited the papers (both from our laboratory). The disease resistance data were collected from highly controlled field experiments using plants artificially inoculated with soil-borne pathogens. These data were collected in separate experiments because our yield trial experiments were conducted under conditions designed to ELIMINATE the confounding effects of diseases caused by soil-borne pathogens. It is IMPOSSIBLE for us to convey how difficult it is to achieve the quality of results we got (from disease-free on-farm testing) and from controlled experiments designed to repeatably and reliably quantify resistance to soil-borne pathogens.

Taking a step back, we could simplify our paper by removing the analyses of the disease resistance traits. This would not detract from the most important conclusions about genetic gain. We included the two disease traits (and clearly explained the motivation for that) in the paper because of the overriding importance of factors that led to the Green Revolution in the first place. We did a really nice job of putting the genetic gain story in a broader context. We improved the text to better explain the datasets we used, why we used them, and how and why they differ.

2. Can they also show / visualize and quantify the genetic composition of the training set? Can they do the same for the set of genotypes for which they want to predict? Can they show the genetic relations that exist between genotypes in the training set and genotype in the set to be predicted? Why is the training set adequate for the prediction task defined by the authors?

Yes. We included a supplement showing the genetic composition of the training datasets (individuals with genotypes and phenotypes) and prediction datasets (individuals with genotypes but NOT phenotypes). And we did our best to explain the details in the text and on-line methods. The important thing that the reviewer may have missed here is that we had TWO prediction datasets: living individuals with pedigrees and genotypes and extinct individuals with pedigree but NOT genotypes). We obviously couldn't genotype extinct individuals but had their pedigree records and used those to get the best approximation possible (from ss-BLUP) of where they fit in history. We worked hard on the revisions to address these points and explain that our goal was to use genomic prediction to study and reconstruct historical trends instead of predicting the performance of future hybrids (= genomic selection). We did not apply genomic prediction for the purpose of genomic selection.

Just like the genetic part, also the phenotypic part of the training set is not fully clear to me. The phenotypic data come from different locations and years. Why is this set of phenotypic data representative of the conditions for which the authors want a genomic prediction? What is the target population of environments? Is it the current conditions? Is it the conditions for the coming years? How much genotype by location interaction is there, how much genotype by year interaction, how much genotype by location by year interaction? Are these interactions taken into account in any way? Were the environmental conditions between 1775 and now homogeneous? Probably not, so how do we know that past cultivars would not perform better under past conditions than current conditions and would have a genetic advantage in earlier times? The spectrum of diseases will have changed over time, and so will measures to control those diseases. Such changes complicate estimates for genetic gain over the long period over which the authors claim to have information. I found the amount of sampling of environmental conditions rather limited to make such strong conclusions on genetic revolutions in strawberry as the authors do.

3. Why is this set of phenotypic data representative of the conditions for which the authors want a genomic prediction?

We collected data on the training population from a single location (Salinas, CA) over two years with multiple harvests of fruit from the onset of production through the summer solstice (June 21). The training population included photoperiod sensitive (pfpf) and insensitive (PF_) hybrids. The former produce fruit through the solstice but cease flowering around the solstice. We harvested through the summer solstice to avoid the confounding effect of photoperiod sensitivity on yield (a point we clearly documented in the paper).

Salinas is a typical coastal California strawberry production environment and the only such environment in California where we have university infrastructure needed to conduct such an experiment. The other factor that dictated the environment design for our study was cost. These experiments are extremely expensive. The cost of strawberry production is approximately \$227,336. We estimate that the expenses associated with conducting the on-farm experiments reported in our paper were in the \$450,000-550,000 range. This includes the on-farm experiments (with modern cultivars = hybrids) that we conducted in Oxnard with photoperiod sensitive hybrids (pfpf) and Santa Maria and Prunedale, CA with photoperiod insensitive hybrids (Pf_). We are confident that the data reported in our paper are sufficient to draw the conclusions we did about genetic gains that drove the Green Revolution in California. As noted in the original paper, "we collected and analyzed 250,702 phenotypic observations from 257 harvests (46 to 52 harvests/environment) of fruit of 762 hybrids grown on coastal California farms from Oxnard to Prunedale, CA". We have provided more granularity on these numbers for the training population experiment (Salinas) and the modern hybrid experiments (Oxnard, Santa Maria, and Prunedale). The latter were FULL SEASON harvests of a smaller number of hybrids (compared to the training population) by commercial growers that reflect the

full range of environments and production conditions relevant to our study, predictions, and conclusions.

To provide the editors and reviewers with additional background, testing 500 hybrids (

There is a subtle aspect of the question asked by the reviewer that we were careful to explain in the original paper but which appears to still be confusing. We did not apply genomic prediction in the genomic selection sense. The experts in the field (including these reviewers) apply genomic prediction in the *genomic selection* sense where they are using predictions from current genotyped and phenotyped individuals to estimate the breeding values of future genotyped individuals and selecting the latter on the basis of breeding values alone (= genomic selection). With genomic selection, genotypes and phenotypes of training population individuals (current) are used to predict the phenotypes of selection candidates (future individuals) from genotypes alone. This is NOT how we used genomic prediction. We used the genotypes and phenotypes of training population individuals, genotypes of living individuals, and genealogies of both to model PAST genetic changes using genomic prediction approaches. We used the analogy of archaeogenetics to explain what we set out to achieve. Our approach was thoroughly explained in the introduction and discussion; however, we recognize that the newness and uniqueness of the ideas we presented, which are what we thought made the paper interesting, needed further reinforcement. We have done our best to do that in the paper.

4. What is the target population of environments? Is it the current conditions? Is it the conditions for the coming years?

We provided yield data for hybrids tested in every important coastal production environment where strawberries are commercially grown in California. We did our best to clearly state that in the original paper; however, we have gone through the paper to ensure that this is clear.

Our TARGET environments were those found in coastal California where strawberries are currently produced and have been produced since the start of the Green Revolution.

There is no doubt that production practices have changed dramatically over the last 300 years. There is no way for us to replicate how strawberries were grown 60 years ago let alone in the 1800s or 1900s. We are not trying to predict the FUTURE (“coming years”) at all. We are trying to model genetic changes from PAST to PRESENT and are 100% confident that we accurately portrayed the genetic changes achieved by breeding. We used hybrids between modern cultivars (elite parents) and wild and nearly wild individuals (exotic parents) to model those genetic changes under CURRENT environmental conditions.

We did our best to make it clear that our analyses were limited to California population hybrids tested in coastal California environments. These are the environments where 88-90% of strawberries are produced in the US, which is why we have argued that the genetic gains reported in our paper drove the expansion of strawberry production in the US. The yield and production increases reported by the UN-FAO fully support our conclusions. Our results and conclusions provide the first definitive quantitative estimates and documentation of the magnitude of genetic changes from breeding in California.

We are confident that the yields (and genetic gains) estimated in our test environments accurately represent yields across our target environments (see the answer to the next question for further support).

5. How much genotype by location interaction is there, how much genotype by year interaction, how much genotype by location by year interaction? Are these interactions taken into account in any way?

We added statistics and text to answer these questions and address these points. Yes we accounted for genotype x environment interactions in our statistical analyses. We showed that 70-92% of the observed phenotypic variation was heritable, which by definition meant that non-genetic sources of variation had little impact on our inferences or conclusions. To provide more granularity and detail, we added estimates of genotype x environment interaction variances and showed that genotype x environment interactions were non-factors in our studies. Most importantly, rank changes (cross-over interactions) were not observed among hybrids.

In our training population, the GxE interaction is <13% of the phenotypic variance and on average is closer to 0.05 among all of the analyses traits.

For the yield trials (fig 6) the GxE for marketable yield is 10-15% depending on the combination of environments analyzed.

We have provided an additional supplemental figure depicting the final time point for each hybrid among different years and locations. There is some rank change in the middle of the pack, but very little among the top performers, where rank change interactions are most impactful on selection decisions.

6. Were the environmental conditions between 1775 and now homogeneous? Probably not, so how do we know that past cultivars would not perform better under past conditions than current conditions and would have a genetic advantage in earlier times?

We are confident that past cultivars would not perform better under past than current conditions. There is a wealth of literature to back this up (we cited the most relevant and seminal sources). The reviewer seems to be asking us to speculate on the “genetic advantage” of past cultivars grown in past conditions. The science of genetics wasn’t even invented by 1775. We are not sure what the reviewer meant by “past cultivars” and we certainly can’t know or duplicate past conditions. How would we even test this? We should not be expected to speculate on this point either. We have presented compelling scientific evidence for the genetic changes achieved by breeding in strawberry and stand 110% behind our conclusions. The exotic parents of the hybrids we tested were thoughtfully selected to be progressively more wild (less domesticated). We unequivocally showed that the fruit yield and weight increases from exotic (past) sources and modern (current) cultivars have been substantial in strawberry.

7. The spectrum of diseases will have changed over time, and so will measures to control those diseases. Such changes complicate estimates for genetic gain over the long period over which the authors claim to have information. I found the amount of sampling of environmental conditions rather limited to make such strong conclusions on genetic revolutions in strawberry as the authors do.

We stand behind our claim that genetic gains from breeding in the CA population (at the University of California, Davis) drove the Green Revolution in California. We stressed in the paper that the yield and production increases observed in the US were the result of genetic gains (improved cultivars) *and* improved production practices. The genetic gain estimates from our common garden experiment are completely aligned with the UN-FAO data documented in the paper. We freely admit that our study was small in scale, especially when you compare that to what can and has been achieved in grain crops like maize and wheat. We did the absolute best we could to estimate genetic gains with the physical and financial resources we had available to produce data.

We agree that the spectrum of diseases have changed over time; however, the two diseases we covered in our paper are still problems in strawberry and are the ONLY diseases caused by soil-borne pathogens that were prevalent and documented in strawberry BEFORE the start of the Green Revolution. As explained in the paper and our response to reviewers, our yield trials were conducted under conditions designed to prevent diseases caused by soil-borne pathogens. We could eliminate the discussion about the two disease traits without affecting the central message in our paper, although we would like to keep it and believe it is valuable to the article.

This brings me to a final major point, can we get some diagnostic insights about the quality of the genomic predictions as these genomic prediction models underlie the main argument of the paper? I couldn't find any numbers in the paper that show to me that the genomic predictions are reliable. Can some cross validation accuracies and mean squared errors of prediction be given? It is true that genomic heritabilities looked high, but this may be a consequence of some extremely bad genotypes that inflate the heritability by inducing a strong contrast between genotypes that is going to dominate within-genotype differences with a high heritability as a consequence. Especially for disease resistances such a phenomenon may occur.

The reviewer is correct, we omitted accuracy metrics in the original submission and should have provided that information. We have rectified that and added text to address the points raised by the reviewer.

Before explaining the information we added about accuracy, we wanted to address a critically important point about the purpose of the genomic prediction analyses in our paper. One of the *main arguments* in our paper is that genetic gains from breeding drove a Green Revolution in California. This was supported by genetic gains directly estimated from the *observed phenotypes* of the hybrids we studied (Table 1). The genomic prediction analyses were not needed or used to make that argument. The other *main argument* in our paper was that substantial genetic variation exists in strawberry and has not been eroded breeding.

The parents of the hybrids in our study were purposefully selected to sample extant genetic variation in the population that has been the primary source of cultivars and other genetic resources that drove the Green Revolution in California (and hence the US). Those parents included historically and commercially important Green Revolution cultivars, several of which were used as parents and phenotyped in diverse coastal California environments where strawberries are commercially produced.

We used genomic prediction to model domestication-associated genetic changes (both observed and predicted) in strawberry and show that significant genetic variation exists in strawberry (the phenotypes alone were sufficient to show this too).

We have included a new supplemental figure that shows the accuracy = $1 - \sqrt{\text{PEV} / \text{VA}}$ for each hybrid in our analysis. As we stated in our article, the BLUP accuracy increases with genetic and pedigree relatedness to our training populations and is maximal for those testing individuals that were genotyped and do not exclusively rely on pedigree information.

Reviewer #3 (Remarks to the Author):

Feldmann et al's MS presents the change with generations of selection of strawberry plant traits, in particular yield and yield components, fruit quality (soluble sugars, acidity, anthocyanins and ratios), and resistance to diseases. The study is based on phenotypic data collected in one location over two years for a large collection of genotypes. Presented results show rapid genetic progress of all yield components, accompanied by a decrease in soluble content and in disease

resistance. It is argued that a major cause of yield increase is the loss of photoperiodic response, which allows longer plant cycle.

A main difficulty is that the genetic material used in this study is not clear to me. The 'Method' section mentions 560 hybrid offspring of 30 parents either photoperiod-sensitive or photoperiod insensitive (line 789-). However, Fig 3 presents 6,419 hybrids with known date of release, whereas another paragraph of the 'Method' section mentions 15,649 individuals estimated using ssBLUP. It would be indispensable that the genetic material is better described (possibly with a table), which genotypes were used for what, and which presented values are measured, which are estimated. How were dates of released of offsprings or of hybrids described by ssBLUP known? This is crucial to understand Fig. 3, and is not clear enough.

I had a second difficulty, the objective of the study was not fully clear for me, beyond the fact that strawberry yields increased as much, or more, than those of cereals. Indeed, many questions arise from this study, which are not addressed:

- Was this progress accompanied by an increased genotype x environment interaction and year of release x environment interaction? The analysis of such interactions may shed light on the origins of the very large variability observed at Fig. 3. The protocol used in experiments (one site only) does not allow analysing this point, but this analysis could be performed, at least in part, if yield progress presented in Fig 2 was analysed in smaller areas than the whole USA. For instance, Lobell et al (2021, Nature food) analysed yields in counties over several years, and derived GxE interaction based on these datasets.
1. REVIEWER COMMENT. Was this progress accompanied by an increased genotype x environment interaction and year of release x environment interaction? The analysis of such interactions may shed light on the origins of the very large variability observed at Fig. 3.

Genotype x environment interactions were minimal and cross-over genotype x environment interactions were non-existent in our study. We expand out discussion of this point and included the statistical evidence.

The origin of the “very large variability observed at Fig. 3” was described in detail in our original manuscript. We explained that phenotypic data for individuals in the historically, scientifically, and commercially important California population were non-existent. We further explained that we purposefully selected the parents of the hybrids to broadly sample extant genetic variation in the California population. We strengthened our explanation of these points where we could in the manuscript.

The genetic variation displayed in Fig. 3 was exactly what we hoped to find and uncover in our study and, as our heritability estimates and other statistics show, have nothing to do with genotype x environment interactions. The “very large variability” showed that our parent selection strategy achieved the goal of quantifying genetic variation in the California population and demonstrating that significant genetic variation. This had nothing to do with genotype x environment interactions or “year of release x environment interaction”. We checked our paper front to back to ensure we clearly and thoroughly addressed these points.

2. REVIEWER COMMENT. The protocol used in experiments (one site only) does not allow analysing this point, but this analysis could be performed, at least in part, if yield progress presented in Fig 2 was analysed in smaller areas than the whole USA. For instance, Lobell et al (2021, Nature food) analysed yields in counties over several years, and derived GxE interaction based on these datasets.

We understand that the scale and scope of our strawberry study was nowhere to what can be done in maize, soybean, wheat, and other grain crops where massive public databases exist. Such data do not exist for strawberry. We produced and analyzed 250,000 phenotypic observations to support our inferences and conclusions.

We appreciate that studies in maize and soybean like those referenced by the reviewer (Lobell et al. 2014, 2021) provide deep insights into interactions between genotypes and environments, but this has no relevance to our study. Our statistics and cumulative marketable fruit yield plots (Fig. 6) show that hybrid x environment interactions only accounted for 10-15% of the phenotypic variation. The reviewer is asking us to answer impossible questions about genotype x environment interactions in strawberry when the data from our experiments directly show that they were unimportant for the hybrids and environments sampled in our study.

The Lobell et al. (2014, 2021) studies are brilliant and interesting but far beyond the scale and scope of anything that can be done in strawberry or that needs to be done to answer the questions addressed in our paper. Lobell et al. (2014) had free access to “field-scale yield data on nearly one million maize and soybean fields in the United States.” We had to produce 100% of the data for our study at a cost of approximately a half million dollars for the field experiments and phenotyping components alone.

- A striking feature of Fig. 3 is the very large variability for nearly all traits, in particular yield components. Some genotypes released recently had yield and yield components equal or lower than those in 1850. This is acknowledged at the end of the result section, but would merit to be analysed more deeply. Can this be linked to inaccurate estimation of the date of release, or of the confounding effect of adding offspring in the analysis? As the figure 3 stands, the general pattern for all studied genotype, denoted with the dashed line, was obviously not followed by many genotypes. It would be indispensable to analyse which genotypes presented a response to breeding and which did not, together with hypotheses that may explain this fact. Perhaps some breeding schemes gave a priority of quality in relation to yield? This would show if a multi-variable study was carried out.
3. REVIEWER COMMENT. A striking feature of Fig. 3 is the very large variability for nearly all traits, in particular yield components. Some genotypes released recently had yield and yield components equal or lower than those in 1850. This is acknowledged at the end of the result section, but would merit to be analysed more deeply. Can this be linked to inaccurate estimation of the date of release, or of the confounding effect of adding offspring in the analysis?

As explained above, “very large variability for nearly all traits” is EXACTLY what we hoped to find and see in our study. We actually observed “very large variability” for EVERY trait. These comments explained that the history and composition of the study population must not have been explained clearly enough in the paper.

- First, the reviewer used the terms “genotypes released” and “release dates”. We reported the birth dates of individuals from pedigree records to establish the chronology. The variability displayed in the figure has nothing to do with inaccuracy of “release date”
- Second, 100% percent of the “genotypes” shown in Fig. 3 are asexually (clonally) propagated hybrid individuals. Strawberry genetic resources (hybrid individuals) are maintained as clones. These include named and ‘released’ cultivars and, in our instance, hundreds of other asexually propagated hybrid individuals that were never *released* as cultivars. Our study revolved around the historically, scientifically, and commercially important collection of such individuals that have been reserved at the University of

California, Davis (the CA population). This population has been the source of the most commercially successful and impactful strawberry cultivars ever developed. The clonal genetic resources (hybrid individuals) phenotyped in our study included important sources of genetic variation for agriculturally important traits that have been preserved in the germplasm collection at the UC Davis, e.g., fruit quality and disease resistance. We purposefully sampled the broadest range of individuals in that collection to develop a comprehensive picture of extant genetic variation in the CA population, something that had never been done.

- There was no “confounding effect of adding offspring in the analysis”. The parents in our genomic prediction study were hybrid individuals, as were their offspring (strawberry cultivars are hybrids of hybrids). A strawberry population is analogous to a human population or any domesticated animal population where parents = generation t hybrids and offspring = generation $t + 1$ hybrids. Only a small fraction of the hundreds of thousands of offspring (hybrids) observed since 1924 in the CA population have been “released” as cultivars (77 to be exact).

- Although correlations between traits are presented in Fig. 4, this analysis is too superficial as it is carried out over the whole population. I would expect that those genotypes that did not clearly respond to breeding may show different correlations between traits than those with high response (see above, for instance quality vs yield). The construction of a ‘phenotypic space’, i.e. the structure of correlations within the studied population would be indispensable

With all due respect, we completely disagree that our analyses were “too superficial” and strongly disagree with the supposition that genetic correlations caused by the pleiotropic effects of genes under selection could be different in genotypes that did not “respond to breeding” (whatever that means) than in genotypes that did. On the contrary, the genetic correlations reported in our paper clearly show and reflect the exact biological patterns and relationships among these traits in strawberry.

The reviewer claimed that “genotypes that did not clearly respond to breeding may show different correlations between traits than those with high response”. There is not a shred of evidence for this in strawberry. Several authors, including us, have shown that selection for increased fruit weight and firmness is associated with decreases in sugar and acid. We stand behind our original analyses and conclusions, which perfectly fit with the patterns shown in the genetic gain figure.

- Fig. 5 is difficult to interpret as it is: again, its analysis is too superficial. It is clear that non-photoperiodic genotypes end up with higher yield than photoperiodic ones, but curves have markedly different shapes: were yields of photoperiodic genotypes similar to non-photoperiodic ones at the same age? Are differences due to the cycle duration, and presumably leaf area? Do differences apply to all traits mentioned in Fig. 3? A more detailed analysis would be necessary, so the reader can understand why non photoperiodic genotypes have better yields.

These comments told us that something must be missing in our narrative and online methods, so we went back to the drawing board. We found the briefness of narrative (keeping under 4,000 words) challenging. We added text to the narrative and on-line methods to add more detail and granularity to the statistics behind Fig. 5.

1. It is clear that non-photoperiodic genotypes end up with higher yield than photoperiodic ones, but curves have markedly different shapes: were yields of photoperiodic genotypes similar to non-photoperiodic ones at the same age? Are differences due to the cycle duration, and presumably leaf area?

To answer the last question, the differences are caused by cycle length and growing degree days, which are greater for day-neutral than short-day cultivars. We were only trying to illustrate in the paper that the introgression of a single gene from a wild ancestor (PF) and 60 years of non-trivial breeding at UC Davis in the CA population created cultivars with phenomenal yields that are DOUBLE those of short-day cultivars.

We disagree that the “curves have markedly different shapes”. To the contrary, the shapes of the curves are strikingly similar but have different widths, heights, and steepness for reasons explained in the paper and here. Our data are from on-farm experiments with actual farmers and perfectly illustrate the production cycles used in California to produce 88-90% of the strawberries grown in the US. The curves represent the actual (real world) growth and harvest cycles of strawberries. They are different because the planting, production, and harvest cycles are different in the three production districts in California, starting with Oxnard (south), Santa Maria (central), and Prunedale (north). These three districts (target environments) cover 100% of the commercial production in California.

The widths and heights of the cumulative marketable fruit yield curves shown in Fig. 5 perfectly capture strawberry production patterns in California. We tested short-day cultivars in the south zone and day-neutral cultivars in the central and north zones because that is where they are commercially grown. Short-day cultivars grown in the north zone STOP flowering in June, typically by the summer solstice, which means that their yields go to zero. This is about the time of the year when yields really start picking up for day-neutral cultivars in the central and north zones.

We don't understand the controversy here or what the reviewer meant by “a more detailed analysis”. We are simply reporting full-season, on-farm data to demonstrate the yield differences between short-day and day-neutral cultivars to support our point that the introduction of competitive day-neutral (photoperiod insensitive) cultivars with yields that are roughly double those of short-day cultivars played a critically important part in the Green Revolution.

The results from these full-season on-farm experiments were designed to provide readers with benchmarks for comparison to our common garden experiment (in Salinas) with nearly 500 hybrids (used for genomic prediction). The genomic prediction training population included both photoperiod sensitive and insensitive hybrids and was only harvested and phenotyped through the summer solstice to avoid the confounding effects of day-length sensitivity. As noted above, photoperiod-sensitive cultivars cease flowering in June, typically by the summer solstice. The fruit harvested by the summer solstice (June 21) was set by flowers produced after May 21. The summer solstice has been a consistently reliable cut-off date for studying photoperiod-sensitive and insensitive genotypes (hybrids) in common garden experiments. Our study designs were carefully planned out to take this into account.

Minor points

- In many paragraphs, the style is not that for a scientific journal. It is understandable that authors are enthusiastic with their results and with the genetic progress of strawberry, but a more factual writing would greatly help the reader to understand the results

We respectfully disagree that our writing style “is not suited for a scientific journal”. To the contrary, our scientific writing and *storytelling* were guided by advice published in a Nature (<https://www.nature.com/articles/d41586-019-00546-7>) article titled “Essential elements for high-impact scientific writing”. We used an active voice, avoided the “passive of modesty”, and focused on telling our story rather than chronologically downloading facts. We appreciate that some readers prefer factual, chronological scientific writing. To that end, we went completely

through the paper to try to eliminate overly enthusiastic statements and adverbs that were not needed and to ensure that we effectively communicated the important facts. We hope the editor will find that our writing and storytelling faithfully follows the advice in the aforementioned Nature article and is worthy of publication in Nature Communications.

- There are many repetitions, which need a careful editing

We carefully search for repetitive statements and revised the paper accordingly. We did our best to eliminate redundant and repetitive statements.

- The introduction is way too long

We respectfully disagree. The introduction was less than 700 words long. Specific suggestions were not provided by the reviewer to eliminate information or reduce the introduction. We welcome suggestions from the editor that could improve our paper by shortening the introduction; however, we found it difficult to identify text to eliminate that wouldn't detract from the information needed to provide background and introduce the purpose of our study.

- The first paragraph of the result section has a strange status : authors do not provide a quantitative assessment of data presented in Fig. 1, which looks like a figure for an extension journal rather than a demonstration, which is not brought by this paragraph.

We did not expect a reaction like this, calling the first paragraph “a strange status” and suggesting that this “looks like a figure for an extension journal”. These comments caused us to reflect on what was missing in our message and how we might have better communicated the information shown in that figure. We agree with the reviewer that the first paragraph needed work. We decided that paragraph was not the best way to start the paper. To address this, we removed the first section (paragraph) and worked hard to integrate the important points from that section into the section where we described genetic gains underpinning the Green Revolution (we open the paper with that section now). We pointed out the importance of the introgression of genetic variation from wild ancestors over the entire history of strawberry breeding, which extends through the first 20 years of the Green Revolution (those ancestors are depicted in Fig. 3). The fruit shown in the figure and the caption of the figure captured the qualitative historical details behind the genetic gains, which is what we were trying to achieve.

The comment about us not providing a “quantitative assessment of data presented in Fig. 1” was confusing because we were providing visual (qualitative) evidence for fruit morphology changes to complement the quantitative changes that were rigorously documented in the paper. Our feeling was that a picture was worth a thousand words. Fig. 1 (now Fig. 3) is a carefully researched and assembled depiction of the domestication history from inception through the Green Revolution (the caption clearly documented that). We added statements in the text that hopefully bring the purpose and value of this figure out more clearly for the reader. Numerous excellent papers discuss genetic gains for yield in maize (for example) that show images of a single pair of parents and their hybrid offspring that are far cruder and less detailed than what we are illustrating in Fig. 3, e.g., see Fig. 1 in Schnable and Springer (2013), one of the seminal reviews we cited, which shows the parent-hybrid figure for maize that we mentioned (this exact figure and many renditions of it have appeared in numerous seminal papers on heterosis in maize). The development of Fig. 3 in our paper (previously Fig. 1) required a painstaking effort by us and the artist to carefully and accurately document morphological changes in the fruit. We drew upon a vast collection of historic documents, many obscure and difficult to find. Sandra Doyle (the artist) created beautiful artwork (52 individual watercolors) that accurately depicts size and shape variation among wild and domesticated strawberry individuals, from the

early founders to modern cultivars. We would like to retain the figure but understand if the editors want us to remove it. The text we moved and edited is shown in red below.

Fig. 3 illustrates the changes in fruit weight, shape, symmetry, and physical appearance spanning the history of *F. × ananassa* domestication (1714-present) and the recurring importance of small-fruited wild relatives as sources of genetic variation from inception through the early years of the Green Revolution (A, C, F, and I). We depicted fruit of the earliest *F. chiloensis* parent (original Frézier clones) as whitish red (Fig. 3A), as described by Frézier (1717). Surface defects, deformities, and asymmetrical shapes were prevalent among early hybrids and heirloom cultivars (A-E in Fig. 3) (Darrow 1966; Staudt 2003). The terminal nodes and branches in the chronology (H and J in Fig. 3) highlight the magnitude and speed of the fruit morphology changes that accompanied the 'Green Revolution' genetic gains reported here (Table 1).

We appreciate that our study had limitations, which are discussed here. First, our analyses were limited to genetic resources developed and preserved at the University of California, Davis. We did not and do not have access to genetic resources associated with other breeding programs, several privately held and have never been publicly documented. We cannot even know what others hold in undocumented clonal genetic resource collections associated with breeding programs. Third, the genetic resources used in our study (the 'CA' population) are connected to the century-long history of strawberry breeding at the University of California (1924-present) and most importantly to genetic gains underpinning the Green Revolution documented here (1952-present). We appreciate that others have contributed to genetic gains and the global success of strawberry; however, those contributions have not been documented. We had no way of knowing or documenting the genetic gains achieved by other public or private institutions. We assert that genetic resources developed at UC Davis were fundamentally important to the little-known strawberry Green Revolution. Fourth, unlike maize and other grain crops with well-documented data resources critical for estimating historic genetic gains, public yield data are either completely lacking or grossly inadequate in strawberry. Finally, our reconstruction of the genealogy and documentation of the breeding history of the CA population was essential for estimating genetic gains and modeling the domestication history of strawberry. We have no way of knowing if the necessary pedigree and genotypic data exist for other populations.

Reviewers' Comments:

Reviewer #3:

Remarks to the Author:

This is the third time I review this MS. I have raised concerns about GxE in the last two reviews, the current version does not solve them.

It is expected that GxE is small when the variations of E are limited. The same applies to heritability, which is high here but would probably decrease if more contrasting environments were taken into account. I respect author's statement that results would be the same if more contrasting environments were explored, but my own experience suggests the opposite so I cannot be convinced

Authors also explain at length why they do not follow other suggestions by referees about the necessity of a better quantitative analysis, in particular for Fig. 3 or for cross validation. Again, I respect that, but this cannot convince me.

Overall, authors may well be right in not following reviewer's points, but as a reviewer I regret to say that, because these points were not addressed, I cannot follow the conclusion of the MS. Having said that, the decision belongs to the editor

Reviewer #4:

Remarks to the Author:

The authors have satisfactorily addressed my comments and questions.

Response to Referees

Genotype-by-Environment Interaction

We have done our best to address the $G \times E$ interaction questions and criticisms raised by Referee #4. The additions and improvements include:

- discussing the limitations of our studies in greater depth with particular attention to genotype-by-environment interactions;
- providing additional experimental evidence and citations to previous studies (Shaw and Larson 2001, 2008; Knapp et al. 2023) that document the consistency and stability of strawberry hybrid performance (absence of consequential cross-over interactions) observed over the last 30 years in coastal California environments where strawberries are commercially grown and where 88-90% of strawberries are produced in the US;
- adding labels to Figure 2 to highlight the absence of significant yield rank changes (absence of cross-over interactions) between check cultivars in our 'modern hybrid benchmarking' study (3 locations x 2 years), a pattern observed and replicated in studies conducted in at least 10 coastal California locations over nearly 30 years (Shaw and Larson 2001, 2008; Knapp et al. 2023);
- moving the subsection describing the modern hybrid benchmarking study to the beginning of the RESULTS section, which improved the flow and sequencing of information, provided evidence for the stability of hybrid performance from multiple locations and years (Figure 2; Supplementary Figure S1; Supplementary Table 1), and highlighted that the introduction of hybrids carrying the *PERPETUAL FLOWERING* gene doubled yields in California;
- adding Figure 4 and associated statistics that highlight the stability of hybrid performance in our 'genetic gain' study (1 location x 2 years);
- discussing in detail that the scope of that study was limited by the number of environments;
- reiterating that the UN-FAO data unequivocally show that: (a) yields have increased 2,755% in the US (88-90% of which are produced in California); (b) the yield increases reported for the US include data from states and environments where the yields are lower than California; and (c) that our genetic gain estimates are aligned with the linear yield increases in the US reported by the UN-FAO.

L128-132 (Introduction)

We added statements in the INTRODUCTION to document that UC Davis cultivars (hybrids) developed in a single location (either Irvine or Watsonville, CA) have been commercially competitive and successful in 82 countries over the last 50 years. We added these statements to emphasize that $G \times E$ interactions have not prevented the accurate identification of outstanding hybrids that have achieved commercial success in an incredible range of environments far beyond the single environment where they were identified. The worldwide impact of these cultivars and of the Green Revolution breeding accomplishment at UC Davis are little-known. We are not arguing that $G \times E$ interactions are non-existent or unimportant. We are arguing and have demonstrated in our paper that $G \times E$ interactions are primarily caused by changes in the magnitude of differences among cultivars (hybrids), and not by dramatic changes in ranks among hybrids across environments (direction). The ranks changes observed in our study and the previous studies we cited have been small, as discussed below and in the paper.

L225-252 (Modern Hybrid Benchmarking Study and Previous Studies)

We improved and expanded the discussion about genotype-by-environment interactions in this paragraph. We cited previous studies (Shaw and Larson 2001, 2008; Knapp et al. 2023) that corroborate our claim that $G \times E$ interactions were inconsequential and did not affect our conclusions about genetic gains or the stability of hybrid performance. There are at least 20 years of data from hybrid performance testing in coastal California environments that demonstrate the stability of yield of strawberry cultivars (hybrids) across locations and years (reviewed and cited in the revised manuscript). We are not arguing and have not argued that $G \times E$ interactions are non-existent; however, we have argued that they were inconsequential in our studies and previous studies. Most importantly, cross-over $G \times E$ interactions (rank changes) between the best cultivars (e.g., the highest yielding cultivars) and worst cultivars (e.g., the lowest yielding cultivars) were non-existent in our study. We added labels to the high and low yielding check cultivars in Figure 2 that clearly illustrate this. The phenotypic means of those check hybrids highlight the strong positive rank correlations that we observed for yield among hybrids across environments. The same patterns have been reported in previous studies of UC Davis cultivars tested in diverse coastal California locations (Shaw and Larson 2001, 2008; Knapp et al. 2023).

The on-farm locations used in our 'modern hybrid' and 'genetic gain' studies cover the full range of coastal (target) environments where strawberries are commercially produced in California. We provided strong evidence for the stability of hybrid performance across these environments. The most commercially successful and highest yielding cultivars known were included in every experiment and were tested in eight environments (4 locations x 2 years) over three growing seasons (2015-16, 2016-17, and 2017-18). This is a critical point because the

L396-429 (Genetic Gain Study)

This paragraph provides an expanded and improved discussion about the $G \times E$ interaction question raised by Referee #4. Referee #4 appropriately criticized our previous narrative, which we agree lacked sufficient depth and detail. To address this, we discussed the limitations of the genetic gain study in depth, added a figure (Figure 4) that provides a better foundation for discussing the results of that study, and cited previous studies that substantiate the stability of hybrid performance that we observed is a pattern that has been consistently observed for modern cultivars (hybrids) tested in coastal California environments over the last 20 years or more (Shaw and Larson 2001, 2008; Knapp et al. 2023).

We completely agree with Referee #4 that deeper sampling of environments would likely increase non-genetic variances and decrease heritability and that part of the increase in non-genetic variances could be caused by $G \times E$ interactions; however, there is no evidence from our studies or previously published studies to suggest that $G \times E$ interactions would invalidate our conclusions about genetic gains. We explain below why we have made that claim and why we are confident that our results and conclusions will stand the test of time.

We have shown and argued that cross-over interactions were inconsequential in the four hybrid performance experiments reported in our paper (3 locations x 2 years for modern hybrids developed since 1988 and 1 location x 2 years for genetic gain study hybrids). Figure 4 shows the phenotypic means of 545 hybrids observed over two years in Salinas, CA (the genetic gain study). We included the between-year rank correlation estimates in Figure 4, which were strongly positive for fruit yield, count, weight, and firmness (0.79-0.91). These are the four traits that drove the Green Revolution in California (Figure 4 A-D). These data demonstrate the incredible stability of hybrid performance and absence of *significant* cross-over genotype-by-environment interactions across our studies. That claim is further supported by the data shown in a paper we recently published (Knapp et al. 2023) where we compared the yield of a newly

developed ‘summer-plant’ cultivar (‘UC Eclipse’) with ‘Portola’, the cultivar that currently dominates the summer-plant market in California. ‘Portola’ also happens to be one of the check cultivars used in our hybrid benchmarking study (labeled in Figure 2) and was the parent of 73 elite x elite and 15 elite x exotic hybrids in our genetic gain study (Figure 4; Supplementary Table 2). The yield of ‘UC Eclipse’ was 54% greater than ‘Portola’ across three locations (Ventura, Nipomo, and Santa Maria, CA) over three years in the study we cited (Knapp et al. 2023). We observed *zero yield rank changes* between ‘UC Eclipse’ and ‘Portola’ across these nine coastal California environments. This further supports and strengthens our conclusions about genetic gains because some of the *best hybrids* in our genetic study had ‘Portola’ as a parent. We could not cite Knapp et al. (2023) earlier because it was both submitted and accepted for publication after we submitted our previous revision of this paper.

Figure R1. Three genotype-by-environment interaction scenarios are portrayed for the 10 highest yielding hybrids (black lines) and 10 lowest yielding hybrids (red lines) observed in our Salinas, CA genetic gain study over the 2017-18 and 2017-18 growing seasons. The left panel displays the phenotypic means that were actually observed for those hybrids. The middle panel (simulated magnitude G x E interaction) portrays a hypothetical situation where the G x E interaction is substantial and caused by a change in the magnitude of differences between the elite x elite hybrids (black lines) and elite x exotic hybrids (red lines) between years. The right panel (simulated cross-over G x E interaction) portrays a hypothetical situation where the G x E interaction is substantial and caused by a change in the differences between the elite x elite hybrids (black lines) and elite x exotic hybrids (red lines) between years. The middle and right panels are the classic G x E interaction patterns found in textbooks.

We understand and appreciate that the single location limited the scope of our genetic gain study. We added statements that acknowledge that and have done our best to qualify and temper our conclusions accordingly. We concede that the *magnitude* of the genetic gains we reported could be overestimated because of the underestimation of G x E interactions; however,

we have argued that the *direction* of the genetic gains are 100% accurate and would not change. To highlight this, we are attaching a figure that illustrates the yields of the 10 lowest and 10 highest yielding hybrids in the genetic gain study (shown in the the left panel labeled “Observed” in Figure R1). Those include the low and high yielding hybrids we used to estimate genetic gains (EMM1 - EMM2), where EMM1 is the phenotypic mean for the highest yielding hybrid and EMM2 is the phenotypic mean for the lowest yielding hybrid.

The high yielding hybrids shown in Figure R1 are the same hybrids that were among the high yielding in the modern hybrid benchmarking study (3 locations x 2 years), e.g., Royal Royce, Valiant, Fronteras, and Victor (which are now labeled in Figure 2 of the revised manuscript). To put the data from our studies into perspective with the data cited in Knapp et al. (2023), ‘Portola’ was among the lower yielding hybrids in our modern hybrid benchmarking study (Figure 2), but produces yields that are substantially greater than the low yielding elite x exotic hybrids (EMM2) we used to estimate genetic gains (Figure 4 and Figure R1 below).

The probability that the yields of low yielding (elite x exotic) hybrids (which included a wild relative as one of the parents and an heirloom cultivar as the other parent) would even come close to or surpass the yields of the high yielding hybrids in our study (which are the most high yielding hybrids known) when tested in additional coastal California environments is *zero*. These *hypothetical* G x E scenarios are portrayed in the middle and right panels of Figure R1. The middle panel depicts magnitude G x E interactions where the yields of elite x exotic hybrids (red lines in Figure R1) were equal to the highest yielding elite x elite hybrids in 2017 and dramatically different in 2018. The likelihood of our worst elite x exotic hybrids yielding anywhere close to our best elite x elite hybrids is zero. The cross-over G x E scenario depicted in the right panel of Figure R1 is even less likely. The simulated’ scenarios shown in Figure R1 highlight the implausibility that our conclusions about genetic gains are wrong.

To back this up further, the yields of the two highest yielding hybrids (16C086P019 and 16C565P002) in the genetic gain study were 1,143.6 and 1,071.9 g/plant, respectively (Supplementary Table 2). The yields of the two lowest yielding hybrids (16C094P022 and 16C094P022) were 16.2 and 30.1 g/plant, respectively (Supplementary Table 2). The numbers (e.g., 16C086P019) are the UC Davis IDs for the hybrids in Supplementary Table 2. Supplementary Table 2 is an EXCEL file with the estimated marginal means (EMMs) for each trait and pedigrees of the hybrids.

The parents of the low yielding hybrids in the genetic gains study were ‘Puget Reliance’ (an heirloom cultivar developed for production in the Pacific Northwest) and ‘Del Norte’ (an ecotype of the wild relative *F. chiloensis*). These were the wildest (least domesticated) hybrids in our genetic gain study. We have argued that testing hybrids from upper and lower tails of the yield distribution in additional environments anywhere in the world would *not* result in the disappearance of statistically significant differences between the two groups of hybrids shown in Figure R1. As we explained above, the simulated or hypothetical G x E scenarios depicted in Figure R1 would require significant magnitude G x E interactions where the differences between hybrids from the two extremes disappear in certain environments or where the ranks of hybrids were reversed (an extreme cross-over G x E interaction where the low yielding hybrids become the high yielding hybrids and vice versa). The difference between the best and worst hybrids (1,143.6 - 16.2 = 1,127.2 g/plant) would drop to zero in the first case (as depicted by the *magnitude* G x E interaction shown in the middle panel of Figure R1) or completely change sign (become -1,127.2) in the second case (as depicted by the *cross-over* G x E interaction shown in the right panel in Figure R1). The likelihood of either of these scenarios is zero for the hybrids we used to estimate genetic gains.

We have thoroughly documented the genetic backgrounds of the elite and exotic parents of hybrids developed for the genetic gain study. The elite parents include the highest yielding

cultivars known (strawberry cultivars are hybrids of hybrids, analogous to individuals in human populations). The exotic parents include a wild relative ('Del Norte') and a hybrid between an heirloom cultivar developed in the 1920s ('Lassen') and an extinct wild relative ('Oso Flaco'). These exotic parents are perfect examples of what breeders had in hand in 1950 (because they are exactly what Royce S. Bringham and Victor Voth had in hand when they initiated their Green Revolution breeding work in the 1950s). We are confident that the genetic gain estimates we reported from our single location study (in Salinas, CA) over two years accurately reflect the genetic changes that have occurred in the California population over the history of the UC Davis strawberry breeding (1927-present).

We color coded the elite x elite and elite x exotic hybrids in Figure 4 to emphasize the extreme differences we observed for different traits in the genetic gains study. Genetic gains were estimated between hybrids from *opposite ends* of the distributions shown in Figure 4. Hybrids in the lower tails of the fruit yield, count, weight, and firmness distributions (red points in Figure 4) are the least domesticated (most exotic) and represent cultivars that existed before the 1950s. Conversely, hybrids in the upper tails of the fruit yield, count, weight, and firmness distributions (blue points in Figure 4) are the most domesticated (most elite) and include modern UC Davis cultivars that dominate production today. If our study were to be repeated in additional coastal California environments, we are confident that significant *magnitude* or *cross-over* G x E interactions (as depicted by the hypothetical scenarios shown in the middle and right panels of Figure R1) would be observed between the lowest yielding elite x exotic and highest yielding elite x elite hybrids (Figure 4).

Finally, we emphasized in the revised manuscript that the genetic gain estimates we reported for yield in California are aligned with the 2,755% yield increase reported by the UN-FAO for the US from 1961 to 2022. Even if our genetic gain estimates were cut in half they would be still be substantial and mirror the linear increase in yield reported by the UN-FAO from 1961 to present. We have made a strong case that genetic gains from breeding over the last 60 years at UC Davis have had a profound impact on strawberry production in the US (and world for that matter) and drove the strawberry Green Revolution in California. We reported that 764 cultivars documented in our global database have UC Davis parents in their pedigrees (L126-132). We pointed out that 18 of the 20 highest yields reported by the UN-FAO have been in the US over the last 20 years (L314-318). We added these metrics not to boast (we cannot take credit for historical genetic gains) but to document the incredible achievements of others and the global impact of the UC Davis breeding program on strawberry production around the globe. We cite papers that show that the Driscoll's strawberry breeding program (a household name in many parts of the world) was built on the same genetic foundation as the UC Davis breeding program. Within California, UC Davis has 61% market share, while Driscoll's reportedly has 19% market share (their exact share is unclear because of the opaqueness of reporting by the private-sector).

Accuracy of Genomic Predictions

Since the last review comments were received, the first author (Mitchell Feldmann) has developed a complete draft of a companion paper that provides extensive analyses of genomic prediction accuracies for the training population hybrids used in our genetic gain study. The title of that paper is "Intersection of Quantity and Quality: Genomic Prediction for Improving Productivity and Fruit Quality in Strawberry". That paper focuses entirely on genomic selection and provides a wealth of genomic prediction accuracy estimates for the training population we used. We are submitting that paper to The Plant Genome next week. We realize that the genomic selection paper has not been accepted for publication, but it provides in-depth answers to the genomic prediction accuracy questions raised by reviewers.

Reviewers' Comments:

Reviewer #4:

Remarks to the Author:

In the revised manuscript, the authors delved deeper into the limitations of their studies, with a specific focus on genotype-by-environment interactions. They acknowledge that the magnitude of the genetic gains they reported could be overestimated because of the underestimation of G x E interactions. But they tried every effort to support that the direction of the genetic gains is 100% accurate and would not change. Overall, the revised manuscript and their response appear sound.

If the authors have adequately addressed genomic prediction accuracy estimates for the training population in a companion paper, it is acceptable for them to reference those results in this manuscript.